# Accelerating Leigh syndrome drug discovery through deep learning screening in brain organoids

Carmen Menacho [1,2,40], Satoshi Okawa [3,4,40], Iris Álvarez-Merz [5], Annika Wittich [6], Mikel Muñoz-Oreja[7], Pawel Lisowski[8,9,10,11], Mario López Martín[12], Tancredi Massimo Pentimalli [8,9,13], Shiri Zakin[14], Mathuravani Thevandavakkam[14], Caleb Jerred[1,2,15], Selene Lickfett [1,2], Laura Petersilie[5], Agnieszka Rybak-Wolf [8,9], Annette Seibt[1], Diran Herebian[1], Gizem Inak[8,38], Susanne Brodesser [16], Andrea Zaliani [6], Barbara Mlody[8,39], Justin Donnelly[13], Kasey Woleben[17], Francesc Xavier Soriano [18], Jose C. Fernandez-Checa[19,20,21,22], Natascia Ventura [15,23,24], Sidney Cambridge[25], Ertan Mayatepek [1], Antonella Spinazzola [26], Markus Schuelke [27,28], Nikolaus Rajewsky [8,9,28,29,30,31], Andrea Rossi [32], Alex Peralvarez-Marin [12], Felix Distelmaier [1], Ethan Perlstein[14], Ian J. Holt[7,33,34,35], Emma Puighermanal [36], Ole Pless [6], Christine R. Rose [5], Antonio Del Sol [3,33,37,41] ✉ & Alessandro Prigione [1,41] ✉

Leigh syndrome (Leigh) is an untreatable mitochondrial disorder characterized by lactic acidosis and basal ganglia and midbrain pathology, leading to psychomotor regression and early death. We previously uncovered impaired neuronal morphogenesis in Leigh cerebral organoids carrying *SURF1* gene variants. Leveraging this phenotype, we here develop a deep learning algorithm tailored for cell type-specific drug repurposing screening. In parallel, we perform a survival drug screen in a yeast model of Leigh. The two approaches independently converge on azole compounds, two of which - talarozole and sertaconazole - rescue neuronal morphogenesis in Leigh neurons and lower lactate release and improve growth rate in Leigh midbrain organoids. Mechanistically, these compounds modulate the retinoic acid pathway and membrane-associate lipid metabolism. The findings highlight azoles as promising candidates for Leigh and demonstrate the potential of combining in silico screens with human brain organoids as new approach methodologies (NAMs) to advance the discovery of therapeutics addressing rare neurodevelopmental disorders.

Leigh syndrome (Leigh) (OMIM #256000), also known as "infantile subacute necrotizing encephalomyelopathy", is a severe neurometabolic disorder affecting children[1]. Like other mitochondrial diseases, Leigh is caused by inherited gene variants in the nucleus or mitochondrial DNA (mtDNA) hampering the oxidative phosphorylation (OXPHOS) machinery that is responsible for cellular energy generation[2]. Hence, mitochondrial diseases typically impair tissues with high energy requirements such as nerves and muscles[3]. In

particular, Leigh affects the central nervous system regions of basal ganglia and midbrain, leading to motor impairment, intellectual disability, lactic acidosis, and early mortality[4–8].

One of the most frequently mutated gene in Leigh is *SURF1* (surfeit locus protein 1, NM_003172.2), which encodes an assembly factor for cytochrome c oxidase (COX), the complex IV of the mitochondrial respiratory chain and terminal enzyme responsible for transferring electrons from cytochrome c to molecular oxygen that is then reduced to water[4,5]. The *SURF1* gene is highly conserved and ablation of the yeast homologue *SHY1* (ΔSHY) retards growth upon nutrient removal[6]. A *SURF1* knock-out piglet model displayed severe lethal neurodevelopmental phenotypes[7]. However, *Surf1* knock-out mice failed to develop Leigh-like phenotypes and instead exhibited prolonged lifespan[8,9], suggesting that mice possess compensatory mechanisms for COX deficiency that humans lack[9]. Due to the paucity of faithful mammalian models, it has been challenging to study Leigh caused by *SURF1* defects to identify possible interventions[10,11]. Indeed, Leigh is currently incurable[12] and there is a high need for effective strategies aiming at accelerating the drug discovery process.

To develop innovative models of Leigh[13,14] based on human-relevant new approach methodologies (NAMs), we previously combined patient-derived induced pluripotent stem cells (iPSCs) and CRISPR/Cas9 engineering to generate a brain organoid model of Leigh harboring *SURF1* variants[15]. Compared to isogenic controls, Leigh cerebral organoids (COs) exhibited aberrant progenitor organization, suggestive of defective neuronal morphogenesis[15]. Accordingly, Leigh iPSC-derived dopaminergic neurons showed impaired differentiation and branching capacity that resulted in defective firing[15]. The findings suggested that Leigh may interfere with neurodevelopment and particularly with the ability of neurons to form appropriate connections, leading to faulty wiring[16]. Cortical brain organoids of Leigh caused by other gene variants demonstrated a similar impairment in neurodevelopment[16,17]. Mitochondria might indeed play a central role in neural commitment[18–20], and differences in mitochondrial activity might underlie the species-specific temporality of brain development[21]. Pre-clinical and clinical studies revealed that signs of Leigh and other mitochondrial disorders may manifest before birth[22–25], potentially explaining the developmental delay seen in individuals with Leigh[26–28]. Collectively, these data suggest that Leigh impacts human brain development. Hence, approaches aiming at counteracting these defects and promoting neuronal morphogenesis might be of therapeutic value.

Here, we sought to exploit our brain organoid model of Leigh to uncover repurposable drugs. We recently showed that screening repurposable compounds in iPSC-derived neural cells led to the identification of sildenafil as a potential treatment of Leigh caused by *MT-ATP6* variants[29]. To further accelerate this process and allow the identification of repurposable candidates before performing time-consuming screens in cells, we establish here a deep learning (DL) algorithm for cell type-specific drug repurposing based on single-cell RNA-sequencing (scRNAseq) of Leigh COs. Concurrently, we screen a library of 2,250 FDA-approved drugs in ΔSHY yeasts. Both approaches independently identify azole compounds, two of which - sertaconazole and talarozole - effectively promote neural commitment in Leigh neural cells and rescue the aberrant growth rate and lactate release of Leigh midbrain organoids (MOs). In addition to modulating neuronal bioenergetics, sertaconazole and talarozole impact membrane-associated lipid metabolism and engage with the CYP26A enzyme activating the retinoic acid (RA) pathway. Taken together, we discover two drug candidates that could be repurposed for the treatment of individuals with Leigh. Our findings suggest that human neuronal morphogenesis may

represent a therapeutic target for Leigh and underscore the power of combining in silico screens with brain organoid technology to develop human-based platforms for accelerating drug discovery of untreatable neurodevelopmental disorders.

## Results

### Deep learning-based framework for drug repurposing and yeast screen identify repurposable drugs for Leigh

To accelerate Leigh drug discovery, we employed deep learning (DL), a subset of machine learning that uses multi-layered artificial neural networks to learn from large amounts of data. We established a DL algorithm for cell type-specific drug repurposing based on scRNAseq of Leigh COs. We reanalyzed our published scRNAseq datasets of isogenic Leigh COs[15] to determine key affected populations. Uncommitted precursors showing markers of radial glia (RG) identity were present in both Leigh COs and control COs, while more committed intermediate progenitor cells (IPCs) and neurons were mostly absent in Leigh COs (Fig. 1a-b). Using pseudotime analysis[30] to reconstruct the continuous trajectory of cell development in COs, we confirmed that neuronal differentiation was following the directionality RG > IPCs > neurons (Fig. 1c-d, Supplementary Fig. 1a). Therefore, strategies capable of promoting neuronal fate in Leigh cells might represent innovative treatment interventions.

We developed a multi-stage, DL-based framework based on ChemPert, our recently developed database of perturbations in non-cancer cells[31]. The goal of the framework was to predict optimal drugs that drive a transcriptional state transition from an initial state (i.e. RG identity) to a desired target state (i.e. IPCs or neurons) (Fig. 1e). We first inferred protein-level perturbations as activation (1), inhibition (−1), or no perturbation (0). Unlike conventional drug-response tools, our framework learned the upstream interventions required to induce a specific downstream gene expression profile (Supplementary Fig. 1b-f). By decoupling protein-level perturbation prediction from the drugs used in training, this approach enabled prioritization of candidate drugs, including those not present in the training data, thereby making the framework suitable for non-cancer drug repurposing, where perturbation transcriptomics datasets are often limited in scope and coverage compared to oncology-focused resources. The framework operated over a curated molecular interaction graph comprising 20,455 protein nodes, assembled from integrated protein–protein interactions, signaling pathways, transcriptional regulation, and ligand–receptor relationships. We evaluated the framework using a strictly independent hold-out 47 test datasets associated with diverse cell conversion processes, including differentiation, metabolic shifts, and cell fate reprogramming (Supplementary Data 1, Supplementary Data 7). We compared our DL approach against gene set enrichment analysis (GSEA)-based methods and random ranking. Within the top 1st percentile of ranked drugs, our framework recalled 9.2% of known hits, outperforming GSEA (best case: 2.1%) and random ranking (1%). These results demonstrated that our DL framework added value beyond conventional pathway enrichment by enabling more accurate early prioritization of candidate drugs for experimental validation.

Given our perturbation input (transcriptomic profile of initial state and target state), the framework computed differentially expressed transcription factors (DETFs). We used this defined set of DETFs to uncover drugs capable of triggering cellular conversion from RG directly to neurons (DETF1) or from RG to IPCs and then to neurons (DETF2) (Fig. 1e, f). DETF influence was then propagated through the graph up to five hops, capturing both upstream and downstream effects. The framework also computed topological and functional attributes for each protein, such as shortest path lengths to DETFs, DETF reachability, and shared Reactome pathway counts, which were not part of the propagation step but were integrated as additional features. These features were then passed through a variational autoencoder (VAE) trained to denoise and regularize the unconstrained

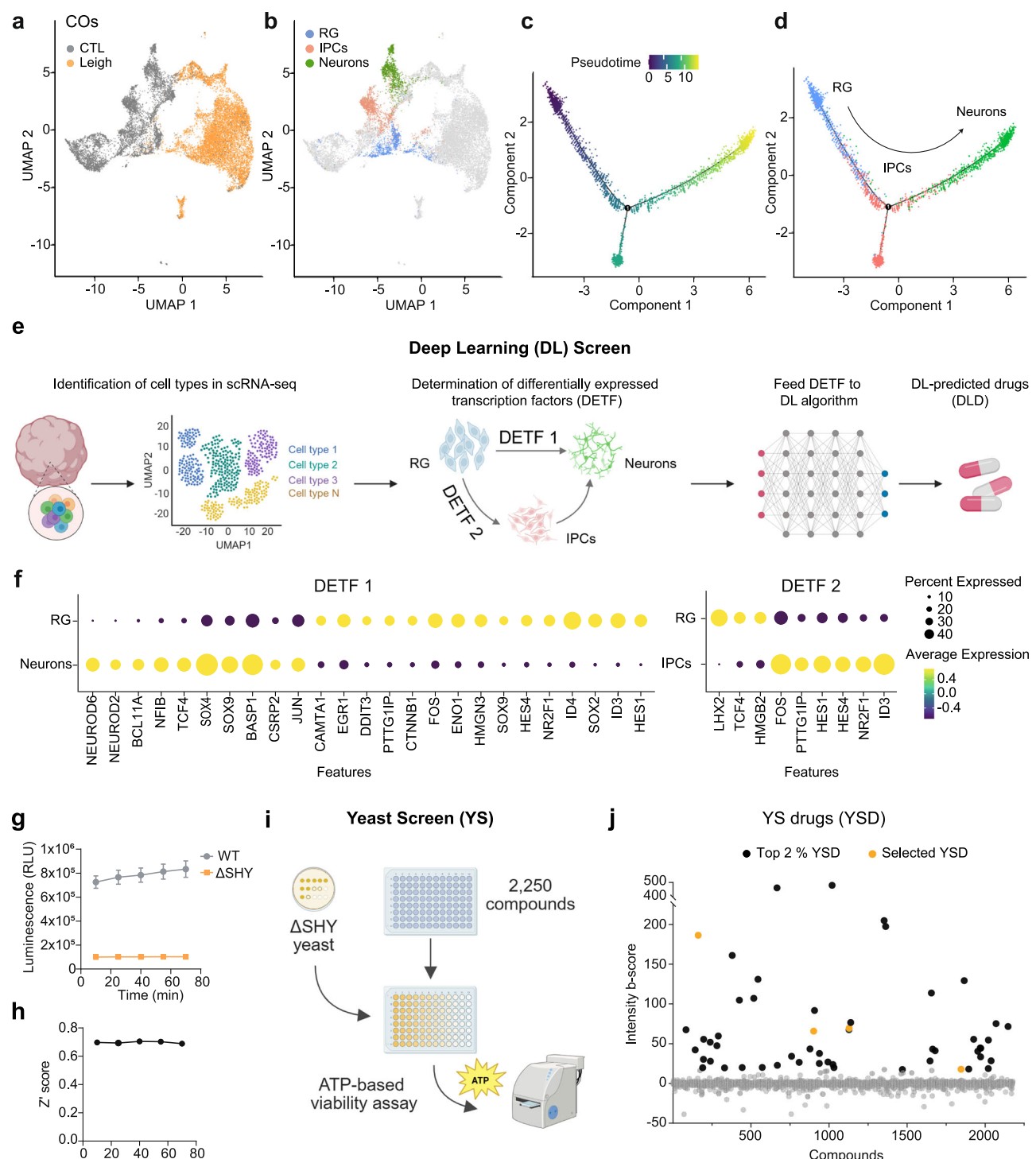

**Fig. 1 | Deep learning and yeast screens to identify repurposable drugs for Leigh.** **a**, **b** UMAP plots of scRNAseq of cerebral organoids (COs) from Leigh and isogenic control (CTL) showing uncommitted progenitors with radial glia (RG) identity and neural committed cells with identity of intermediate progenitor cells (IPCs) and mature neurons. **c**, **d** Pseudotime analysis reconstructing the developmental trajectory of the three populations following the directionality RG>IPCs>-Neurons. **e**, **f** Schematic of the deep learning (DL) algorithm to identify drugs promoting differentially expressed transcription factors (DETF)-specific cell fate conversion (created in BioRender. Menacho, C. (2026) https://BioRender.com/

siziatp). **g**, **h** Survival assay in wild-type (WT) yeast and knock-out yeast for the *SURF1* homologue *SHY1* (ΔSHY). Dots in (**g**) show mean +/- SD of *n* = 3 independent experiments (**i**) Schematic representation of yeast screen (YS) strategy (created in BioRender. Menacho, C. (2026) https://BioRender.com/vaey7ea). **j** Screen of 2,250 FDA-approved repurposable compounds in ΔSHY yeast. Dots: individual drug responses to 20 μM treatment at 24 h; black dots: top 2% yeast screen drugs (YSDs); orange dots: YSDs selected for validation. Source data are provided as a Source Data file.

graph-based input (Supplementary Fig. 1b). The decoded output from the VAE, which preserved the original feature dimensionality, was then used as input to the classifier. The decoded input was processed by an ensemble of three independently trained feedforward neural networks (FNNs). Each model predicted one of three perturbation labels (1, −1 or 0) for every protein in the network. The ensemble output was determined by majority voting across the three FNN models (Supplementary Fig. 1b). The framework computed a Bayesian enrichment score (BES) for each drug. This score integrated: 1) hypergeometric enrichment between predicted perturbed proteins and known drug targets, 2) empirical robustness (*p*-value) calculated from randomized input feature matrices, and 3) normalized rank weighting (Supplementary Fig. 1c–f). The final BES reflected both statistical significance and model confidence. Drugs with no overlap (hypergeometric $p = 1.0$) received a BES of zero, and remaining drugs were ranked in descending BES order. To prioritize drug candidates from the prediction set, we selected drugs ranked within the top 2.5th percentile of the BES. This cutoff was chosen based on the test recall curve, which plateaued at 2.5% (Supplementary Fig. 1f-g), indicating diminishing returns for lower-ranked drugs. This cutoff resulted in 140 drugs for DETF1 and 143 drugs for DETF2, which we further prioritized based on annotation of metabolic function, given the central role of mitochondrial metabolism in Leigh pathophysiology and neural differentiation processes. This strategy highlighted a total of 34 DL-predicted drugs (DLDs). Among these, we selected 5 based on their safety profile in humans (DLD1, DLD3, DLD4, DLD5 from DETF1, and DLD2 from DETF2).

In parallel to the DL algorithm approach, we carried out a rescue screen in Leigh yeast cultures using a library of 2,250 repurposable drugs. The rescue assay was based on survival of wild-type (WT) and *SURF1* homologue *SHY1*[6] knock-out (ΔSHY) yeast strains upon nutrient removal (Fig. 1g–i, Supplementary Fig. 2a-b). Among the top 2% of rescuing yeast screen drugs (YSDs) (Fig. 1j, black dots), we selected 4 (YSD1, YSD2, YSD3, YSD4) that belonged to the class of azole-containing compounds, as this was among the most strongly affected classes of rescuer drugs (Fig. 1j, orange dots). Altogether, using the two screening strategies, we identified 9 candidate repurposable compounds.

### Identified compounds sertaconazole and talarozole promote Leigh neuromorphogenesis

We previously demonstrated that Leigh variants impact the ability of neural progenitor cells (NPCs) to switch to OXPHOS and commit to neuronal fate[15]. Here, we used our high-content analysis (HCA) pipeline to quantify neuromorphogenesis[32] and investigate NPC commitment defects in Leigh. Through overexpression of the transcription factor Neurogenin 2 (*NGN2*)[33], we converted Leigh NPCs and isogenic control NPCs into induced neurons (iNs) in about 5-6 days (Supplementary Fig. 2c). In agreement with previous data obtained in Leigh midbrain dopaminergic neurons (mDANs)[15], the number of neurons was significantly reduced in Leigh iNs compared to control iNs (Fig. 2a, Supplementary Fig. 2d). The mean neurite length was slightly lower in Leigh iNs compared to control iNs, although the difference did not reach statistical significance (Supplementary Fig. 2e).

We next applied the HCA pipeline to assess the impact of the 9 candidate compounds (5 DLDs and 4 YSDs) on neuronal number and morphology. We treated Leigh iNs differentiating from NPCs with three different concentrations for compound (all dissolved in DMSO) and compared compound treated Leigh iNs to Leigh iNs exposed to DMSO only (Fig. 2b). We counterstained the plates with the nuclear dye Hoechst to monitor the effect of compounds on cellular viability.

Among DLDs, talarozole (DLD5) showed the most pronounced effect, increasing by two-fold the number of neurons in Leigh iNs in a Concentration-dependent manner for concentration of 1 μM and 10 μM (Fig. 2c, d). 10 μM talarozole also increased the mean neurite length in Leigh iNs (Supplementary Fig. 2g). Based on Hoechst signal, no

neuronal toxicity was observed at 1 μM and 10 μM concentration, but 50 μM talarozole treatment led to 50% cell death (Supplementary Fig. 2f). Interestingly, talarozole showed one of the highest BES values in the DL screen (BES = 9.034277; 38th out of 5692 drugs, -99.3rd percentile) (Supplementary Fig. 1h, i). We also assessed three non-predicted drugs (NPDs) that were outside the top 2.5% BES cutoff and found that they did not increase the number of neurons in Leigh iNs (Supplementary Fig. 2j, k).

Among YSDs, we identified sertaconazole (YSD4) as the most promising compound. 10 μM sertaconazole led to 1.5-fold increase in neuronal amount and to higher mean neurite length in Leigh iNs (Fig. 2e, f, Supplementary Fig. 2i). Similarly to talarozole, 50 μM sertaconazole proved to be toxic in Leigh iNs (Supplementary Fig. 2h). Therefore, we focused only on lower concentrations for further analyses with these two azole compounds.

Next, we assessed whether the two azoles could lead to a concentration-dependent rescue of ΔSHY yeast upon nutrient removal. Talarozole was effective in increasing by more than two-fold the survival of mutant yeast in a concentration-dependent manner both at 24 and 48 h of treatment (Fig. 2g). Sertaconazole also displayed a concentration-dependent rescue effect in mutant yeast up to almost two-fold at 24 and 48 h (Fig. 2h).

Overall, based on validations conducted in SURF1-mutant human Leigh neurons and mutant yeast, we concluded that the azoles sertaconazole and talarozole were promising drug candidates to be repositioned for Leigh.

### Midbrain organoids as a 3D model of Leigh recapitulating disease features

We reasoned that the next step for assessing the potential effectiveness of the identified azoles would be to address their effect in a more complex human model. We previously employed whole-brain cerebral organoids (COs) to model Leigh[15]. However, since basal ganglia and midbrain regions are typically affected in Leigh[34–36], here we aimed to generate Leigh midbrain organoids (MOs) to assess the impact of compounds on this brain region under the condition of SURF1 deficiency.

In light of the defective neuronal generation of Leigh NPCs (Fig. 2a), we inferred that impaired neural commitment of NPCs contributes to the disease pathogenesis. Therefore, we generated MOs starting from NPCs[37] (Supplementary Fig. 3a) and used scRNAseq to benchmark our MOs with respect to previously published MOs derived from NPCs[38] (Supplementary Fig. 3b). The MOs that we generated contained a relatively high number of tyrosine hydroxylase (TH)-positive mDANs (31%) by day 35 (Supplementary Fig. 3c), indicating that our protocol led to the generation of functionally relevant cell types at this early time-point. We focused our analysis on MOs grown until day 30, when we could detect the expression of both progenitor cells (positive for SOX2 and NESTIN) and mature neurons (positive for MAP2 and for mDAN markers TH and NURR1) (Supplementary Fig. 3d).

To determine whether Leigh MOs could recapitulate neuromorphogenesis defects seen in Leigh COs (Fig. 1a-d), we carried out scRNAseq in Leigh MOs compared to isogenic control MOs (Fig. 3a). Unsupervised clustering identified eight main clusters in MOs (Supplementary Fig. 4a, Supplementary Data 2). Based on their gene expression, we assigned the clusters to the following categories: proliferative cells (cluster 0), aberrant neurons (cluster 1), aberrant undifferentiated cells (cluster 2), healthy neurons (cluster 3), RGs (clusters 4 and 6), other cell types (cluster 5), and IPCs (cluster 7) (Supplementary Fig. 4b, Supplementary Data 2). Dopaminergic markers were mainly expressed in the population of neurons and IPCs (Supplementary Fig. 4c). Based on this classification, we performed pseudotime analysis of RG, IPCs and healthy neurons (Fig. 3b-e, Supplementary Fig. 4e). Similar to the results obtained in COs, we obtained a directionality of differentiation from RG > IPCs > neurons (Fig. 3c-d).

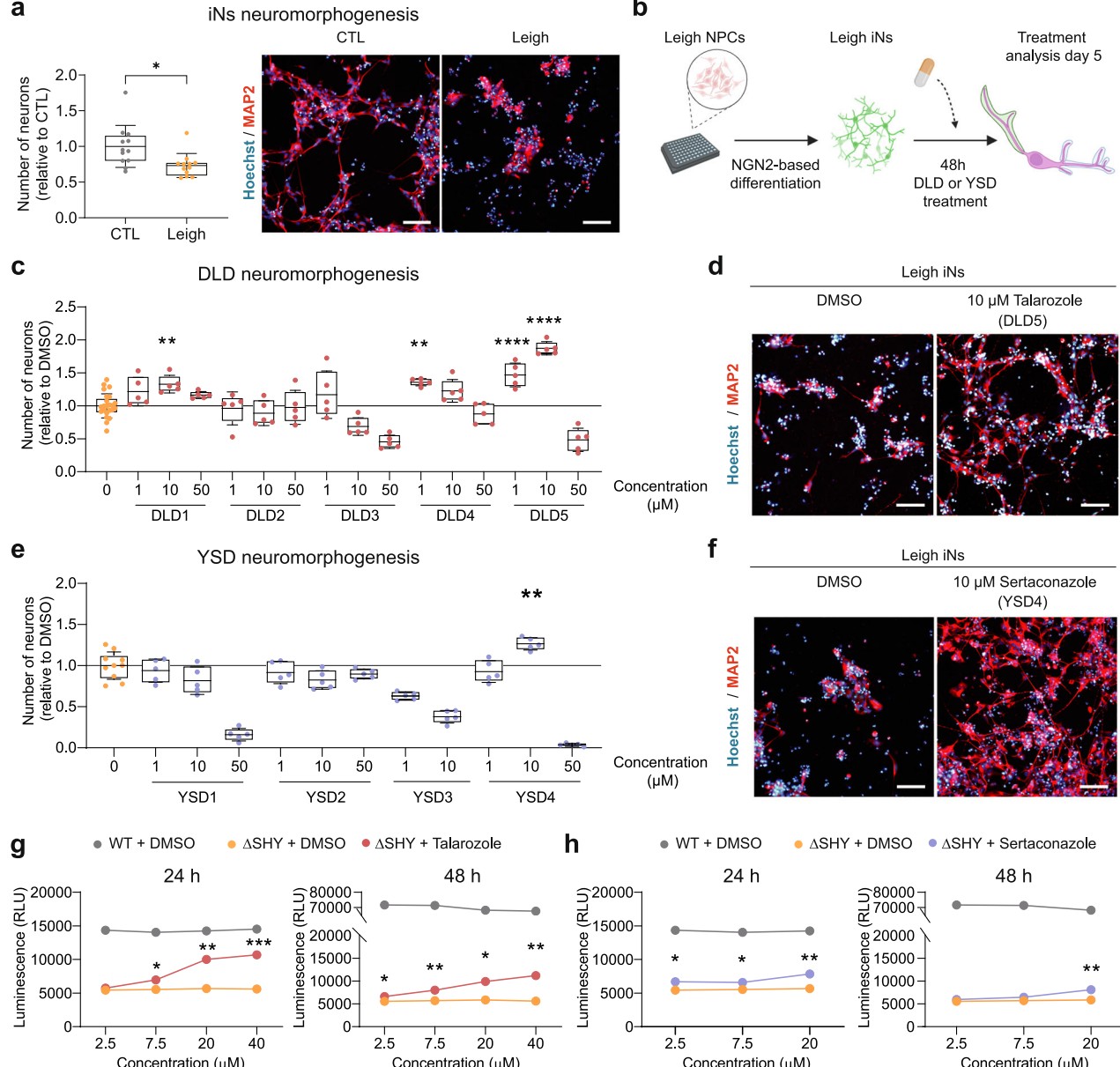

**Fig. 2 | Neuromorphogenesis assay in Leigh iNs validate talarozole and serta-conazole. a** Number of induced neurons (iNs) from Leigh NPCs and isogenic control (CTL) NPCs quantified with high-content analysis (HCA) based on neuronal dendritic marker MAP2 at day 5 of differentiation. Boxplots: mean (center), 25/75 percentiles (box), and 1x SD (whiskers). Individual dots: $n = 12$ biological replicates; *$p < 0.05$, unpaired two-tailed t test. Scale bars: 100 μm. **b** Schematic of the 2D neuromorphogenesis screening in Leigh iNs using DL-predicted drugs (DLDs) and yeast screen drugs (YSDs) (created in BioRender. Menacho, C. (2026) https://BioRender.com/31a4745). **c–f** Quantification of Leigh iNs neurons after treatment with DMSO ("0") or with DLDs or YSDs and representative images for DLD5 (talarozole) and YSD4 (sertaconazole). Boxplots: mean (center), 25/75 percentiles (box), and 1x SD (whiskers). Individual dots: $n = 5$ biological replicates, related to DMSO-treated condition. Statistics are shown only for ameliorating compounds, **$p < 0.01$, ***$p < 0.005$, ****$p < 0.001$, one-way ANOVA, compound-treated Leigh iNs vs DMSO-treated Leigh iNs. Scale bars: 100 μm. **g, h** Concentration-dependent validation of talarozole and sertaconazole on yeast survival after 24 h or 48 h of increasing concentrations. Dots: mean +/- SD; n = 3 replicates, *$p < 0.05$, **$p < 0.01$, ***$p < 0.001$, two-way ANOVA. Source data are provided as a Source Data file.

These results validated our Leigh MOs model and confirmed that it recapitulated key features seen in Leigh COs.

We next investigated the differential gene expression between Leigh and controls in the identified populations within MOs. The Leigh signature of the IPCs population highlighted the downregulation of genes associated with neuronal morphogenesis (*NFIB, ROBO3, MAGI2*), generation of mDANs (*MEST*), mitochondrial movements (*KIF5C*), and retinoic acid (RA) metabolism (*CYP26B1*) (Fig. 3f, Supplementary Data 3). Correspondingly, pathway enrichment analysis revealed downregulation of

"nervous system development" and "generation of neurons" pathways in Leigh IPCs (Supplementary Fig. 5b). Upregulated genes in Leigh IPCs comprised genes regulating growth and proliferation (*SLC35F2, KLF4, PKIB, MYC, EEF2*), lipid metabolism (*FASN*), and mitochondrial respiratory chain genes (*MT-ATP6, MT-ND4*), including those belonging to central COX enzyme that is impaired by *SURF1* variants (*MT-CO1, MT-CO2, MT-CO3*) (Fig. 3f, Supplementary Data 3). Upregulated pathways in Leigh IPCs included "ribosome biogenesis", "cellular respiration", and "cholesterol biosynthetic process" (Supplementary Fig. 5a).

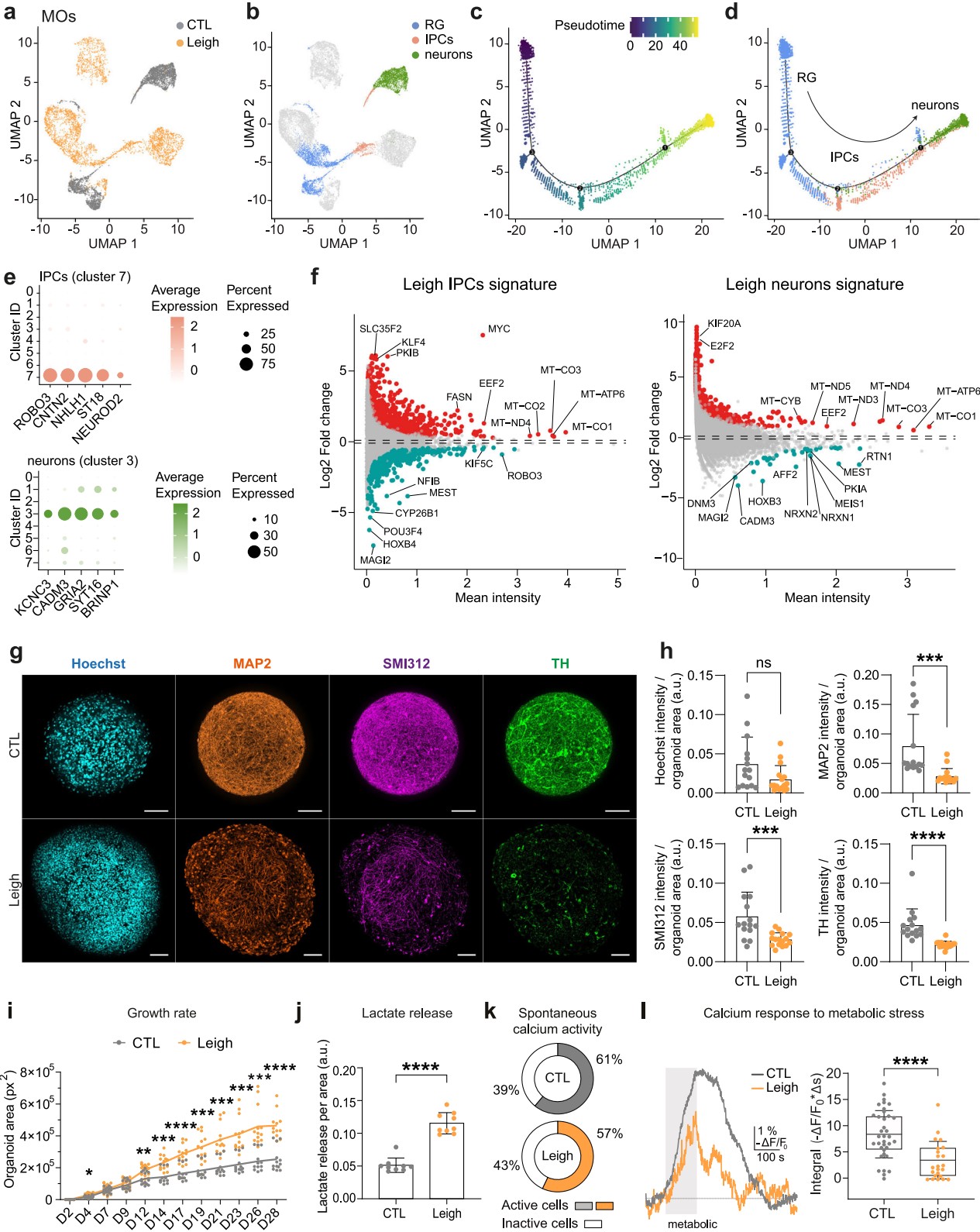

In the neuronal population, the Leigh signature highlighted the downregulation of genes involved in neurogenesis (*CADM3, MAGI2, RTN1, AFF2*), mDANs development (*MEST*), and synaptic function (*DNM3, NRXN1, NRXN2*) (Fig. 3f, Supplementary Data 3), and the downregulation of pathways associated with "synapse assembly" (Supplementary Fig. 5d). Similarly to IPCs, Leigh neurons exhibited upregulation of genes related to cellular proliferation (*EEF2, KIF20A*)

and mitochondrial respiration (*MT-ATP6, MT-ND5, MT-CO1, MT-CO2, MT-CO3*) (Fig. 3f, Supplementary Data 3). Upregulated pathways in Leigh neurons included "DNA metabolic process" and "regulation of cell cycle process" (Supplementary Fig. 5c).

RG appeared less impacted by Leigh pathology, as seen by the scRNAseq clustering of COs (Fig. 1a-b) and MOs (Fig. 3a, b). Nonetheless, the Leigh signature of RG still encompassed the

**Fig. 3 | Midbrain organoids (MOs) recapitulate Leigh disease features.**
**a, b** UMAP plots of scRNAseq of Leigh MOs and CTL MOs showing uncommitted progenitors with RG identity and neural committed cells with identity of IPCs and mature neurons. **c, d** Pseudotime analysis of RG, IPCs, and neurons in MOs. **e** Dot plot highlighting top five genes enriched in IPCs (cluster 7) and neurons (cluster 3). **f** Pseudo-bulk analysis depicting the disease signature in the progenitor population (Leigh IPCs) and the neuronal population (Leigh Neurons) within Leigh MOs compared to CTL MOs. **g, h** Organization and quantification of MAP2-positive dendrites, SMI312-positive axons, and tyrosine hydroxylase (TH)-positive neurons in CTL MOs and Leigh MOs. Bar plots: mean +/- SD of signal intensity normalized to organoid area. Dots: individual MOs (CTL: 15 MOs, Leigh: 16 MOs); **$p < 0.01$, unpaired two-tailed t test. Scale bars: 100 μm. **i** Growth rate of MOs relative to the initial size on day 2. Dots: individual MOs (CTL: 13-16 MOs; Leigh: 13-16 MOs); $n = 3$ independent experiments; *$p < 0.05$, **$p < 0.01$, ***$p < 0.005$, ****$p < 0.001$, two-way ANOVA. **j** Lactate release from MOs normalized to organoid area. Bar plots: mean +/- SD. Dots: individual MOs (CTL: 9 MOs; Leigh: 9 MOs); ****$p < 0.001$, two-tailed Mann-Whitney U test. **k** Active cells with spontaneous calcium activity: 25 out of 41 in 7 CTL MOs, 20 out of 35 in 6 Leigh MOs. Cells with calcium signals ≥ 2 x SD were considered as active. **l** Calcium response in MOs following metabolic stress (2 min of glucose-free medium + 2 mM 2-deoxyglucose + 5 mM sodium azide). Boxplots: mean (center), 25/75 percentiles (box), and 1x SD (whiskers). Dots: individual cells within MOs (CTL: 7 MOs; Leigh: 6 MOs); ****$p < 0.001$, two-tailed Mann-Whitney U test. Source data are provided as a Source Data file.

downregulation of neuronal genes (*RELN, TUBB3, MEST*) and the upregulation of genes related to mitochondrial respiration (*MT-ND4, MT-CO1, MT-CO2, MT-CO3*) and lipid metabolism (*FASN*) (Supplementary Fig. 4d, Supplementary Data 3).

In agreement with the transcriptional signature suggestive of defects in neuronal morphogenesis and mDANs generation, we detected significantly reduced expression of axonal marker SMI312, dendritic marker MAP2, and mDANs marker TH in Leigh MOs compared to isogenic control MOs (Fig. 3g, h). Leigh MOs were significantly larger than control MOs (Figs. 3g, i), in line with the observed increase in proliferation and cell cycle-related genes observed by transcriptomics.

Leigh MOs released more lactate in the media compared to control MOs (Fig. 3j). Elevated lactate in blood and cerebrospinal fluid represents a key diagnostic indicator of Leigh alongside specific brain abnormalities[39]. This feature is usually interpreted as a consequence of defective OXPHOS, which increases the reliance on glycolysis for ATP production, leading to a buildup of lactate[40–42]. The increase in lactate release was in agreement with the transcriptomics findings of Leigh MOs indicating a dysregulation of mitochondrial metabolism.

Lastly, we assessed the functionality of MOs by measuring calcium activity in whole organoids, as previously described[37]. Both Leigh MOs and control MOs were functionally active, with overall similar number of cells showing spontaneous calcium activity (Fig. 3k, Supplementary Fig. 5e–g). However, under condition of acute energy deprivation (2 min of glucose starvation with inhibition of glycolysis and OXPHOS), Leigh MOs displayed an aberrant response to metabolic stress, with a reduced ability to increase the calcium signal in comparison to control MOs (Fig. 3l, Supplementary Fig. 5h, i). This impaired response to metabolic stress could reflect a critical pathological feature of Leigh[1,43].

In summary, Leigh MOs showed transcriptional profiles reminiscent of those seen in Leigh COs and recapitulated key disease features including increased lactate release and aberrant response to metabolic stress. These findings suggest that Leigh MOs could serve as an effective model of Leigh on which to test treatment interventions.

### Sertaconazole and talarozole rescue Leigh phenotypes in midbrain organoids

We next evaluated the effect of sertaconazole and talarozole on our MO model of Leigh (Fig. 4a). We established 0.1 μM sertaconazole or 1 μM talarozole as drug concentrations that could be used for long-term treatment of MOs with no apparent toxic effects in healthy control MOs (Supplementary Fig. 6a-b). At these concentrations, the two azoles significantly reduced the abnormal growth rate of Leigh MOs, although not to the levels seen in isogenic control MOs (Fig. 4b). Treatment of Leigh MOs with sertaconazole and talarozole decreased the amount of lactate in the media by 20% (Fig. 4c). Although the azoles did not significantly rescue neuronal generation or morphology in Leigh MOs (Supplementary Fig. 7a-b), talarozole treatment did increase the number of TH-positive neurons within MOs (Fig. 4d). Talarozole treatment also resulted in a higher number of active cells within Leigh MOs based on their calcium activity (Fig. 4e, Supplementary Fig. 6c) and increased by two-fold the calcium signal of Leigh MOs in response to metabolic

stress (Fig. 4f, g, Supplementary Fig. 6d). These results indicate that while identified azoles did not fully revert the disease phenotypes of Leigh MOs, they led to significant rescue of key pathological aspects, with talarozole showing higher effectiveness than sertaconazole.

We then investigated the effects of the azole compounds on mitochondrial anchoring, which is crucial for neurite branching[44,45]. Indeed, the levels of mitochondrial anchoring-related genes were lower in Leigh mDANs and Leigh COs compared to controls (Supplementary Fig. 6e) and the anchoring gene *KIF5C* was downregulated in Leigh MOs (Fig. 3f, Supplementary Data 3). Concordantly, Leigh mDANs displayed abnormally increased mitochondrial motility with fewer mitochondria that were anchored or stationary (Supplementary Fig. 6f–h). These findings were not recapitulated by chemical inhibition of mitochondria using the uncoupling agent FCCP, which instead led to decreased mitochondrial motility (Supplementary Fig. 6g-h). Hence, the increase in mitochondrial motility seen in Leigh mDANs might not be due to simple mitochondrial impairment but rather to specific disease-related mechanisms, possibly linked to defective mitochondrial anchoring. Nonetheless, treating Leigh mDANs or Leigh MOs with sertaconazole or talarozole did not rescue the expression of anchoring proteins (Supplementary Fig. 6i-l).

To gain mechanistic insight into the action of the azoles, we performed scRNAseq of Leigh MOs treated for 30 days with sertaconazole or talarozole (Fig. 4a, Supplementary Fig. 7c, Supplementary Data 4). The azoles primarily affected the neuronal population of Leigh MOs. The sertaconazole signature in Leigh neurons included the upregulation of genes related to lipids and cholesterol (*STARD4, FABP3, MVD, SCD, HMGCR, HMGCS1*) and neuronal generation (*TUBB3, ELAVL3, MAP1B, DPYSL2*) (Fig. 4h), and the upregulation of pathways associated with cholesterol and lipid metabolism (Fig. 4i). Downregulated genes by sertaconazole in Leigh neurons included mitochondrial genes (*MT-ND3, MT-CO1, MT-CO2, MT-CO3*) and genes related to RA pathway (*CRABP1*) (Fig. 4h).

Talarozole similarly upregulated genes related to lipids and cholesterol (*SCD, HMGCS1*) and neuronal generation (*TUBB3, ELAVL3, FXYD1, GJA1*) in Leigh neurons (Fig. 4j), and upregulated pathways associated with cholesterol and lipid metabolism (Fig. 4k). It downregulated genes regulating cell morphology and embryonic growth (*LRATD1, EGR3*) (Fig. 4j). The effect of the two drugs on the IPCs population within Leigh MOs was less pronounced but still showed an upregulation of genes associated with neuronal generation and lipid metabolism (Supplementary Fig. 7d).

Thus, single-cell transcriptomics indicated that the two azoles shared an effect on lipid metabolism and neuronal generation. These findings suggest that the two compounds counteracted key transcriptional changes induced by Leigh in the neuronal population of midbrain organoids.

### Sertaconazole and talarozole impact bioenergetics and lipid metabolism

To validate the transcriptomics results, we investigated the impact of the two azoles on energy metabolism and lipid metabolism in Leigh

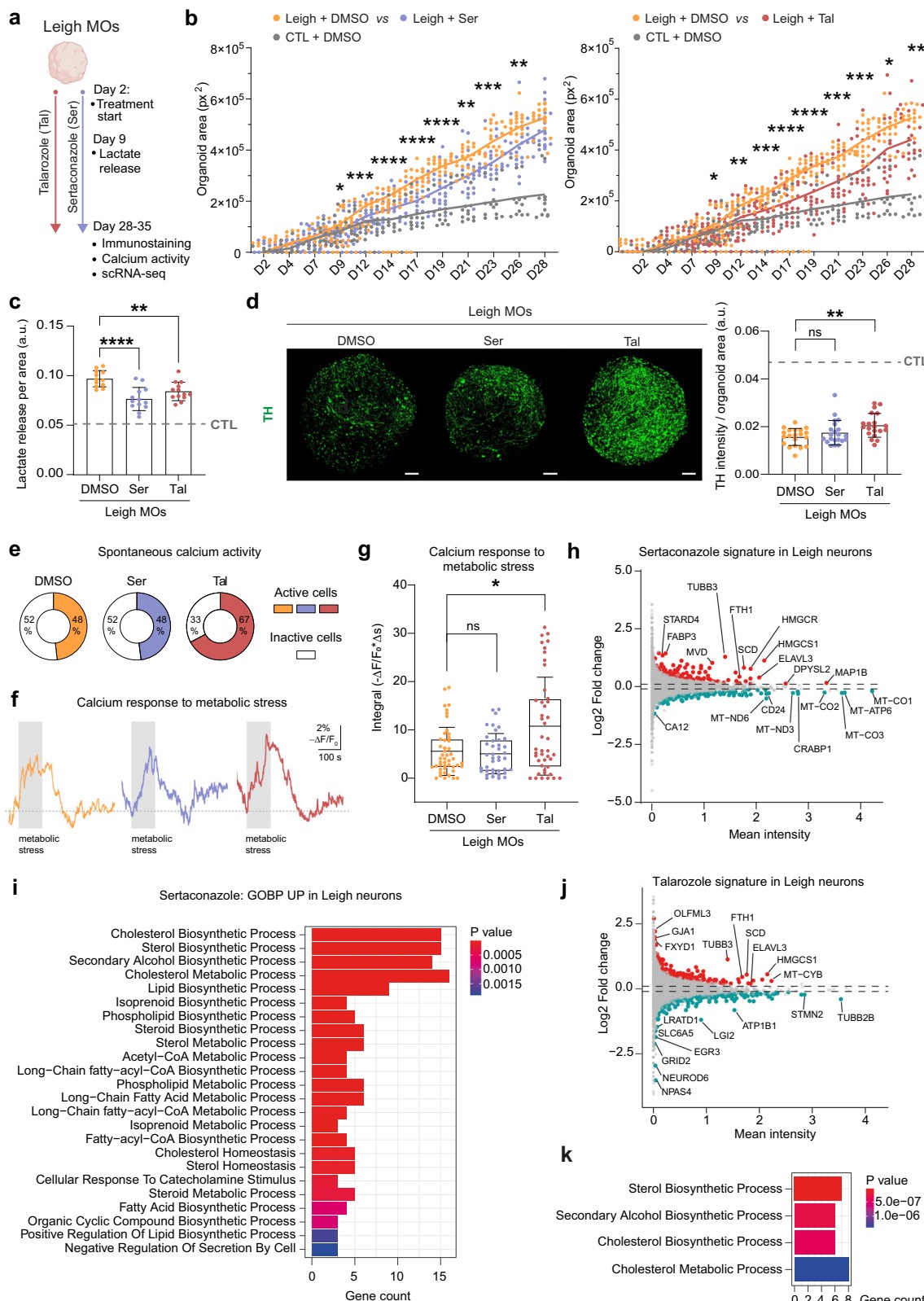

neural cells. Targeted metabolomics showed impaired AMP/ATP ratio by more than two-fold in Leigh NPCs compared to isogenic control NPCs (Fig. 5a, Supplementary Fig. 8a-b). The defective AMP/ATP ratio in Leigh NPCs was reversed by talarozole, while the recovery of AMP/ATP ratio by sertaconazole did not reach statistical significance (Fig. 5a, Supplementary Fig. 8a-b). Key glycolytic metabolites were defective in Leigh NPCs but were not rescued by the azoles

(Supplementary Fig. 8a-b). Nonetheless, both sertaconazole and talarozole normalized the level of tricarboxylic acid cycle (TCA) intermediate succinyl-CoA (Fig. 5b, Supplementary Fig. 8c-d), and talarozole also normalized the levels of TCA intermediate oxaloacetate (Fig. 5c, Supplementary Fig. 8c-d).

Lipidomics highlighted the impact of the azoles on membrane-associated lipids in Leigh NPCs (Fig. 5d-e, Supplementary Data 5).

**Fig. 4 | Sertaconazole and talarozole rescue key phenotypes in Leigh MOs.**
**a** Schematic of treatment in Leigh MOs (every other day starting from day 2) with 1 μM talarozole, 0.1 μM sertaconazole, or DMSO (created in BioRender. Menacho, C. (2026) https://BioRender.com/ewo91gu). **b** Growth rate of Leigh MOs treated with sertaconazole or talarozole compared to DMSO. Dots: individual MOs (DMSO: 20-25 MOs, Ser: 21-25 MOs, Tal: 21-25 MOs, CTL: 13-16 MOs); $n = 3$–4 independent experiments; *$p < 0.05$, **$p < 0.01$, ***$p < 0.005$, ****$p < 0.001$, two-way ANOVA. Growth rate of CTL MOs shown for reference. **c** Lactate release in treated Leigh MOs normalized to organoid area. Bar plots: mean +/- SD. Dots: individual MOs (DMSO: 12 MOs, Ser: 13 MOs, Tal: 13 MOs); $n = 3$ independent experiments; **$p < 0.01$, ****$p < 0.001$, one-way ANOVA, compound-treated Leigh MOs vs DMSO-treated Leigh MOs. Dotted gray line: average of 9 CTL MOs for reference. **d** Organization and quantification of TH-positive neurons in treated Leigh MOs. Bar plots: mean +/- SD of signal intensity normalized to organoid area. Dots: individual MOs (DMSO: 20 MOs, Ser: 21 MOs, Tal: 20 MOs); $n = 6$ independent experiments; *$p < 0.05$,

**$p < 0.01$, one-way ANOVA, compound-treated Leigh MOs vs DMSO-treated Leigh MOs. Scale bars: 100 μm. **e** Active cells within MOs: 27 out of 56 in 9 DMSO-treated Leigh MOs (48%), 39 out of 58 in 8 talarozole-treated Leigh MOs (67%), and 24 out of 50 in 8 sertaconazole-treated Leigh MOs (48%). **f** Representative calcium traces of Leigh MOs (average of 3 signals) upon metabolic stress (2 min of glucose-free medium + 2 mM 2-deoxyglucose + 5 mM sodium azide). **g** Calcium responses to metabolic stress. Boxplots: mean (center), 25/75 percentiles (box), and 1x SD (whiskers). Dots: individual cells within MOs (DMSO: 9 MOs; Ser: 7 MOs; Tal: 7 MOs); $n = 3$ independent experiments; *$p < 0.05$, two-tailed Mann-Whitney U test.
**h**–**k** Differentially expressed genes (DEG) and corresponding upregulated pathways in Leigh neurons based on pseudo-bulk analysis of scRNAseq of Leigh MOs treated with sertaconazole or talarozole. One-sided Fisher's exact test; Benjamini–Hochberg FDR-adjusted $P$ values. Source data are provided as a Source Data file.

---

Among this lipid category, glycerophospholipids phosphatidylethanolamine (PE) and phosphatidylglycerol (PG) were mainly modulated by sertaconazole (Fig. 5d, Supplementary Fig. 7e). Sphingolipids, and particularly hexosylceramides (HexCer), showed a 50% reduction in Leigh NPCs, which was reverted by sertaconazole and not by talarozole (Fig. 5e, Supplementary Fig. 7f-g). In agreement with the transcriptomics results (Fig. 4i, k), cholesterol levels were significantly modulated by both azoles (Fig. 5f).

We next performed additional experiments to further assess the impact of membrane-associated lipids and cholesterol, given their importance for neuronal morphogenesis[46,47] and their potential disruption in mitochondrial diseases[48]. The levels of neutral lipids quantified with BODIPY staining showed a significant increase in Leigh NPCs treated with sertaconazole for 24 or 48 h (Fig. 5g-h). Using the cholesterol-specific probe perfringolysin O (PFO), we visualized and quantified membrane-bound cholesterol. The levels of membrane-bound cholesterol were significantly reduced in Leigh NPCs compared to control NPCs (Fig. 5i-j). Treatment with sertaconazole or talarozole for 48 h significantly increased membrane-bound cholesterol in Leigh NPCs, with sertaconazole leading to a more prominent effect (Fig. 5i-j).

Taken together, the two azoles impacted mitochondrial energy metabolism and membrane-associated lipids in Leigh neural cells, validating the findings indicated by scRNAseq of MOs. In particular, talarozole was effective in restoring mitochondrial bioenergetics, whereas sertaconazole principally affected lipid metabolism.

## Sertaconazole and talarozole engage with RA and PPARγ pathways

To delve deeper into the mode of action of the two azoles, we analyzed their molecular structure to predict their pathway engagement.

We first employed the docking tool GOLD, which requires pre-defined binding regions for local docking[49]. We focused our search on cytochrome P450 26 (CYP26) enzymes, specifically CYP26A1 and CYP26B1, which represent the main protein targets of talarozole and belong to the retinoic acid (RA) pathway[50]. In fact, the transcriptomics of Leigh MOs showed downregulation of *CYP26B1* (Fig. 3f) and modulation of RA-associated genes by the azole compounds (Fig. 4h, Fig. 4j). Since the crystal structure of CYP26A1/B1 is not available, we performed the analysis on its homology model CYP120[51,52]. Sertaconazole and talarozole showed a bi-aromatic moiety close to the iron center of heme, with talarozole positioned in closer proximity to the center than sertaconazole (Fig. 6a). Thus, both azoles could bind CYP26A1/B1, although talarozole with higher affinity. Based on the structure of available azoles, we identified talarozole as the one with the highest binding affinity to the RA enzyme among the five azoles that we tested in iNs (DLD5, YSD1-4) (Fig. 2c-f, Supplementary Fig. 9a-b). In agreement with the effect of the azoles on the RA pathway in Leigh, Leigh mDANs exhibited upregulation of enzymes metabolizing RA (e.g. *CYP26A/B1*) and downregulation of RA receptors (e.g. *RORB*)

(Fig. 6b), and the treatment with sertaconazole or talarozole repressed the elevated expression of *CYP26B1* in Leigh iNs (Fig. 6c).

To complement these results, we applied the machine learning docking algorithm DiffDock[53], which uses generative artificial intelligence (AI) to search the whole protein without the need of defining a binding site, thereby allowing unbiased global docking information based on a learned confidence score. To ensure physical plausibility, we refined the top poses determined by DiffDock using the docking tool SMINA[54], which applies a physics-based scoring function. This hybrid approach represents an efficient and scalable solution, allowing for the exploration of the molecular interactions of the azoles more broadly to obtain an unbiased search with physically accurate refinement. Using this pipeline, we confirmed that sertaconazole and talarozole could bind to CYP26A1 and CYP26B1 (Fig. 6d-e). Talarozole showed a stronger binding capacity as seen by lower SMINA score and tighter distribution (Fig. 6e). In agreement with the GOLD docking analysis, talarozole appeared as the azole with the highest affinity for the RA enzyme (Supplementary Fig. 10a).

Modulation of proliferator-activated receptor gamma (PPARγ) has been suggested as a possible mechanism of action of talarozole in osteoarthritis[55]. Moreover, PPARγ was recently identified as a therapeutic target for Leigh based on findings obtained in a mouse model of Leigh caused by depletion of the complex I gene *Ndufs4*[56]. Therefore, we applied the same docking pipeline to assess the potential engagement of the azoles on PPARγ. We used available crystal structure of PPARγ[57] and investigated the effects of azoles on the human ligand-binding domain based on three poses (2FVJ: PPARγ bound to PA-082[58], 3B1M: PPARγ bound to Cerco-A[59], and 6L8B: PPARγ ligand-free[60]). Both azoles showed potential binding affinity, with sertaconazole exhibiting increased affinity for the 3B1M pose and talarozole for 2DVJ (Fig. 6f-g). Indeed, among all azole compounds, sertaconazole and talarozole were the ones showing the greatest binding capacity for PPARγ (Supplementary Fig. 10b).

Lastly, we aimed to determine whether sertaconazole or talarozole could modulate RA or PPARγ activity in live Leigh neural cells. We used luciferase reporter assays in NPCs treated with the azoles (Fig. 6h). Since CYP26A1 is the most inducible gene by RA, we transfected NPCs with a luciferase-reporter under the control of the CYP26A1 promoter[61]. Quantification of luciferase signal demonstrated that Leigh NPCs have reduced CYP26A1 promoter activity compared to isogenic control NPCs at basal level (with DMSO treatment only) (Fig. 6h). Treatment with talarozole in Leigh NPCs was sufficient to restore CYP26A1 promoter activity, while sertaconazole appeared ineffective (Fig. 6h). The combination of the azoles also did not lead to enhanced RA gene expression activation (Fig. 6h).

For assessing PPARγ activity, we transfected NPCs with a PPRE (peroxisome proliferator-activated receptor response element) luciferase reporter[62]. This tool captures the activity of PPARγ, which binds

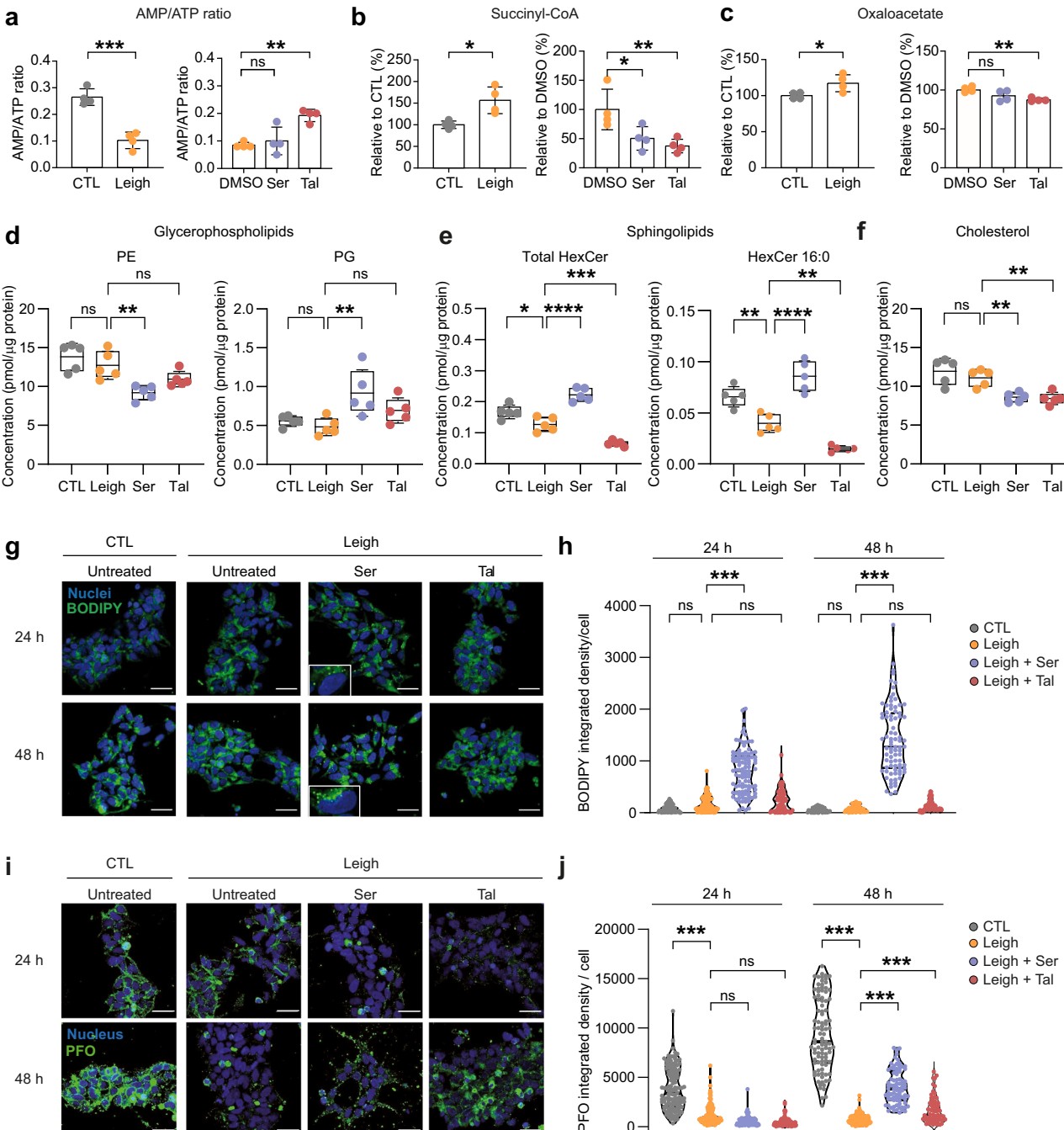

**Fig. 5 | Sertaconazole and talarozole modulate bioenergetics and membrane-bound lipids in Leigh neural cells. a–c** Targeted metabolomics in CTL NPCs and Leigh NPCs untreated, or treated with DMSO, 10 μM sertaconazole, or 10 μM talarozole for 24 h. Bar plots: mean +/- SD. Dots: biological replicates ($n = 4$) (dots); ns: not significant, *$p < 0.05$, **$p < 0.01$, ***$p < 0.005$; unpaired two-tailed t test for CTL vs Leigh and one-way ANOVA for treatments. **d–f** Targeted lipidomics in CTL NPCs and Leigh NPCs treated with DMSO, 10 μM sertaconazole, or 10 μM talarozole for 24 h, showing glycerophospholipids phosphatidylethanolamine (PE), phosphatidylglycerol (PG), hexosylceramides (HexCer), HexCer 16:0, and cholesterol. Boxplots: mean (center), 25/75 percentiles (box), and 1x SD (whiskers). Dots: biological replicates ($n = 5$); *$p < 0.05$, **$p < 0.01$, ***$p < 0.005$; one-way ANOVA.

**g, h** Neutral lipids stained with BODIPY in CTL NPCs and Leigh NPCs treated with DMSO, 10 μM sertaconazole, or 10 μM talarozole for 24 h or 48 h. Violin plots: median and quartiles of integrated density/cell. Dots: values of individual NPCs; $n = 3$ independent experiments; ns, ***$p < 0.005$; one-way ANOVA Kruskal-Wallis test. Scale bars: 100 μm. **i–j** Membrane-bound cholesterol measured with PFO staining in CTL NPCs and Leigh NPCs treated with DMSO, 10 μM sertaconazole, or 10 μM talarozole for 24 h or 48 h. Violin plots: median and quartiles of the integrated density/cell. Dots: values of individual NPCs; $n = 3$ independent experiments; ns, ***$p < 0.005$; one-way ANOVA Kruskal-Wallis test. Scale bars: 100 μm. Source data are provided as a Source Data file.

to PPRE leading to its gene activation, and was used in *Ndusf4* KO mice to uncover PPARγ as a therapeutic target of Leigh[56]. However, unlike the *Ndusf4* KO mouse, Leigh NPCs showed only *a* ~10% decrease in PPRE activity that did not reach statistical significance (Fig. 6h).

Nevertheless, talarozole treatment led to a significant 1.5-fold increase in PPARγ activity in Leigh NPCs (Fig. 6h). Sertaconazole treatment had no impact on PPARγ activity, and there was no combinatorial effect of sertaconazole and talarozole (Fig. 6h).

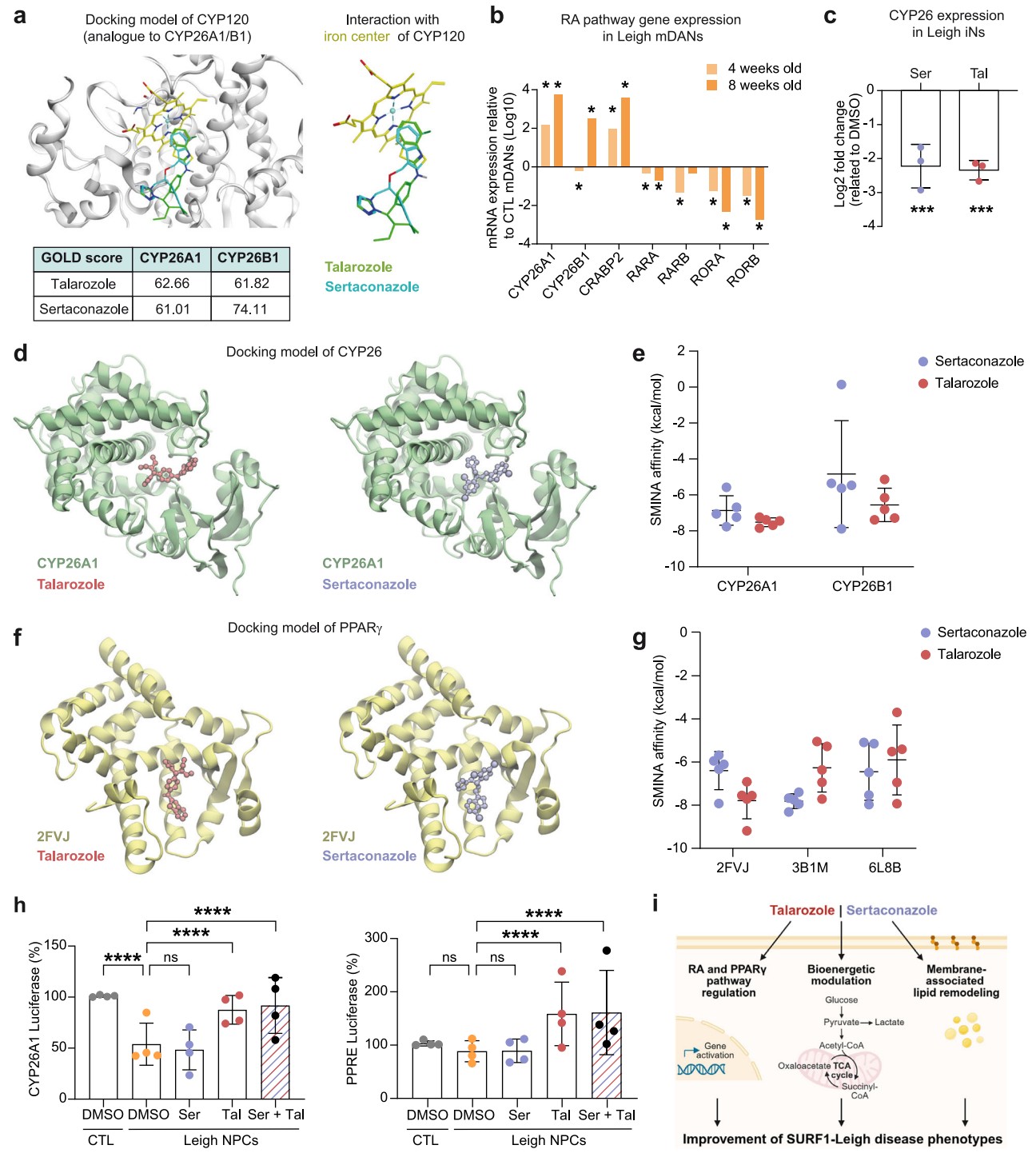

**Fig. 6 | Impact of sertaconazole and talarozole on RA and PPARγ in Leigh neural cells. a** Predicted docking for sertaconazole and talarozole on the CYP120 pocket based on GOLD analysis. **b** Expression of retinoic acid (RA)-related genes in Leigh mDANs compared to CTL mDANs. Two-tailed Wald test with Benjamini-Hochberg correction for multiple comparisons (FDR < 0.05); *p < 0.01, fold change expression from bulk RNAseq dataset. **c** CYP26 mRNA expression in Leigh iNs treated with sertaconazole or talarozole compared to DMSO-treated Leigh iNs. Bar plots: mean +/- SD. Dots: biological replicates (n = 3); ***p < 0.005; unpaired two-tailed t test. **d** Molecular representation of CYP26A1 (green) with the best ranked pose of talarozole and sertaconazole. Residues 116 to 134 (CYP2A1) were removed for visualization. **e** Docking results of talarozole and sertaconazole in CYP26A1 and CYP26B1 using SMINA affinity scores. Dots: mean +/- SD of the top 5 Diffdock poses, re-docked, and energy-minimized with SMINA. (**f**) Molecular representation of

PPARG 2FVJ (yellow) with the best ranked pose of talarozole and sertaconazole. Residues 260 to 305 (2FVJ) were removed for visualization. **g** Docking results of talarozole and sertaconazole in PPARG using SMINA affinity score. Dots: mean +/- SD of the top 5 Diffdock poses, re-docked, and energy-minimized with SMINA. **h** Luciferase activity in CTL NPCs and Leigh NPCs transfected with the reporter plasmid CYP26A1-Luc (left) or PPRE-Luc (right) and treated for 24 h with DMSO, 10 μM sertaconazole, 10 μM talarozole, or a combination of 5 μM sertaconazole and 5 μM talarozole. Bar plots: mean +/- SD. Dots: independent experiments (n = 4); data normalized to CTL + DMSO; ****p < 0.0001, two-way ANOVA. **i** Schematic of proposed mechanisms of action for sertaconazole and talarozole in Leigh neural cells (created in BioRender. Menacho, C. (2026) https://BioRender.com/t4fxyht). Source data are provided as a Source Data file.

In conclusion, talarozole appeared as the azole compound most effective in eliciting activity of the RA and PPARγ pathways. Sertaconazole might also be able to engage with these signaling pathways but with lower effectiveness. Other azoles showed even lower binding capacity for RA and PPARγ enzymes, which may potentially explain why they were not found capable of improving neuronal morphogenesis in Leigh iNs.

## Discussion

Regulators are currently promoting drug discovery approaches based on non-animal models such as NAMs to possibly reduce or eliminate the need for validations in animals for drug approval[63,64]. In this context, complex human models such as 3D organoids are emerging as promising alternatives[65]. Nonetheless, despite the recent progress of NAMs in drug validation and toxicity, performing large-scale drug screening in iPSC-derived brain organoids remains challenging. In fact, most of the treatments assessed in iPSC models of Leigh and other mitochondrial diseases have been compounds already known from previous studies[14,66,67]. We recently screened repurposable compounds in iPSC-derived neural cells, leading to the identification of sildenafil as a potential treatment for Leigh caused by *MT-ATP6* variants[29]. However, innovative pipelines are needed to accelerate the screening process by reducing the number of potential drugs to be assessed in complex 3D human NAMs. In this work, we demonstrate that combining disease-specific brain organoids with computational drug discovery can lead to the identification of potential repurposable drugs for Leigh syndrome, an incurable neurological mitochondrial disease with highly unmet medical needs.

There is a growing interest in developing in silico approaches[68] to accelerate the drug discovery process, which is costly, time-consuming, and shows a high attrition rate[69]. Currently, widely used databases for predicting drugs, such as CMap[70], are, however, mainly comprised of transcriptional signatures of cancer cell lines. This feature may hinder the effectiveness of computational predictions for non-cancer diseases. Cancer cell lines exhibit particular signal transduction pathways and gene regulatory networks and unique metabolic features mainly relying on glycolysis[71,72]. In the context of neuronal diseases, these metabolic differences may play a relevant role, given that proper neuronal generation is accompanied by a switch from glycolysis to OXPHOS[73] and interventions based on modulation of mitochondrial metabolism may help tackle neurodegeneration[74]. Here, we developed a DL-based framework based on ChemPert, our recently developed database of perturbations in non-cancer cells[31]. We demonstrated that this approach delivered added value beyond conventional pathway enrichment by enabling more accurate early prioritization of candidate drugs for experimental validation. Since the feedforward artificial neural network (FNN) model was trained on non-cancer cells, we circumvented the issue of possible false positives due to predictions driven by cancer cells.

We applied this pipeline to a scRNAseq dataset that we previously obtained in Leigh COs carrying *SURF1* variants and isogenic control COs[40]. Our framework predicted first genes and then drugs in a cell-type specific manner, thereby allowing us to identify also drugs that were outside of training (as was the case for talarozole). In parallel, we conducted a survival screen in yeast deficient for *SHY1*, the yeast homologous gene of *SURF1*[6]. Yeast screens have been used before in the context of neurological and mitochondrial diseases[75,76], but their predictive power requires validation in more complex models[77]. By combining computational predictions and yeast screen with our human 2D and 3D models of Leigh, we were able to uncover sertaconazole and talarozole as repurposable drugs for Leigh (Supplementary Fig. 11). We also used healthy human brain organoids as NAMs to ensure that two compounds did not elicit toxic effects at the concentrations used to rescue the disease phenotypes (Supplementary Fig. 6a, b) and concluded that the compounds could ameliorate Leigh phenotypes at concentrations that appeared to be non-toxic in healthy cells.

The two azoles shared similarities in chemical structure that might enable them to interact similarly to specific targets, although with different effectiveness (Fig. 6i). Talarozole is a retinoic acid metabolism blocking agent (RAMBA) that inhibits cytochrome P450 CYP26A1/B1 thereby activating the RA pathway[78]. RA signaling plays a key role in regulating differentiation, organogenesis, and neuronal development[79–84]. Using two complementary docking strategies, we confirmed that talarozole was the azole with the highest binding affinity for CYP26A1/B1 among the ones we assessed (Supplementary Fig. 9b, Supplementary Fig. 10a). Accordingly, talarozole was more effective than sertaconazole in activating the RA pathway based on CYP26A1 promoter activation (Fig. 6h). Possibly because of this impact on RA, talarozole showed a stronger rescue than sertaconazole with respect to key disease phenotypes including neuronal generation and functionality in Leigh MOs (Fig. 4d-e), mitochondrial bioenergetics in Leigh NPCs (Fig. 5a-c), and survival in *SHY1* KO yeast (Fig. 2g). Talarozole also enhanced the calcium-mediated response of MOs to acute metabolic stress (Fig. 4f-g). This feature may potentially be of clinical value to Leigh individuals, who typically suffer from sudden metabolic decompensation triggered by events that place excessive demands on the body's energy production, such as infection or surgery[1,43].

Talarozole was initially developed for acne and psoriasis[85]. Although its use in these topical indications has been discontinued, talarozole is now being considered as a potential repurposed drug for osteoarthritis, where it showed benefits in vitro and in vivo related to a reduction of inflammation[86,87]. The anti-inflammatory effect of talarozole has been linked to a PPARγ-dependent mechanism[55]. Interestingly, PPARγ may represent a therapeutic target for Leigh, given that its modulation by cannabidiol (CBD) treatment extended the lifespan and improved disease comorbidities in a mouse model of Leigh due to KO of the Complex I gene *Ndufs4*[56]. Modulation of the PPARγ coactivator-1α (PGC-1α) have also been suggested as Leigh therapeutics[15,88]. Our docking results suggest that talarozole is indeed capable of binding to PPARγ, with higher affinity compared to the other azoles assessed (Supplementary Fig. 10b). Accordingly, talarozole treatment elicited PPARγ activity in Leigh NPCs (Fig. 6h). Therefore, some of the talarozole's beneficial effects seen in our Leigh models may be mediated trough PPARγ activation. At the same time, since our COs and MOs did not contain microglia and included only low numbers of astrocytes at these early stages of development, they may not represent the most suitable models to investigate inflammatory-related aspects. Future studies with more appropriate models should thus be performed to assess the in vivo impact of talarozole and further validate RA or PPARγ as putative interventional targets for Leigh.

Sertaconazole is currently used in topic anti-fungal medications, as it inhibits the synthesis of ergosterol, which is the orthologue of cholesterol and an essential component of fungal cell walls[89]. In agreement with this mode of action, we found that sertaconazole treatment in Leigh models led to modulation of membrane-bound lipids and cholesterol (Figs. 4h-i, 5d-j). Possibly due to structural similarities, talarozole treatment in Leigh models also affected lipid metabolism, although less effectively than sertaconazole (Figs. 4j-k, 5d-j). At the same time, sertaconazole rescued key Leigh phenotypes similarly to talarozole in Leigh MOs, including growth rate defects and abnormal lactate release (Fig. 4b-c). Modulation of lipid metabolism might have contributed to those rescue effects. Indeed, membrane-associated lipids and cholesterol play crucial roles in neuronal branching and morphogenesis[46,47] and their disruption is associated with mitochondrial diseases[48]. Future studies should therefore address the impact of modulating membrane-bound lipids as potential interventions for Leigh and other mitochondrial disorders. Sertaconazole also engaged with the RA and PPARγ pathways but with less effectiveness than talarozole (Fig. 6a-h, Supplementary Fig. 9a-b,

Supplementary Fig. 10a-b). The combinatorial treatment did not elicit an increased activity of those pathways. Further investigations are warranted to explore the potential synergistic effects of these drugs in Leigh or other conditions. In any case, the compounds likely exhibit polypharmacology and their action could be even more multifactorial than the ones suggested by our findings. Giving the rescue effect seen in yeast (Fig. 2g-h), it may also be possible to speculate that the azoles could act as COX stabilizers.

Our study also highlights how innovative complex NAMs based on computational screens in patient-specific brain organoids, can be combined with conventional screens in yeast, which are relatively rapid and cost-effective, to further accelerate drug discovery. This pipeline could represent a blueprint for rare neurodevelopmental disorders. Talarozole, our most effective candidate, had one of the highest BES values (Supplementary Fig. 1h-i), yet it was not recovered by GSEA-style enrichment, CMap-based signature reversal, or ChemPert-based similarity[31,70,90], suggesting that talarozole's prioritization by our DL framework was method-specific and not trivially recapitulated by existing approaches. At the same time, talarozole currently represents a single prospectively validated hit and should therefore be viewed as proof-of-concept example, with further testing of additional top-ranked candidates needed to generalize the framework's predictive performance. The clinical relevance of the drugs that we identified is underscored by the fact that other drugs belonging to the azole family are under consideration for neurological diseases[91,92]. In the context of Leigh, further studies are warranted to better understand the impact of these azoles on different brain regions besides the midbrain, for example, by using other region-specific brain organoids or assembloids generated by combining different region-specific organoids, possibly with included microglia[93]. Additional work is also necessary to evaluate whether the azoles might have positive effects in the context of other variants linked to Leigh, such as those in genes *MT-ATP6* or *NDUFS4*.

Lastly, our findings support a view of Leigh pathogenesis that involves not only early-onset neurodegeneration but also impaired neurodevelopment. Indeed, children affected by Leigh typically exhibit delays in reaching developmental milestones even prior to the onset of regressive phenotypes[26-28]. Antenatal manifestations are also observed in patients with Leigh or other mitochondrial diseases[23,24]. The underlying defects might possibly be related to the impact of mitochondria in neurodevelopment[18,21], potentially indicating that defective mitochondrial function interferes with this physiological neurogenesis[20,94]. Accordingly, defective embryonic development occurs in Leigh *Ndufs4* KO mice[22], which show specific defects in neural stem and progenitor cell proliferation, causing significant delays in neurogenesis and gliogenesis[25]. Brain organoid models of Leigh demonstrated a similar impairment of progenitor organization and defective corticogenesis[15,17]. Taken together, our data suggest that addressing neurodevelopmental aspects of mitochondrial diseases may lead to uncover innovative therapeutics for currently incurable diseases such as Leigh syndrome.

## Methods

### Ethics
iPSC lines were used in accordance with the ethical approval obtained by the Ethics Committee of the Medical Faculty of Heinrich Heine University (study number 2020-967_5).

### Induced pluripotent stem cells (iPSCs)
Leigh iPSCs and isogenic control iPSCs were obtained before using Leigh patient-derived cells and CRISPR/Cas9 editing[15]. Leigh iPSCs carried the *SURF1* variant c.769 G > A (p.G257R) on both alleles in either a Leigh patient background or a healthy control background. The healthy iPSC line was obtained from Dr. Heiko Lickert (Helmholtz Center Munich) (HMGUi001-A)[95]. Isogenic control iPSCs carried no

*SURF1* variants in either the patient background or the control background (HMGUi001-A-64)[15].

### Neural progenitor cells (NPCs)
iPSC-derived Leigh NPCs and isogenic control NPCs used in this study were generated and characterized before[15]. All NPCs were cultured as described[96], with minor modifications. NPCs were grown in Matrigel-coated 6-well plates in sm+ medium, consisting of a base media 1:1 ratio of Neurobasal and DMEM-F12 supplemented with N2 (0.5×), B27 without vitamin A (0.5×), 2 mM glutamine, Myco-Zap-plus-CL (1×), 3 μM CHIR 99021, 0.5 μM purmorphamine, and 150 μM ascorbic acid. Media were exchanged every other day, and cells were passaged weekly at a ratio 1:6 to 1:10 using Accutase treatment. All NPCs were used in experiments between passages 10 and 25. All cells were routinely monitored to ensure lack of mycoplasma contamination.

### Development of DL-based framework for drug repurposing
A multi-stage deep learning framework for predicting direct protein targets of drugs was developed using perturbation transcriptomics data (Supplementary Fig. 1b). Training data were retrieved from ChemPert[31], which provides transcriptomic profiles before and after drug treatment based on non-cancer cell types. We used 24,134 transcriptional signatures of 264 unique, diverse non-cancer cell types exposed to 2905 unique perturbagens (chemical and biological disruptive agents) for the model training (Supplementary Data 7). All drugs considered in this study, along with their known direct protein targets, were also retrieved from ChemPert; only drugs with annotated direct molecular targets were included, resulting in 5,686 total reference drugs. Signaling proteins and their molecular interaction data were compiled from multiple curated databases, including protein–protein interactions from iRefIndex[97], HIPPI[98], IID[99], signaling networks from OmniPath[100], ReactomeFI[101], KEGG[102], NicheNet[103], transcriptional regulatory interactions from TRRUST[104], CHEA[105], HOCOMOCO[106], TRANSFAC[107], as well as ligand–receptor interactions from a consolidated set of sources[103,108-119]. TFs were obtained from AnimalTFDB 3.0[119]. The resulting graph consisted of 20,455 protein nodes and 5,928,682 edges. The input features for model training included (1) differentially expressed TFs (DETFs) derived from the difference between initial and final transcriptomic states following perturbation, and (2) baseline gene expression levels of the initial cell state. Each of the 20,455 proteins was embedded in a network-aware feature space using precomputed graph metrics and local network topologies (Step 1). Consequently, each transcriptomic input yielded 20,455 independent training samples, one per protein, with the goal of classifying the optimal perturbation for that protein. These structured features were then denoised and regularized using a VAE (variational autoencoder), and the decoded output, preserving the same dimensionality as the original input, was used as input to a feedforward neural network (FNN) trained to predict direct protein targets of drugs (Step 3). Final drug rankings were generated based on predicted target enrichment using a Bayesian enrichment score (BSE) that incorporated hypergeometric overlap and empirical model robustness (Step 4).

**Step 1: Graph-based feature embeddings.** To capture topological and signaling context around each gene while avoiding memory-intensive full graph convolution, we used the Scalable Inception Graph Neural Network (SIGN) framework[120]. Each node (protein) in the graph was assigned two initial features: its raw expression value in the baseline (pre-treatment) cell state, and a DETF status label. DETF status was defined as follows: genes identified as differentially upregulated TFs were labeled as 1, downregulated TFs as −1, and all other genes as 0. When the number of DETFs exceeded 100 for a given perturbation, a maximum of 100 DETFs were randomly selected to serve as source nodes for propagation. This step was necessary to avoid oversaturation of the graph during message propagation, wherein too

many active source nodes diminish the ability to differentiate downstream nodes based on their signaling proximity. Although alternative strategies such as fold-change-based ranking were considered, fold change alone is not always a reliable indicator of regulatory importance and may overlook TFs with moderate expression changes but high centrality or signaling relevance. Moreover, fold-change thresholds can introduce bias toward cell type- or dataset-specific noise. Random selection avoids these pitfalls by preserving unbiased sampling across the DETF pool and improves overall generalizability across diverse perturbation contexts. Using this DETF labeling as node input, we applied SIGN-based message passing in both upstream and downstream directions through the network to compute graph convolution embeddings:

$$Z = Concat\left(X, \{A_{fwd}^k X\}_{k=1}^K, \{A_{rev}^k X\}_{k=1}^K\right), for\ k = 1, 2, \ldots, K \quad (1)$$

where $X \in \mathbb{R}^{nxd}$ is the input feature matrix where n is the number of proteins (nodes) and d is the number of input features per protein (e.g. expression + DETF label), $A \in \mathbb{R}^{nxn}$ is the adjacency matrix of the directed signaling graph (normalized), and K is the maximum number of message passing steps, which was set to 5 in this study. The propagated signals captured the multi-hop influence of DETFs on surrounding proteins and vice versa, generating a topological embedding for each protein that reflected its context in the perturbation-responsive network. The following statistics were also computed for each protein across the propagated feature dimensions: sum, mean, maximum, minimum, median, standard deviation, 25th, and 75th percentiles. To further enrich these embeddings with biologically relevant context, we appended the following network-derived features: the number of shared Reactome pathways between each protein and DETF, the minimum path length from each protein to all DETFs, and the percentage of DETFs reachable from each protein within the graph. Then, a second-degree polynomial expansion was applied using the PolynomialFeatures function from scikit-learn (degree=2, interaction_only=False). Following this expansion, any features that exhibited constant values across all samples were removed. This process produced a 20,455 by 1870 feature matrix per perturbation transcriptomics dataset. In preparation for supervised training, the drug' effects on their direct protein targets were labeled as follows: 1 for activation, −1 for inhibition, and 2 for unknown or ambiguous effects (either 1 or −1), and 0 for no effect. To reduce computational complexity without compromising representativeness, we applied MiniBatch KMeans clustering[121] with 50 clusters and sampled an equal number of examples from each cluster to construct a diverse and representative subsets. The resulting expanded feature matrix was then column-wise min-max normalized (i.e. normalized across samples (proteins) for each feature) by:

$$z_j = \frac{x_j - \min(x)}{\max(x) - \min(x)} \quad (2)$$

to ensure scale compatibility for downstream modeling, where x is each feature vector across all genes.

**Step 2: Denoising and regularization with variational autoencoder (VAE).** To enable robust training and mitigate variability introduced by heterogeneous transcriptomic data sources, including differences in experimental protocols, platforms (microarray, bulk RNA-seq, and scRNA-seq), and sample origins, we trained a VAE on the graph-based protein feature matrix. The VAE served to denoise and regularize the high-dimensional input space, producing a smoother and more consistent latent representation. The encoder consisted of five fully connected layers with progressively smaller sizes, starting from an initial hidden size of 1500 and halving at each layer down to a 100-dimensional latent space. The encoder used batch normalization

and ReLU activations, without dropout. The decoder mirrored the encoder but omitted batch normalization to preserve the original data distribution during reconstruction. The latent space was parameterized by a mean and log-variance vector, and the reparameterization trick was applied to sample latent representations. Log-variance was clamped between −5 and 5 to avoid instability. To assess the quality of learned representations and guide hyperparameter selection, clustering performance was evaluated on both the VAE latent space and the decoded output space. Specifically, Mini-Batch KMeans clustering (with k = 3 clusters, corresponding to known target effect classes: 0 = no effect, 1 = activation, and -1 = inhibition. 2 = unknown effect was excluded from this analysis.) was applied to each representation. Optimal hyperparameters were selected based on the highest adjusted Rand index (ARI) and normalized mutual information (NMI) scores achieved across candidate configurations. A 1-fold validation strategy was used, with 10% of gene rows randomly held out as the validation set. Given the size and homogeneity of the resulting training pool, 5-fold cross-validation was not necessary and would provide limited additional benefit at significantly higher computational cost. The ARI and NMI for clustering in the VAE-decoded space were 0.2465 and 0.3457, respectively, nearly identical to those obtained from the raw input features (ARI = 0.2464, NMI = 0.3458). In contrast, clustering within the latent space yielded substantially lower ARI (0.1144) and NMI (0.2164). Although ARI and NMI do not directly assess training performance, these results indicate that the VAE preserved biologically meaningful structure while denoising and regularizing the input space. The absence of artificially inflated clustering performance suggests that the VAE did not introduce synthetic features or over-regularization, which could lead to overfitting or loss of generalizability to unseen samples. Based on these results, we used the VAE-decoded feature space for all downstream classification tasks. The final set of parameters was batch size of 1024, learning rate of 0.001, and $\beta = 0.001$ to weight the KL divergence term in the loss function.

**Step 3: FNNs for drug target prediction.** To predict direct protein targets of drugs from the VAE-decoded graph embedding features, we trained an FNN model with a decaying architecture implemented in PyTorch Lightning. The model consists of fully connected layers whose widths decrease exponentially by a decay factor from a user-defined initial hidden-layer size. This design helps compress and refine the high-dimensional input features into a more discriminative representation for classification. Each hidden layer includes batch normalization, LeakyReLU activation, and dropout, and the final layer outputs logits for three classes: 1 = activation, −1 = inhibition, and 0 = no effect, using integer-encoded labels (1, 2, and 0, respectively). Class 2 (ambiguous effect, either 1 or −1) was excluded from training due to its interpretational uncertainty to ensure that the model was trained only on high-confidence labels, enhancing prediction reliability. To reduce potential sources of training bias, the sample count for each class was adjusted to the mean of the original class sizes (451,476) via downsampling and synthetic data generation using the VAE decoder from original protein feature inputs. Hyperparameters, including batch size, learning rate, dropout, initial layer size, number of layers, L1/L2 regularization, and weight decay, were tuned using a grid search based on performance on a randomly held-out 10% validation set. Given the large number of well-regularized training samples and the computational cost of k-fold cross-validation multiplied by the hyperparameter combinations, we used a 1-fold validation strategy. This approach provided stable and representative performance estimates, with marginal gain expected from further splits. Model performance was evaluated using the F1 score, false positive rate (FPR), and false negative rate (FNR). Following the optimization of hyperparameters, 21 best-performing models were selected, and ensemble model evaluation was conducted by testing all triplets of the 21 models to assess the robustness and potential performance gains from model aggregation.

In those ensemble models, final predictions were made via majority voting. To avoid ties, only individual models and triplets were included; pairs and four-model ensembles were excluded. The final ensemble model achieved a mean F1 score, FPR, and FNR of 0.700, 0.129, 0.305, respectively on the validation set (Supplementary Fig. 1c-e), and this model was used for subsequent applications. The final three model configurations are each using a learning rate of 0.005, a dropout rate of 0.0, an initial hidden layer size of 1500, and five hidden layers. The first configuration used a batch size of 1024 with no L1 or L2 regularization, the second configuration increased the batch size to 2048 and L1 and L2 regularization values of 0.2 and 0.1, respectively, and the third configuration also used a batch size of 2048 but did not include any regularization.

**Step 4: Ranking drugs by Bayesian enrichment score (BES).** The framework computes a one-sided hypergeometric p-value that quantifies whether the predicted targets are statistically enriched for each drug's known targets, i.e.,

$$p_{hypergeo} = \sum_{i=k}^{\min(n,N)} \frac{\binom{n}{i}\binom{M-n}{N-i}}{\binom{M}{N}} \tag{3}$$

where $M$ is the total number of proteins (i.e. the size of the network), $n$ is the number of known targets of the drug, $N$ is the number of proteins predicted as targets, and $k$ is the number of overlapping proteins between known and predicted targets. Next, to assess the robustness of drug predictions and control for potential false positives due to noise or uninformative features in input, we computed an empirical p-value (pemp) for each drug using a randomized null model. Specifically, the decoded input feature matrix, where each row corresponds to the feature vector of a protein, was randomly shuffled across rows, while preserving protein identities. Then, the FNN model was applied. The drug was considered "predicted" if at least one of its target proteins was among the predicted targets. This process was repeated 100 times and the pemp for that drug was defined as the fraction of the 100 randomized trials in which it was predicted. Drugs with low pemp values are thus considered robust predictions unlikely to occur under a random input distribution. We incorporated this empirical confidence into a posterior weight under a Bayesian prior belief that only 1% of the considered drugs are true positives, i.e. able to induce desired gene expression changes:

$$\pi_0 = 0.99 \Rightarrow posterior_i = 1 - \pi_0 \cdot pemp_i \tag{4}$$

where $i$ denotes each drug. This posterior is then modulated to form a confidence weighting factor (CWF) by a soft exponential confidence weighting function, favouring drugs with highly confident predictions:

$$confidence\,weighting\,factor\,(CWF)_i = \exp\left[-\lambda \cdot (1 - pemp_i)\right] \tag{5}$$

with $-\lambda = 1.0$ by default. The Bayesian-weighted confidence score becomes:

$$confidence\,weighting\,factor\,(CWF)_i = \exp\left[-\lambda \cdot (1 - pemp_i)\right] \tag{6}$$

This sigmoid-like transformation ensures that enrichment near the top-ranked percentile is prioritized, reflecting the practical goal of selecting a small number of high-confidence candidates for downstream validation. The final BES for each drug is computed as a weighted sum of posterior confidence, statistical enrichment $(1 - p_{hypergeo})$, and the normalized rank of the p-value:

$$BES_i = \omega_{posterior} \cdot s_{posterior,i} + \omega_{hypergeo} \cdot \left(1 - p_{hypergeo,i}\right) + \omega_{hypergeo_{rank}} \cdot r_{hypergeo,i} \tag{7}$$

where $r_{hypergeo,i}[0 \in 1]$ is the normalized rank (1 = best), $\omega_{posterior}, \omega_{hypergeo}, \omega_{hypergeo\,rank}$ are tunable scalar weights. In our implementation, we used $\omega_{posterior} = 20, \omega_{hypergeo} = 1.0, \omega_{hypergeo\,rank} = 1.0$, thereby placing greater emphasis on high-confidence model predictions while also rewarding statistical and rank-based enrichment. Drugs with hypergeometric p-value = 1.0 (i.e. no single drug target predicted) were automatically assigned a BES of 0. All non-zero BES values were then ranked in descending order, with ties resolved using the min method.

## DL-based framework test performance and selection

To evaluate the generalizability of the DL-based framework, we assembled 47 independent test datasets centered on chemical drugs that induce cellular conversion processes, including differentiation, development, metabolic shift, reprogramming, and immune activation (Supplementary Data 1). Although some of the original datasets also included protein ligands or receptor-based agents, only small-molecule chemical drugs were considered in this evaluation, as the primary aim was to assess the tool's translational potential in small molecule drug discovery/repurposing. All gene expression datasets used in the test set were completely excluded from the training and validation phases, ensuring an unbiased performance assessment. To broaden the test set's scope, we allowed chemical perturbation knowledge (i.e. drugs known to promote specific cell-state transitions) and transcriptomic readouts to originate from different but mechanistically aligned studies. This design increased the diversity of test conditions while maintaining relevance between gene expression signatures and the biological actions of candidate drugs. During evaluation, we further filtered candidate drugs to remove those known to act on apoptotic pathways, as such agents are primarily relevant in cancer contexts and are not considered suitable for our cell conversion cases. Across the test set, the model achieved an F1 score, FPR, and FNR of 0.528, 0.167, and 0.472, respectively, in identifying drug target proteins (Supplementary Fig. 1c-e). Next, using the BES, we assessed whether the predicted protein targets overlapped significantly with known targets of each drug. Given the lack of well-defined positive and negative drug sets, we focused on early recall of true positives within top percentiles, a clinically relevant metric for downstream experimental validation. The DL-based framework achieved a 9.2% average recall of true positive hits within the top 1st percentile of ranked drugs (Supplementary Fig. 1f), outperforming gene set enrichment analysis (GSEA)-based methods, where the best-performing case reached only 2.1%, and random ranking yielded 1%, as expected. At the 2nd percentile cutoff, the DL-based framework achieved 16.3% recall, compared to 6.1% for the best GSEA result. This measure emphasizes the framework's ability to prioritize strong candidates from large pools. In the GSEA-based baseline, enriched pathways were identified from the input DETFs using Gene Ontology Biological Process (GOBP), Reactome, and KEGG via enrichR[122], applying adjusted p-value cutoffs of 0.05, 0.001, and 0.005 to each database. Differentially expressed genes (DEGs) within the enriched pathways were then used as predicted target proteins, which were subsequently used to rank drugs based on a BES. These results demonstrate that the DL-based framework offers improved early enrichment performance over GSEA-based methods, enabling more effective identification of candidate drugs.

To prioritize drug candidates from the prediction set, we selected drugs ranked within the top 2.5th percentile of the BES based on the test recall curve, which plateaued at 2.5% (Supplementary Fig. 1f-g). After

removing duplicates, this cutoff resulted in 140 drugs for DETF1 and 143 drugs for DETF2. We then prioritized the hits based on their postulated primary target or mode of action and selected those with known modulation of metabolism according to Chembl (https://www.ebi.ac.uk) and Drug Bank (https://go.drugbank.com/). This strategy resulted in a total of 34 DL-predicted drugs (DLDs) (20 for DETF1 and 14 for DETF2). Among these, we selected 5 based on their safety profile in humans (DLD1, DLD3, DLD4, DLD5 from DETF1, and DLD2 from DETF2).

### Generation of midbrain organoids (MOs)

MOs were generated following a published protocol[37], with some modifications. To start the generation of MOs, NPCs were detached using Accutase to obtain a single cell suspension. NPCs were then counted using an automatic cell counter, with a gating size of 5–19 µm, and seeded onto low-attachment U-bottom 96-well plates at a density of 9000 cells/150 µl media per well. Seeding medium was composed of a basal medium containing 1:1 ratio DMEM-F12:Neurobasal, N2 (0.25×), B27 without vitamin A (0.25×), glutamax (1×), MycoZap-Plus-CL (1×), 0.4% (w/v) polyvinyl-alcohol (PVA) supplemented with the following small molecules: 0.5 µM Purmorphamine, 3 µM CHIR 99021 and 100 µM ascorbic acid. After 2 days, seeding medium was replaced with a basal medium supplemented with 100 µM ascorbic acid, 1 µM Purmorphamine, 1 ng/ml BDNF, and 1 ng/ml GDNF to induce ventral patterning. Four days later, the medium was exchanged to a maturation medium, composed of the basal medium supplemented with 100 µM ascorbic acid, 2 ng/ml BDNF, 2 ng/ml GDNF, 1 ng/ml TGF-β3, and 100 µM dibutyryl-cAMP. On day 6, a single concentration of 5 ng/ml activin A was added. From day 7 on, maturation medium (without activin A) was refreshed 3 times a week. For experiments with chronic treatment, MOs were treated from day 2 until the day of the assay (either until day 9 or day 28–35) with either 0.1 µM sertaconazole, 1 µM talarozole, or DMSO (vehicle). DMSO samples contained 0.1% (v/v) DMSO, which was equivalent to the amount of solvent in the samples that received compound treatment. Treatments were refreshed upon every media change every other day.

### MO growth rate analysis, viability assay, and lactate release

To measure organoid size, pictures were taken every other day with a Nikon Eclipse TS2 inverted microscope using a 4× objective and ensuring good contrast between the background and the organoid. Organoid area and perimeter were measured using CellProfiler (v. 4.2.5). Color images were first converted into grayscale images using the modules "Color to gray" and "Invert for printing". Next, grayscale images were used as an input for the module "Identify Primary Objects" to identify a bright object (organoid) of a diameter ranging 300–1200 px on a dark background. The properties of identified objects could then be measured using the module "Measure Object Size Shape". Undesired identified objects (such as debris) were filtered out using the "Filter Objects" module, based on their compactness, with a maximum value of 4. After filtering, the module "Measure Object Size Shape" was again used to measure the properties of the filtered identified objects (organoid). The results of this module were exported to Excel files for analysis. The organoid area measured at day 2 was used as a reference starting point and was therefore subtracted from each individual value obtained. thereafter. At least 5 organoids per cell line for each time point from 3 independent experiments were analyzed. For the viability assay, control MOs were treated from day 2 to day 11 every second day with different concentrations (0.1 µM, 1 µM, 5 µM, 10 µM) of talarozole or sertaconazole. DMSO control samples contained 0.1% (v/v) DMSO, which is equivalent to the amount of solvent in the samples that received compound treatment. On day 9, organoid viability was measured using the CellTiter-Glo cell viability assay following a previously published protocol[37]. Briefly, the medium of each well was adjusted to 55 µl, and the plates with MOs together with the kit

´s reagents were brought to room temperature (RT) for 30 min. Next, an equal volume of 55 µl of CellTiter-Glo reagent was added, and samples were shaken on a Thermomixer at 900 rpm for 5 min. Afterwards, plates were incubated for 25 min at RT protected from light. The contents of the plate were transferred to white/opaque 96-well plates with 2 technical replicates (20 µl each) per sample. Luminescence of each sample was recorded using a Tecan plate reader. Extracellular lactate released by MOs in the media was measured by collecting 100 µl of maturation medium at day 9 during MOs generation. Media from at least 5 individual MOs from 3 different batches were collected for analysis and stored at −20 °C until used. The manufacturer's instructions from the lactate assay commercial kit were followed for measuring the absorbance of each sample using a clear 96-well microplate assay and Tecan microplate to read the absorbance at 580 nm. The calculated amount of lactate per well was normalized to the respective organoid size and expressed as fold-change compared to control (when untreated) or to DMSO-treated sample (when treated with sertaconazole or talarozole) of their respective batch.

### Single-cell RNA sequencing (scRNAseq) of untreated MOs

35 days-old MOs from control iPSCs were dissociated for scRNAseq using the papain dissociation system (Worthington). Papain was prepared by adding 5 ml of EBSS to one papain vial and incubated for 10 min at 37 °C. Additionally, 500 µl of EBSS was added to one DNAse vial and gently mixed. Half of the volume of DNAse in EBSS was then added to the papain vial, resulting in a final concentration of approximately 20 units/ml of papain and 0.005% DNase. About 48 individual MOs were collected onto a 6 well-plate, washed with PBS, and incubated with 2 ml of pre-warmed papain/DNase solution for 5–10 min at 37 °C in an orbital shaker. Using a 1 ml pipette, the organoids were gently triturated by mixing up and down and placed back into the incubator for another 5–10 min for further enzymatic digestion. The digested organoids were then collected in a 15 ml tube, and 5 ml of ovalbumin was added. Using a 225 mm polished glass pipette, the mixture was further triturated until mostly dissociated. The cell suspension was then filtered through a 30 µm cell strainer and transferred to a new 15 ml tube. The cell counts were determined, and cells were pelleted by centrifuging at $300 \times g$ for 5 min. After removing the supernatant, the pellet was resuspended in 200 µl of ice-cold PBS and 800 µl of freezing-cold 100% methanol was added. The cell suspension was snap-frozen in liquid nitrogen and stored at −80 °C until library preparation. After rehydration of the samples, libraries were generated using 10x Chromium and sequenced with the Illumina NovaSeq 6000 platform. Samples were subjected to scRNAseq using 10× Genomics Chromium Single Cell 3' Gene Expression system with feature barcoding technology for cell multiplexing. Demultiplexed, raw paired-end scRNAseq data were processed using the Cell Ranger (v. 7.10) from 10x Genomics. Data were analyzed in three biological replicates, where each replicate was composed of 48 individual MOs. Sequencing was carried out in two separate batches. In the first batch, replicates 1 and 2 were processed, while replicate 3 was processed in the second batch. In this manner, we assessed reproducibility of the sequencing results across different independent runs. To compare our MOs to other MOs generated before, we employed scRNAseq datasets of MOs that were previously generated at different time points[38]. The datasets were obtained from GSE133894. Cell types were annotated following the approach described before[19], where eight cell types were determined based on the expression levels of their marker genes (endothelial cells, non-mDANs, mDANs, glia, NBs in vitro, NPCs, and pericytes). The percentage of each cell type with respect to the total cell counts was computed. Integration of the two datasets was performed using the *FindIntegrationAnchors()* function described before[71] with default parameters using the Seurat R package (v. 4.2.0)[123]. The scRNAseq data of MOs can be retrieved under accession code NCBI GEO: GSE271852. scRNAseq data of Leigh COs and isogenic control COs were carried out

as before[15] and can be retrieved under accession code NCBI GEO: GSE126360. Pseudotime analysis was performed on the RG, IPCs, and neurons populations in those scRNAseq data using Monocle3 R package[30]. Input genes were differentially expressed genes among the three populations with the adjusted p value cutoff <= 0.05. The DDRTree algorithm was used for the inference.

## scRNAseq of compound-treated MOs

35-day-old MOs from isogenic CTL and from Leigh treated with either 0.1% DMSO (v/v), 0.1 μM sertaconazole, or 1 μM talarozole were dissociated for scRNAseq following the instructions of the Neural dissociation kit (Miltenyi) with a few modifications. Between 40-50 individual MOs were collected onto a 6 well-plate, washed with PBS, and incubated with 1 ml of Enzyme mix 1 for 5 min at 37 °C in an orbital shaker. Using a 1 ml pipette, the organoids were gently triturated by mixing up and down 10 times before adding Enzyme mix 2 for a second incubation. After 5 min incubation, the dissociation was stopped by adding 5 ml of stop solution. Cells were filtered through a 40 μm cell strainer and centrifuged at 300 × g for 5 min at 4 °C. After removing the supernatant, the pellet was resuspended in 4 ml of 0.04% BSA (bovine serum albumin) and further strained through a 70 μm cell strainer. The BSA solution was removed after a second centrifugation, and the obtained pellet was processed for fixation following the manufacturer's instructions (10X Genomics). The pellet was resuspended in 1 ml of fixation buffer for overnight incubation at 4 °C. The next day, samples were centrifuged at 850 × g for 5 min at RT, and the supernatant was removed. The obtained pellet was mixed 5 times in 500 μl of Quenching Buffer together with 50 μl pre-warmed Enhancer and glycerol (final concentration 10%). scRNAseq was carried out as described above. Unbiased clustering was performed using the *FindClusters()* function at a resolution of 0.3, followed by 2D visualization with the *RunUMAP()* function. Dot plots of selected marker genes for each cluster were generated using the *DotPlot()* function. Pseudotime analysis of selected cell clusters representing RG, IPCs, and neurons was carried out using the Monocle3 R package[30]. Input genes were differentially expressed genes among the three populations, filtered by an adjusted *p*-value ≤ 0.05. The DDRTree algorithm was used for trajectory inference. Gene set enrichment analysis (GSEA) was performed on differentially regulated genes in each cluster between LS and control organoids using the enrichR R package[122] with the "GO_Biological_Process_2021" database.

## Immunostaining of MOs

Free-floating MOs were fixed with 4% PFA for 20 min at RT, followed by three 10 min washes with PBS. Organoids were blocked and permeabilized with a solution consisting of 3% BSA, 0.5% Triton-X-100 and 0.05% sodium azide in PBS, for 1 h at RT. Primary antibodies were diluted in blocking solution and incubated overnight at 4 °C. Afterwards, MOs were washed 3 times with PBS and incubated overnight at 4 °C with secondary antibodies at a 1:1000 dilution in blocking solution. Next, organoids were incubated for 1 h at RT with Hoechst 33342 (1:1000) and washed three times with PBS. Images were acquired using the ZEISS Axio Observer microscope with an apotome 3 as a z-stack, which were then deconvoluted using the Zeiss blue software default settings and z-projected with maximum intensity. Antibodies are reported in Supplementary Data 6. For quantification of image intensity of Hoechst, MAP2, SMI312 and TH within MOs, immunostained MOs z-projected images were analyzed using Columbus software (v. 2.9.0). Prior to analysis, images were downscaled using the greyscale function in Photoshop (v. 12.1 ×64) and resized from 300 to 72 pixels per inch. The images were converted from 8-bit to 16-bit channels to retain image information and saved as.tif files. An image analysis pipeline was established using different building blocks. MO objects were identified based on the stained channel using the "Find Image Region" block. The "Fill Region" and "Border" functions within the

"Select Region" block were utilized to cover the entire organoid area. Within this defined area, the intensities of each channel were calculated using the "Calculate Intensity Properties" block and given as the average pixel intensity. Furthermore, morphological parameters such as area, width, length (all measured in μm), and roundness were assessed using the "Calculate Morphology Properties" block. For statistical comparison, we calculated the ratio of intensities of respective channels and the area of the individual MO.

## Intracellular calcium imaging of MOs

28-37 -day-old MOs were stained with the membrane-permeable form of the calcium indicator Fura-2 (Fura-2 AM; 1 mM stock solution in 20% pluronic/DMSO; 15 μM final concentration) dissolved in artificial cerebrospinal fluid (ACSF), containing (in mM): 138 NaCl, 2.5 KCl, 2 CaCl$_2$, 1 MgCl$_2$, 1.25 NaH$_2$PO$_4$, 18 NaHCO$_3$, and 10 glucose (pH 7.4; osmolarity ~310 mOsm/l) at 37 °C in a humidified atmosphere with 95% O$_2$ and 5% CO$_2$ for 30 min, followed by additional 30 min in ACSF without dye. A single MO was placed onto an Omnipore membrane filter and stabilized by a grid in the recording chamber. MOs were continuously superfused (2–2.5 ml/min) with ACSF bubbled with 95% O$_2$ and 5% CO$_2$ and allowed to equilibrate for ~10 min before starting recordings. Experiments were carried out at $37 \pm 1$ °C. Wide-field calcium imaging was performed using an upright microscope equipped with a 40×/N.A. 0.8 water immersion objective. Fura-2 AM was alternately excited at 380 and 400 nm using a monochromator, and images were acquired at a frequency of 1 Hz with a complementary metal-oxide-semiconductor (CMOS) camera. Fura-2 AM emission was collected at $510 \pm 40$ nm from regions of interest (ROIs) manually drawn around individual cell bodies. Background correction was performed as previously described[124]. Spontaneous calcium activity was analyzed during a 5 min recording period, in which no other manipulation was performed. Cells with calcium signals ≥ 2x standard deviation (SD) were considered active. Metabolic stress was induced by perfusing preparations with a glucose-free ACSF containing 2 mM 2-deoxyglucose and 5 mM sodium azide for 2 min. Changes in fluorescence were normalized to the baseline and are expressed as percentage change.

## Generation of iNs for neuromorphogenesis assay

To generate induced neurons (iNs), NGN2 lentiviruses were produced using HEK 293 cells seeded at a 70% confluence in a 150 cm² dish in DMEM medium[125]. Once attached, the medium was refreshed with DMEM supplemented with 25 μM chloroquine and cells were transfected with Lipofectamine 2000 according to the manufacturer´s protocol using the following plasmid mix: 8.1 μg pMD2.G, 12.2 μg pMDLg-pRRE, 5.4 μg pRSV-Rev, 5.4 μg FUW-M2-rtTA, 14.3 μg TetO-FUW-NGN2, 14.3 μg TetO-FUW-EGFP[126,127]. The next day, the medium was exchanged with DMEM-F12 containing 10 μM sodium butyrate. The supernatant containing viral particles was collected 24 and 48 h later. To concentrate the virus particles, the supernatant was first centrifuged at 500 × g for 10 min to remove cells and debris and then mixed with 1 volume of cold Lenti-X concentrator to every 3 volumes of lentivirus-containing supernatant. This mixture was kept overnight at 4 °C and further centrifuged at 1500 x g for 45 min at 4 °C. The obtained pellet containing the virus was diluted 1:100 in PBS. Virus aliquots were stored at −80 °C until further use. For neuronal morphogenesis screening, NPCs were seeded onto Matrigel-coated 96-well black Greiner-bio plates at a density of 40,000 cells/well in 100 μl of sm+ medium. The next day, the media were refreshed with sm+ supplemented with 4 μg/ml polybrene and 2.25 × 10⁶ transducing units of NGN2 lentivirus. After overnight incubation, NPCs were washed 3 times with PBS and the media were refreshed with sm+ containing 2 μg/ml of doxycycline to start neuronal induction (day 0). On day 3, medium was exchanged to NGN2 medium, consisting of Neurobasal, B27 supplement with vitamin A (0.5×), NEAA (1×), 2 mM Glutamine, MycoZap-Plus-CL (1×), 10 ng/ml human NT-3, 10 ng/ml human BDNF

supplemented with 2 µg/ml doxycycline. On day 3, treatments with DLDs and YSDs were administered in the media using 3 different concentrations (1, 10, and 50 µM). Final concentration of DMSOs in the media were either 0.1% or 0.4% (for 50 µM treatment of DLD4 and DLD5), which was equivalent to the amount of solvent in the samples that received compound treatment. On day 5, after 48 h incubation with the compounds, iNs were fixed by adding equal volumes of 8% PFA to the medium. Cells were incubated in a final dilution of 4% PFA for 20 min and washed 3 times with PBS for 10 min/wash. For staining, iNs were blocked and permeabilized for 1 h at RT with blocking solution (3% BSA, Triton-X-100, 0.05% sodium azide in PBS). Primary antibody incubation with anti-MAP2 (1:1000) in blocking solution was carried out overnight at 4 °C. Afterwards, cells were washed 3 times with PBS and treated with Alexa Fluor anti-Guinea pig AF568 secondary antibody (1:1000) and Hoechst 33342 (1:2500) in blocking solution for 1 h at RT. Next, after 3 washes with PBS, wells were filled with 0.05% sodium azide in PBS and stored at 4 °C wrapped with parafilm until imaged. Antibodies are reported in Supplementary Data 6. iNs were imaged with the high-content microscope Operetta using the 0.4 NA 20x air objective. For each well, 20 areas across the well were selected for imaging. The areas remained constant for each plate analyzed. Images of Hoechst and MAP2 channels were then analyzed with CellProfiler[128] to analyze neuronal morphogenesis as previously described (https://zenodo.org/records/6642365)[18]. Data were summarized by calculating the mean per well. To compare different plates, numbers were compared to the mean of the DMSO-treated wells.

## Yeast drug screen and concentration-dependent validation

Wild-type (WT) and SURF1-KO mutant (ΔSHY) strains were assessed for growth deficiency in a non-fermentable growth medium [YP medium (1% yeast extract, 2% bacto-peptone) plus 2% lactate] at 30 °C. A 384-well optimization assay was developed: 1) to identify the right starting density of cells that accentuates the growth difference phenotype and best assay time, 2) to establish that the assay has a good Z', which is a measure of assay quality and the likelihood of false positives and negatives in the screen[129]. In this assay, the Z' score was 0.7, indicating that the assay quality is excellent. Overnight cultures were diluted to an optical density (OD) of 0.025. 50 nl of 10 mM test compounds were added to columns 3-22 of 384-well plates to reach a final concentration of 20 µM, while 50 nl of DMSO was added to control columns 1–2 and 23–24, reaching a final concentration of 0.72 % (v/v). ΔSHY and WT cell suspensions were adjusted to an OD of 0.025, and 25 µl of mutant cells were added to columns 1-22, and WT cells to columns 23–24. The plates were spun at 250 × g for 1 min, sealed with a gas-permeable seal, and covered with a water-filled lid, then incubated at 30 °C for 24 h. After incubation, seals were removed and plates were equilibrated at RT for 15 min. 25 µl of Bactiter-glo solution was added to each well, plates were shaken for 2 min, incubated for 10 min, and luminescence was read using an Envision plate reader. Test compounds used in yeast drug screening belong to the UCSF Pharmakon library, including 1600 repurposable drugs that are FDA-approved with an existing safety record (Supplementary Table 1). A list of hits was generated, with the top 2% selected for further analysis. Concentration-dependent validations for sertaconazole and talarozole were carried out using the same procedures for either 24 h or 48 h.

## Generation of midbrain dopaminergic neurons (mDANs)

The generation of mDANs from NPCs was performed following a previously published protocol[15,130]. To start differentiation, NPC media was switched to a mixture of 1:1 of Neurobasal:DMEM-F12, N2 (0.5×), B27 with vitamin A (0.5×), 2 mM glutamine and Mycozap-Plus-CL (1×) supplemented with 200 µM ascorbic acid, 100 ng/ml FGF8, and 0.5 µM purmorphamine for 7 days. The following two days, the concentration of ascorbic acid and purmorphamine was brought down to 100 µM and 0.25 µM, respectively. On day 9, cells were passaged with Accutase

and seeded onto Matrigel-coated plates with 10 µM ROCK inhibitor in maturation media, made of a mixture 1:1 Neurobasal:DMEM-F12, N2 (0.5×), B27 with vitamin A (0.5×), 2 mM glutamine, Mycozap-Plus-CL (1×), with the addition of 200 µM ascorbic acid, 10 ng/ml BDNF, 10 ng/ml GDNF, 500 µM dibutyryl-cAMP, and 1 ng/ml of TGF-β3. The media was refreshed every other day, and the neurons were kept in culture for 4 and 8 weeks to reach different maturation stages.

## Mitochondrial movements in mDANs

mDANs cultures were grown on 35 mm dishes with coated bottom and 1.5 coverslips. 25 nM MitoTrackerRed CMXRos was added for 10 min and then replaced with DA culture media. Live-cell imaging recordings were conducted using a spinning disk microscope with incubating conditions of 37 °C with 5% $CO_2$. Cells were imaged every 2 sec using a 40× oil objective. The raw image files were stored as 16-bit in ".nd2" format at 337 × 337 µm (1024 × 1024 pixel) at an interval of 2 sec and a total of 200 images per series (total time per series: 6:40 min). The image pre-processing was carried out with Fiji (ImageJ 1.52 h) adapted from a previous publication[131]. To compensate for photobleaching during the time series, the "Bleach Correction" tool was applied, followed by a top-hat spatial filter to increase gray values of mitochondrial objects. The total mitochondria count was calculated by creating a binary image where the minimum gray value (min) was set to min=Mean+StdDev and the maximum gray value (max) was set to max= Mean+StdDev+1, where "Mean" and "StdDev" were obtained by using the "Measure" tool. Next, the "watershed" tool was applied to break up large mitochondria networks. Total number of mitochondria was quantified using the "Analyze Particles" tool with standard settings and a size preference for 8–200 pixel. The moving mitochondria were defined by a particle size of at least 6 pixels that changed location over the time course of 4 frames ( = 8 s). This was accomplished by subtracting the following 4th frame for each frame in the time series (on the top-hat filtered image). Afterward, the subtracted image series was converted into a binary image series. The number of moving mitochondria was similarly counted using the "Analyze Particles" tool with standard settings and a size preference for 6–200 pixel. Finally, the percentage of moving mitochondria per sample was calculated by dividing the average number of moving mitochondria by the average number of total mitochondria, multiplied by 100. The percentage of stationary mitochondria was obtained by the subtraction of moving mitochondria from 100.

## Western blotting

For western blots, MOs were treated from day 2 until day 28-30 of differentiation with either DMSO (0.1% v/v), 0.1 µM sertaconazole or 1 µM talarozole. On day 28-30, 50 to 90 individual MOs per condition were collected in a 1.5 ml tube and washed 3 times with DPBS. MOs were then pelleted using a tabletop centrifuge at 8,000 rpm for 5 min. mDANs were treated with either DMSO (0.1% v/v), 0.1 µM sertaconazole or 1 µM talarozole every other day from day 8 until day 28–30 of differentiation. For each condition, one full 6-well plate of treated mDANs was washed 3 times with DPBS, collected with a cell scraper into a 1.5 ml tube and pelleted using a tabletop centrifuge at 8000 rpm for 5 min. Fresh pellets were weighed and resuspended in appropriate volume of RIPA Lysis Buffer supplemented with proteinase inhibitor and phosphatase inhibitor (1:100) cocktails. Samples were then transferred to tubes containing CK14 ceramic beads and processed using a tissue homogenizer. The suspension was incubated on ice for 30 min, followed by centrifugation for 10 min at 10,000 rcf at 4 °C. Protein levels in the supernatant were then quantified using the BCA protein assay reagent kit. Protein lysates (35 µg) were separated on 4–12% Bis–Tris gels, transferred to PVDF membranes, which were blocked with 3% skimmed milk powder in TBS-T and incubated with antibodies against SNPH, KIF5A and ACTB as a housekeeping protein. The details of all antibodies are reported in Supplementary Data 6. After overnight

incubation at 4 °C, membranes were washed and incubated with secondary antibodies, horseradish peroxidase-conjugated anti-rabbit or anti-mouse. Signals were visualized using ECL solutions and ChemiDoc Touch Imaging System and the intensity of the bands was quantified using Image Lab software (v. 6.1.0). The intensity of the bands of interest (SNPH and KIF5A) was normalized to that of ACTB in the samples. Values of treated MOs are expressed relative to DMSO-treated Leigh MOs.

## Targeted metabolomics
NPCs were grown in Matrigel-coated 6-well plates until reaching 70% confluence. NPCs were either left untreated or were treated with 0.1% (v/v) DMSO, 10 μM sertaconazole, or 10 μM talarozole for 24 h. At this point, accutase was used to detach the cells from the plate. Cells were then centrifuged at 200 × $g$ for 3 min and the obtained pellets were dissolved in PBS to allow for quantification of cells using an automatic cell counter. Using a gating size of 5–19 μm, 1 million NPCs/sample were pelleted for metabolomics and 4 million NPC/sample for lipidomics in a 1.5 ml tube using a tabletop centrifuge at 8,000 rpm for 5 min at 4 °C. The supernatant was discarded, and the pellet was snap-frozen in liquid nitrogen for subsequent measurement. Extraction of the targeted metabolites was performed by adding 300 μl methanol/$H_2O$ (80/20; v/v) to the 1 ×10⁶ cells, which were homogenized by ceramic beads CK14 using Precellys lysing kit. The corresponding isotopically labelled internal standards were added to the extraction solution. The targeted compounds were analyzed by UPLC-MS/MS. The system consists of a UPLC I-Class (Waters) coupled to a tandem mass spectrometer Xevo-TQ-XS (Waters). Electrospray ionization was performed in the negative ionization mode for the compounds fructose-1,6-bishphosphate, phosphoenolpyruvate (PEP) and 2/3-phosphoglycerate[132]. The TCA, energy metabolites (AMP, ADP and ATP) as well acetyl- and succinyl-CoA were measured in the positive ionization mode[133–135]. Mass spectrometric quantitation of the compounds was carried out in the multiple reaction monitoring (MRM) mode. MassLynx software (v. 4.2) was used for instrument's control and data acquisition. Quantitation analysis was performed by TagetLynx XS software.

## Targeted lipidomics
NPCs were prepared as explained for targeted metabolomics. Approximately 4 million cells were homogenized in 400 μl of Milli-Q water using a Precellys 24 Homogenisator (Peqlab) at 6.500 rpm for 30 sec. The protein content of the homogenate was routinely determined using bicinchoninic acid (Supplementary Data 5).

*Glycerophospholipids and cholesterol:* To 50 μl of cell homogenate, 450 μl Milli-Q water, 1.875 ml methanol/chloroform 2:1 (v/v) and internal standards (126 pmol PC 17:0-20:4, 138 pmol PE 17:0-20:4, 111 pmol PI 17:0-20:4, 118 pmol PS 17:0-20:4, 60 pmol PG 17:0-20:4, 50 pmol d7-PA 15:0-18:1, 1.3 nmol d7-cholesterol; Avanti Polar Lipids) were added. Lipids were extracted using the "One-Step Extraction" described before[136]. Glycerophospholipids (PC, PE, PI, PS; PG, PA) were analyzed by Nano-Electrospray Ionization Tandem Mass Spectrometry (Nano-ESI-MS/MS) with direct infusion of the lipid extract (*Shotgun Lipidomics*) as previously described[136,137]. Endogenous lipid species were quantified by referring their peak areas to those of the respective internal standard. The calculated lipid amounts were normalized to the protein content of the cell homogenate. Cholesterol levels in the lipid extract were determined by Liquid Chromatography coupled to Electrospray Ionization Tandem Mass Spectrometry (LC-ESI-MS/MS) as previously described[138].

*Sphingolipids:* To 50 μl of homogenate, 50 μl Milli-Q water, 500 μl methanol, 250 μl chloroform and internal standards (100 pmol d7-ceramide 15:0, 130 pmol glucosylceramide 12:0, 124 pmol sphingomyelin 12:0; Avanti Polar Lipids) were added. Lipid extraction, alkaline hydrolysis of glycerolipids and LC-ESI-MS/MS analysis of ceramides, hexosylceramides, and sphingomyelins were performed as previously described[139,140].

## Cholesterol and lipids imaging
For cholesterol analysis, 40,000 NPCs per well were seeded onto a Matrigel-coated 96-well black Ibidi plates. The next day, NPCs were either fed with normal media (untreated condition) or treated with either 0.1% (v/v) DMSO, 10 μM sertaconazole, or 10 μM talarozole for 48 h. For the 24 h treatment, NPCs were treated with the above-mentioned conditions two days after. Both 24 h and 48 h treatment conditions were fixed on the same day using 4% PFA for 15 min at RT and washed 3 times for 10 min with PBS. For PFO-GST labelling of membrane-bound cholesterol, fixed NPCs were blocked and permeabilized with 10% goat serum in 0.1% Triton-X-100, PBS (0.1% PBST) for 1 h and incubated with 15 μg/ml recombinant PFO-GST for 3 h[141]. After washing the cells three times with 0.1% PBST, anti-GST (1:200) was applied in 10% goat serum 0.1% PBST, overnight at 4 °C. After three washes with 0.1% PBST, Alexa Fluor™ 488 goat anti-mouse IgG (H + L) secondary antibody was applied (1:500) for 1 h at RT. Lastly, wells were washed three times in PBS and cells were preserved in Mounting Medium with DAPI.

For neutral lipid labelling, fixed cells were incubated with 1 μg/ml BODIPY 493/503 in PBS for 20 min, followed by three washes of 5 min with PBS, and mounting medium with DAPI was added to the wells. Fluorescent images were acquired with a LSM 900 Zeiss confocal microscope. Laser power, gain, and offset parameters were kept constant for each experiment; any subsequent adjustments to contrast and brilliance were applied equally to all images. Image analysis and quantification were performed using Fiji ImageJ software using custom-made macros.

## Molecular docking with GOLD
To test sertaconazole and talarozole binding to postulated targets, we performed in silico docking experiments using the docking tool GOLD[49]. We modeled docking to the protein targets of talarozole: cytochrome P450 26 (CYP26) enzymes, specifically CYP26A1 and CYP26B1. Since there is no experimental structure reported to date for CYP26A1/B1, we obtained CYP26A1/B1 from the AlphaFold[142] database based on PDB:2VE3 which reports the structure for retinoic acid bound to cyanobacterial CYP120A1. Overlap between the two protein structures enabled the localization of the iron-heme and the substrate structure within the cavity of CYP26A1/B1[143]. Removing the substrate and redocking it together with sertaconazole and talarozole revealed differences in poses and docking scores. CCDC Gold (v. 2022) was used for docking and CCG MOE (v. 2022.02) was used to visualize the results and generate images. Compound poses were analyzed using docking scores and by the minimal distance of any compound´s atom from the heme-iron reactive center for oxidation. The combination of distance from the iron atom and the relative energy needed to reach the ligand's position can suggest whether compounds act as substrate or as inhibitor, since inhibitors are usually closer to the iron-center, disturbing the oxygen recruitment from heme[144,145].

## Molecular docking with DiffDock and SMINA
The docking protocol was performed using a combination of DiffDock[53] and SMINA[54]. The protein structures of human ligand-binding domain of the peroxisome proliferator-activated receptor gamma were obtained from PDB (2FVJ: PPARγ bound to PA-082[58], 3B1M: PPARγ bound to Cerco-A[59], and 6L8B: PPARγ ligand-free[60]). Residues not belonging to the receptor were removed, and the residues corresponding to missing loops were modelled using CHARMM-GUI[146]. The predicted structure of human cytochrome P450 26A1 and 26B1 was obtained from the AlphaFold Database (O43174 and Q9NR63, respectively)[142,147] and the transmembrane region was removed (residues 1-45 and 1-41, respectively). Each structure was then solvated in

explicit OPC water, adding counter ions to neutralize the system when necessary, and minimized with the Amber ff19sb force field[148] using the steepest descend algorithm in a two-step process: a first minimization with constraints on the heavy atoms of the protein backbone (400 kJ/mol) and side chains (40 kJ/mol), and a second unconstrained minimization. The DiffDock algorithm was used to perform a global, blind docking. Minimized structures were used as receptor input to dock each of the ligands, which were provided in SMILES format. A total of 40 poses per complex (receptor-ligand) were generated, with 20 inference steps each, ranked by confidence score. The top 5 poses were then refined with SMINA, using the pose to generate the searching box and adding 4 Å buffer space to allow ligand re-accommodation. Only a local search with an additional 200 steps of steepest descent energy minimization was performed. The lowest energy refined pose per complex was selected for analysis and plotting.

### PCR and gene expression analyses

For quantitative real-time PCR (qPCR) analysis of CYP26B1 in iNs, 2 million NPCs were seeded onto Matrigel-coated 6 well-plates one day prior to NGN2 lentiviral transfection supplemented with 4 μg/ml polybrene. After overnight incubation with NGN2 lentiviruses, cells were washed 3 times with PBS and the media was refreshed with sm+ containing 2 μg/ml of doxycycline to start neuronal induction (day 0). Three days later, media was exchanged to iN media (see *Neuromorphogenesis analysis in iNs* for media formulation) with either DMSO (0.1% v/v), 10 μM sertaconazole, or 10 μM talarozole. On day 5, treatment was refreshed to incubate for another 6 h to observe changes in gene expression[149]. Afterwards, cells were collected using a cell scraper, pelleted using a tabletop centrifuge at 8,000 rpm for 5 min, and snap-frozen in liquid nitrogen. RNA extraction from pellets was performed with RNeasy kit. cDNA was generated from 2 μg RNA using the first strand cDNA synthesis kit. qPCR experiments were conducted for three technical replicates and three biological iNs replicates with the SYBR green Mastermix and CFX96 Real-Time System (Bio-Rad). After averaging the CT values of the three technical replicates, the $2^{-\Delta\Delta CT}$ method was used. The expression levels of the genes of interest were normalized relative to the average expression of housekeeping genes (*C1ORF43* and *YWHAZ*)[150] and relative to DMSO treatment. Primer sequences are reported in Supplementary Data 6. Expression of anchoring and RA pathway genes was obtained from our previously published bulk RNAseq datasets for Leigh mDANs and Leigh COs[15].

### Statistical analysis

Data are expressed as mean and standard deviation (mean ± SD) where normality of the distribution could be verified, or as median and quartiles (median [1st; 4th quartiles]) otherwise. Significance was assessed using parametric tests (Welch's t-test, ANOVA) for normally distributed data and non-parametric tests (Mann-Whitney U test, Kruskal-Wallis) when normal distribution could not be verified. Unless otherwise indicated, data were analyzed and shown using GraphPad-Prism software (v. 10.2.2) (GraphPad Software, USA) or R (v. 4.10). Figures were prepared with Adobe Illustrator (v. 28.5). Schematic cartoons were made using Biorender.

### Reporting summary

Further information on research design is available in the Nature Portfolio Reporting Summary linked to this article.

## Data availability

There are restrictions to the availability of patient-derived iPSCs due to our ethical approval that does not support sharing with third parties without a specific amendment and does not allow performing genomic studies to respect the European privacy protection law.

Single-cell RNA sequencing (scRNAseq) data are deposited in the Sequence Read Archive (SRA) with bioproject number PRJNA1378526 and can be downloaded at: https://www.ncbi.nlm.nih.gov/bioproject/?term=PRJNA1378526.

Additional scRNAseq data used in this study have been previously deposited in the Gene Expression Omnibus (GEO) database: GSE152915, GSE126360, (https://www.ncbi.nlm.nih.gov/geo/query/acc.cgi?acc=GSE126360), GSE133894, GSE271852.

Lipidomics data are deposited in Metabolomics Workbench repository[151] with project number PR002944 and can be downloaded at: https://doi.org/10.21228/M84C4F. Source data are provided with this paper.

## Code availability

The program code and input files are available at: https://github.com/OSB-codes/DL_drug_repurposing/tree/main.

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

## Acknowledgements

We thank Jens Schwamborn (Luxembourg University) and his group for providing their published MO datasets. We acknowledge support from the Deutsche Forschungsgemeinschaft (DFG) (PR1527/5-1, PR1527/6-1, PR1527/15-1, PR1527/14-1, and PR1527/13-1 to A.P.; RO5380/1-1 to A.R.; Ro2327/13-2 to C.R.R.; AL2956/1-1 to I.A-M.; VE366/12-1 to N.V.), the European Joint Programme on Rare Diseases (EJP RD) and Federal Ministry of Research, Technology and Space (BMFTR) (Cure-eMILS #01GM2002A to A.P., #01GM2002B to O.P.), the Medical Faculty of Heinrich Heine University (FoKo grant to A.P. and S.C.), the European Commission's Horizon Europe Programme (SIMPATHIC #101080249 to A.P. and M.S.), AFM-Téléthon (SildeMITO #28545 to A.P. and M.S.; GenT-Leigh #29044 to A.P.), the United Mitochondrial Disease Foundation (UMDF) and the Leigh Syndrome International Consortium (LSIC) (to A.P.), People Against Leigh syndrome (PALS) (to A.P.), the Foundation Maladies Rare and Association (AMMi) (to A.P.), the patient organizations Mitocon, Cure Mito, Cure ATP6, Mito Foundation, and MitoHelp (to A.P.), and the Eva Luise Köhler Foundation (to A.P. and M.S.), the Studienstiftung des deutschen Volkes (to S.L.), the Agència de Gestió d'Ajuts Universitaris i de Recerca (AGUAR) Generalitat de Catalunya through the Indústria del Coneixement LLAVOR program (grant 2024 LLAV 00069 to E.Pu.), National Science Center OPUS (#2017/27/B/NZ1/02401 to P.L.), and funds from MICINN (PID2021-125079OA-I00 to E.Pu.). A.Sp. and I.J.H. receive research support from the UK Medical Research Council (MR/X002365/1), Muscular Dystrophy UK (17GRO-PG24-0184-1), and the CHAMP Foundation. A.Sp. is the recipient of a Miriam Marks Senior Fellowship, Brain Research UK (202021-26) and receives funds from the Lily Foundation. I.J.H. is supported by grants from the Instituto de Salud Carlos III PI20/00096, the Spanish Ministry of Science, PID2023-151649NB-I00, and the Basque Government Department of Health (Osasun Saila, Eusko Jaurlaritzako) (grants 2021111070; 2022333050; 2018111043; 2018222031). M.M.O. was partially supported by the Ikerbasque, Basque Foundation for Science IKUR strategy, Neurodegenprot project. S.L. was supported by the Studienstiftung des deutschen Volkes. J.F-C acknowledges support from the MICINN (PID2022-142956OB-100 and PID2025-173094OB-I00), the National Cancer Institute PO1 CA281819-O1A1 from the NIH, and the CIBEREHD and the European Horizon´s research and Innovation program HORIZON-HLTH-2022-STAYHLTH-02 under agreement #101095679). We are grateful to Anje Sporbert (Advanced Light Microscopy, MDC, Berlin) for help with the mitochondrial movement recordings and acknowledge the Center for Advanced Imaging (CAi) at Heinrich Heine University Düsseldorf for providing access to the PerkinElmer Operetta CLS (DFG grant # INST 208/760-1 FUGG) and Olympus FV3000 microscope.

## Author contributions

Conceptualization, A.P., C.M., A.D.S.; Methodology, C.M., S.O., A.W., M.M-O., L.P., M.T., S.Z., D.H., A.Se., G.I., C.J., S.L., S.B., I.A-M.; Formal Analysis, S.O., T.M.P., C.M., A.Z., I.A-M., B.M., A.R-W., J.D., M.L.M.; Resources, E.M., A.P., A.D.S., P.L., A.Sp., E.Pe., F.D., A.R., N.R., K.W., F.X.S., J-F-C., N.V., S.C., M.S., E.Pu., C.R.R.; Writing – Original Draft, C.M., A.P.; Writing – Review & Editing, A.P., C.M., S.O., A.D.S., O.P., C.R.R., L.P., A.S., I.J.H., I.A-M., E.Pu.; Supervision, A.P., A.D.S., O.P., I.J.H., C.R.R., F.D., A.P-M.; Visualization, C.M., A.P.; Funding Acquisition, A.P., A.D.S., O.P.

## Funding

## Competing interests

C.M., S.O., A.D.S., and A.P. filed a patent application for the use of talarozole in mitochondrial diseases. C.M., K.W., E.Pe, and A.P. filed a patent application for the use of sertaconazole in mitochondrial diseases. The remaining authors declare no competing interests.

## Additional information

[1]Department of General Pediatrics, Neonatology and Pediatric Cardiology, Medical Faculty and University Hospital Düsseldorf, Heinrich Heine University Düsseldorf, Düsseldorf, Germany. [2]Faculty of Mathematics and Natural Sciences, Heinrich Heine University Düsseldorf, Düsseldorf, Germany. [3]Computational Biology Group, Luxembourg Centre for Systems Biomedicine, University of Luxembourg, Esch-sur-Alzette, Luxembourg. [4]University of Pittsburgh School of Medicine, Vascular Medicine Institute, Pittsburgh, PA, USA. [5]Institute of Neurobiology, Heinrich Heine University, Düsseldorf, Germany. [6]Fraunhofer Institute for Translational Medicine and Pharmacology ITMP, Discovery Research ScreeningPort, Hamburg, Germany. [7]Department of Neurosciences, Biogipuzkoa Health Research Institute, San Sebastian, Spain. [8]Max Delbrück Center for Molecular Medicine in the Helmholtz Association (MDC), Berlin, Germany. [9]Berlin Institute for Medical Systems Biology (BIMSB), Berlin, Germany. [10]Neuropsychiatry and Laboratory of Molecular Psychiatry, Department of Psychiatry and Neurosciences, Charité—Universitätsmedizin, Berlin, Germany. [11]Department of Molecular Biology, Institute of Genetics and Animal Biotechnology, Polish Academy of Sciences, Jastrzebiec n/Warsaw, Poland. [12]Unit of Biophysics, Department of Biochemistry and Molecular Biology Institute of Neurosciences,

Universitat Autònoma de Barcelona, Barcelona, Spain. [13]Charité—Universitätsmedizin, Berlin, Germany. [14]Perlara PBC, Vancouver, WA, USA. [15]Institute of Cell Biology, Heinrich Heine University, Düsseldorf, Germany. [16]Cluster of Excellence Cellular Stress Responses in Aging-associated Diseases (CECAD), Faculty of Medicine and University Hospital of Cologne, University of Cologne, Cologne, Germany. [17]Cure Mito Foundation, McKinney, TX, USA. [18]Celltec-UB, Departament de Biologia Cellular, Fisiologia i Immunologia, Institut de Neurociències, Universitat de Barcelona, Barcelona, Spain. [19]Department of Molecular and Cellular Biomedicine, Institute of Biomedical Research of Barcelona (IIBB), CSIC, Barcelona, Spain. [20]Liver Unit, Hospital Clinic i Provincial de Barcelona, Institut d'Investigacions Biomèdiques August Pi i Sunyer (IDIBAPS), Barcelona, Spain. [21]Centro de Investigación Biomédica en Red (CIBEREHD), Barcelona, Spain. [22]Department of Medicine, Keck School of Medicine, University of Southern California, Los Angeles, CA, USA. [23]IUF-Leibniz Research, Institute for Environmental Medicine, Düsseldorf, Germany. [24]Dept. for the Promotion of Human Science and Quality of Life, San Raffaele University of Rome, Rome, Italy. [25]Institute of Physiological Chemistry, University Medical Center of the Johannes Gutenberg University, Mainz, Germany. [26]Department of Clinical and Movement Neurosciences, UCL Queen Square Institute of Neurology, Royal Free Campus, London, UK. [27]Department of Neuropediatrics, Charité–Universitätsmedizin, Berlin, Germany. [28]NeuroCure Cluster of Excellence, Berlin, Germany. [29]German Center for Cardiovascular Research (DZHK), Berlin, Germany. [30]National Center for Tumor Diseases (NCT), German Cancer Consortium (DKTK), Berlin, Germany. [31]German Center for Neurodegenerative Diseases (DZNE), Berlin, Germany. [32]Genome Engineering and Model Development lab (GEMD), IUF-Leibniz Research, Institute for Environmental Medicine, Düsseldorf, Germany. [33]IKERBASQUE, Basque Foundation for Science, Bilbao, Spain. [34]University of the Basque Country-Bizkaia Campus, Bilbao, Spain. [35]CIBERNED (Center for Networked Biomedical Research on Neurodegenerative Diseases) Ministry of Economy and Competitiveness, Institute Carlos III), Madrid, Spain. [36]Neuroscience Institute, Department of Cell Biology, Physiology and Immunology, Autonomous University of Barcelona, Bellaterra, Spain. [37]CIC bioGUNE-BRTA (Basque Research and Technology Alliance), Bizkaia Technology Park, Derio, Spain. [38]Present address: Axol Bioscience Ltd, Berlin, Germany. [39]Present address: Centogene GmbH, Rostock, Germany. [40]These authors contributed equally: Carmen Menacho, Satoshi Okawa. [41]These authors jointly supervised this work: Antonio Del Sol, Alessandro Prigione. ✉e-mail: antonio.delsol@uni.lu; alessandro.prigione@hhu.de

