## [Transparent Peer Review file · Nature Communications]

Accelerating Leigh syndrome drug discovery through deep learning screening in brain organoids

Corresponding Author: Alessandro Prigione

Version 0:

Reviewer comments:

Reviewer #1

(Remarks to the Author)

In this study Menacho and colleagues propose a combination of drug screenings and deep learning in brain organoids to provide a novel approach for the identification of treatments for Leigh Syndrome. In this regard, this approach has the potential to become an invaluable platform to achieve unbiased identification of novel therapies for Leigh Syndrome and other mitochondrial diseases. Using this platform, authors identify azole compounds as novel candidates with therapeutic potential. Overall, the methodology is innovative and the results are sound. However, there are some points that I believe need to be clarified/addressed to provide mechanistic insight into the potential of azole compounds and, as a result, in the predictive feasibility of the platform.

- As mentioned above, the identification of azole compounds in two complementary screenings provides a strong indication of their potential. However, while there seems to be a clear correlation between the different treatments and the improvement in several parameters measured, there seems to be a lack of causal demonstration of these effects. In my opinion, demonstration that, in addition to gene expression, azole treatment alters the activity of CYP26 or RA (or some other downstream targets), or that blockade of RA activation ablates the beneficial effects of azole compounds would further strengthen their conclusions.
- Along the same lines, the suggestion that the beneficial effect of Talarozole may be mediated by its effect on PPAR γ is extremely relevant and fits with recent reports of this pathway being involved in amelioration of the pathology in mitochondrial disease models. It would be highly informative and provide mechanistic insight if authors could determine whether PPAR γ activity is altered by the treatments.
- As shown by authors, not all azole compounds were effective. Given that it is suggested that the activity may be mediated by their affinity to CYP26, do authors have information on the affinity for this factor of the ineffective/detrimental compounds?
- In Fig 2c (Leigh lactate release), would likely require showing the values of CTL lactate release to provide a clear view of the level of reduction in lactate levels. While it is probable that restoration does not achieve CTL levels this will help interpret the biological magnitude, and significance, of the changes.
- Additionally, while authors are correct that midbrain is affected in Leigh Syndrome, dopaminergic neurons are not the most affected neuronal cell type. Brainstem and basal ganglia are the most affected areas in patients. In this regard, it would be informative that authors would further discuss the use of midbrain organoids and whether this could be applicable to other neuronal types, which may be a limitation of the work to be determined in future experiments. As a minor note, reference 14 does not mention that dopaminergic neurons are strongly affected in Leigh syndrome, which would need to be corrected.

(Remarks on code availability)

Reviewer #2

(Remarks to the Author)

In this manuscript, Menacho and collaborators report the development of a deep learning algorithm coupled with organoids derived from Leigh Syndrome patients for the identification of potential drugs that can promote neuronal commitment. The authors used iPSC models derived from LS patients and validated twoazole compounds which were capable of lowering lactate release and potentially promote neurogenesis. There are several issues with scientific rigor that limit the potential of this study.

1. Leigh causes neuronal dysfunction in vivo, and while the organoid model suggests defects in neurogenesis, promoting differentiation of mutant neurons won't be a viable option. It is unclear what is the main purpose of this system. It is unlikely that Leigh mutations in vivo are blocking the ability of neural progenitors to commit to a neuronal fate.
2. Overall, there is a lack of description of the results. For example, the authors mention "significant defects occurring five days after NGN2 induction in Leigh iNS" but do not mention what these defects are and do not place these results in the context of the disease. Same is true for their deep-learning drug discovery pipeline. Of the drugs selected it is not clear which ones belong to the DETF1 or DETF2 category, and in a few lines narrowed down to 2 drugs increasing neuronal number with unclear mechanistic insight into the effects of these drugs on the transcription profile of these cells.
3. The investigators used previously generated scRNAseq datasets of isogenic cortical organoids with LS mutations to determine the key affected populations and to develop the deep learning strategy, not data of midbrain organoids. While the populations affected may be equivalent, validation of this assumption is imperative.
4. There are several inconsistencies of the data presented throughout the paper. This generated concerns about the variability between different batches of organoids. For example, in Figure 1i the sample derived from LS completely lacks TH staining, but in Figure 3d the results are less dramatic.
5. It is not clear how in Fig. S3d treatment with FCCP would have recapitulated increased motility observed in Leigh model, even from a theoretical standpoint. This would induce QC pathways that would lead to motor adaptors being targeted, thus decreased motility. Additional rationale and explanation for this conclusion is needed.
6. In the data shown in Figures 3g, 3h and S3e, there appears to be differential effects between the DA neurons and organoids in how these motor/anchor genes are impacted. Would be good to see if treating the DA neurons alone results in any rescue. Switching from single cell RNAseq data from cerebral organoids to batch data like a western blot of midbrain organoids doesn't give the appropriate resolution when seeing if the drugs rescue the phenotype in specific cell types. Figure 3e shows how an increase MAP2 with Sertaconazole and increase in SMI312 with Talarozole. The co-treatment could be informative. Figure 3d is clearly from the same staining as 3e, but not in the same panel.
7. Full characterization of the midbrain organoids is needed. Additional midbrain dopaminergic neuronal markers.
8. Issues with rigor: only 2 independent experiments were used in some cases. Some organoids used in different panels and variability in stainings are worrisome. There is no clear information of how many organoids and how many independent differentiations were used for analysis.
9. Limited discussion of the implications of the data. What are the mechanisms for increased neurogenesis? Is theazole compound main mechanism to reduce cell death? Lactate is also released by dying cells, but cell death was not measured in this study.
10. To have a clear idea of the effects of treatments scRNAseq after treatments would be recommended.
11. Assessing lipid metabolism via interrogating cholesterol is a limited, narrow analysis. Lipidomics coupled with changes in expression/activity of key enzymes in identified affected pathways is key.
12. Talarozole targets CYP26A1/B1. But then the authors state, after the modeling, that "both azoles could bind" the target protein based on mRNA expression across models, including DA neurons and iNs. However, no genetic manipulations of the system were included.
13. The use of scRNAseq to determine OCR, ATP, and glucose uptake needs to be validated. The publications cited as proof-of-principle did not use scRNAseq data to infer metabolic activity. Differential gene expression analysis provides the basis for testing potential hypotheses but not direct evidence of changes in metabolic activity.

Minor

- Recommendation: change brain organoid to neural organoid
- Referring to "our" MO should be changed to MO from LS or MO generated in this study
- Scale bars are sometimes different between

(Remarks on code availability)

Reviewer #3

(Remarks to the Author)

The authors describe the development of a deep learning algorithm to predict drug response from non-cancer perturbation and gene expression in order to validate these in their experimental models of Leigh syndrome. To my knowledge, this is a rare application of this predictive modeling in neurodegeneration and therefore is of significant importance to the community. In particular, because more traditional approaches to identify actionable targets in neurodegeneration have resulted - at best - in mixed clinical results.

As highlighted in the title, a key novelty of this work is applying the objective, data-led DL prediction of drugs in Leigh Syndrome

The authors conclude that the algorithm predicted 27 (or 28 - conflicting numbers are reported) chemical compounds, 9 of

which were experimentally tested.

Such a method, if it genuinely enriches the hit rate to identify active drugs, would be of tremendous value to the community. Which is why it is surprising that the authors do not provide any of the key metrics required to evaluate the benefit or added value of the DL method over standard bioinformatics approaches.

1. Performance metrics

A manuscript pivoting on using DL to make predictions must provide performance or metrics of the algorithm (F1, AUC, precision/recall etc). None are provided. What evidence is there that the DL method was useful in recommending drugs that exceeds what can be achieved using standard approaches of assessing pathway enrichment in differentially expressed genes without the application of DL?

2. Neural network architecture

The authors use a simple feed forward neural network with three hidden layers. This simplicity of the NN is sensible given the size of the data set. The general approach described is a standard one, but appropriate for the question being asked.

3. Training and validation

The authors state that they use iterative cross validation. However, the authors do not provide any information on the segmentation of the data, the cross validation results etc. What are the results of the final cross validation? How predictive is the algorithm? What are the false positive and false negative rates?

3. Independent test set

A major problem faced by the community in these approaches is overfitting: optimizing an algorithm on the training data where it cannot perform well outside these specific data. For this reason, an independent hold out test set, unseen by the algorithm during training and optimization is used to assess generalizability. Did the authors keep a hold out independent test set? If so, how did the algorithm perform? If not, how was algorithm generalizability assessed?

4. Selection of drugs for testing

The authors state that the algorithm predicted 27/28 compounds, of which 9 were selected for testing. Did the algorithm ONLY predict 27 compounds? What values were used to define a 'prediction'?

Importantly, what was the null hypothesis testing to demonstrate the value? How were the drugs predicted by DL these superior to what would have been obtained by simple annotation of differentially expressed pathways?

As it stands, it is unclear that the DL algorithm provided any value to the outcome of this study.

(Remarks on code availability)

The authors do not provide information on code availability

I was unable to find the specific Python function developed by the authors in the zip download file.

I also found the manuscript on BioRxiv and searched for the code there but with no avail

Reviewer #4

(Remarks to the Author)

(Remarks on code availability)

Version 1:

Reviewer comments:

Reviewer #1

(Remarks to the Author)

Authors have sufficiently and elegantly addressed my concerns, thus I have no further comments on the manuscript.

(Remarks on code availability)

Reviewer #2

(Remarks to the Author)

This reviewer recognizes the authors' efforts in addressing the reviewers' concerns and revising the manuscript, which presents a promising platform for drug discovery with the potential to advance understanding of rare diseases such as Leigh Syndrome. The improvements addressed some of the concerns that were raised about the clarity and rigor of the work. However, several critical issues remain that need to be addressed to meet the standards for publication in Nature Communications. The main concern is the consistent lack of control samples across multiple figures, which weakens the ability to evaluate the significance and specificity of the reported drug treatment effects.

Control samples are essential in all studies, but particularly in drug discovery research, as they provide a baseline for comparison to determine the specific effects of experimental treatments. In the current manuscript, many figures lack appropriate controls relevant to the drug discovery platform. This omission makes it difficult to interpret whether the observed effects are specifically due to the drugs acting on Leigh syndrome cells or influenced by other factors. For example: Figure 2b, Figure 2g/h (WT strains were not treated with azole compounds), Figure 4 (Control MOs were not treated with sertaconazole or talarozole), and Figure 6h (control NPCs were not treated with the drugs). Without controls in at least these critical experiments, it is unclear whether the platform consistently distinguishes actual drug effects in Leigh Syndrome samples from general effects in controls. Including control samples would allow researchers and labs interested in using this platform to quantify the magnitude of drug effects relative to baseline conditions, evaluate the specificity of drug responses, and demonstrate the robustness of the platform by showing its ability to reliably detect expected outcomes in Leigh syndrome samples and not in controls. If the drugs also enhance neuronal differentiation and activity in control samples, this does not negate their potential for treating rare diseases and may even support the broader use of this platform in various drug discovery applications.

(Remarks on code availability)

Reviewer #4

(Remarks to the Author)

(Remarks on code availability)

We thank the Reviewers for their important suggestions and insightful comments that have led to significantly improve the quality of our manuscript. Please find below our point-by-point reply in blue font.

Reviewer #1 (Remarks to the Author):

In this study Menacho and colleagues propose a combination of drug screenings and deep learning in brain organoids to provide a novel approach for the identification of treatments for Leigh Syndrome. In this regard, this approach has the potential to become an invaluable platform to achieve unbiased identification of novel therapies for Leigh Syndrome and other mitochondrial diseases. Using this platform, authors identify azole compounds as novel candidates with therapeutic potential. Overall, the methodology is innovative and the results are sound. However, there are some points that I believe need to be clarified/addressed to provide mechanistic insight into the potential of azole compounds and, as a result, in the predictive feasibility of the platform.

We thank the Reviewer for understanding the potential innovation and impact of our work. We performed several new experiments to provide additional mechanistic insights. We carried out single-cell RNAseq of Leigh midbrain organoids at the basal level (**Figure 3a-f**) and after treatment with talarozole and sertaconazole (**Figure 4h-k**). The transcriptomics of drug-treated organoids highlighted an impact of the drugs on lipid metabolism that we confirmed using lipidomics analysis (**Figure 5d-f**). We also validated the effects of the drugs on calcium activity in the organoids (**Figure 4e-g**) and on the yeast survival in a dose-dependent manner (**Figure 2g-h**). Lastly, using two complementary modeling strategies we performed additional docking experiments on the different azoles on the retinoic acid receptor and on the PPAR γ receptor (**Figure 6d-g**) which we validated using luciferase-based expression in Leigh neural cells (**Figure 6h**).

We believe that our results now provide increased mechanistic insights into the possible actions of the azole compounds in Leigh neural cells and demonstrate the potential strength of combining deep learning screens with brain organoids transcriptomics to identify potential innovative targets and treatments for rare neurodevelopmental diseases.

- As mentioned above, the identification of azole compounds in two complementary screenings provides a strong indication of their potential. However, while there seems to be a clear correlation between the different treatments and the improvement in several parameters measured, there seems to be a lack of causal demonstration of these effects. In my opinion, demonstration that, in addition to gene expression, azole treatment alters the activity of CYP26 or RA (or some other downstream targets), or that blockade of RA activation ablates the beneficial effects of azole compounds would further strengthen their conclusions.

We agree with the reviewer that our original submission lacked mechanisms and causality. We carried out extensive docking analysis on all the different azoles using two complementary approaches to assess their engagement on activity of CYP26 (**Figure 6a, Figure 6d-e, Figure S9a-b, Figure S10a**) and PPAR γ (**Figure 6f-g, Figure S10b**). We validated this engagement using luciferase-based expression of CYP26 and PPAR γ in mutant neural cells (**Figure 6h**). These results indicate that among all azoles, talarozole appears to be the one most capable of eliciting an activity of both pathways, which fits with our initially reported experimental data.

Sertaconazole instead was less efficient in modulating these pathways but had a bigger impact on modulation of lipid metabolism (**Figure 5d-j**). To clarify these differences, we included additional interpretations in the discussion (**lines 534-588**). We also restructured the whole manuscript and all the figures to present the findings in a clearer and more linear manner and to include more details and interpretations.

- Along the same lines, the suggestion that the beneficial effect of Talarozole may be mediated by its effect on PPAR γ is extremely relevant and fits with recent reports of this pathway being involved in amelioration of the pathology in mitochondrial disease models. It would be highly informative and provide mechanistic insight if authors could determine whether PPAR γ activity is altered by the treatments.

We thank the reviewer for this suggestion. We agree that PPAR γ is a very interesting target in Leigh. We explored its role by docking analysis and found that among all azoles, talarozole had the higher potential to engage with PPAR γ (**Figure 6f-g, Figure S10b**). We confirmed the effect of talarozole using luciferase-based gene expression analysis in neural cells (**Figure 6h**). However, we could not detect any difference in the PPAR γ expression at the basal level in Leigh NPCs compared to isogenic control NPCs. These data are in apparent contrast to previously generated data in a mouse model of Leigh.¹ This could be due to differences in the diseased genes (our work is based on *SURF1* mutations affecting complex IV, while the mice are KO for *Ndufs4* which is part of complex I) or species-specific differences. We included comments in the discussion regarding the interpretation of these data and possible directions for future studies (**lines 551-568**).

- As shown by authors, not all azole compounds were effective. Given that it is suggested that the activity may be mediated by their affinity to CYP26, do authors have information on the affinity for this factor of the ineffective/detrimental compounds?

To answer this important question, we carried docking analysis on all the different azoles to assess their engagement on activity of CYP26 using two different modelling approaches. The results showed that talarozole (ranked first) and sertaconazole (ranked second) were the azoles most likely to be able to bind to this receptor (**Figure S9a-b, Figure S10a**). These results suggest that engagement of the RA pathway might contribute to explaining the effects seen for these azoles in Leigh cells.

- In Fig 2c (Leigh lactate release), would likely require showing the values of CTL lactate release to provide a clear view of the level of reduction in lactate levels. While it is probable that restoration does not achieve CTL levels this will help interpret the biological magnitude, and significance, of the changes.

We apologize for the lack of clarity. The lactate release data for Leigh organoids vs control organoids is shown in **Figure 3j**. The lactate release data for Leigh organoids treated with the

azoles is shown in **Figure 4c**. In this latter graph, we included a horizontal grey dashed line to indicate the lactate release levels seen in control organoids.

- Additionally, while authors are correct that midbrain is affected in Leigh Syndrome, dopaminergic neurons are not the most affected neuronal cell type. Brainstem and basal ganglia are the most affected areas in patients. In this regard, it would be informative that authors would further discuss the use of midbrain organoids and whether this could be applicable to other neuronal types, which may be a limitation of the work to be determined in future experiments. As a minor note, reference 14 does not mention that dopaminergic neurons are strongly affected in Leigh syndrome, which would need to be corrected.

We thank the reviewer for this comment. It is in fact not clearly proven that dopaminergic neurons are specifically degenerating in Leigh. We removed the original sentence regarding dopaminergic neurons and related reference. Instead, we included a section in the discussion where we comment on the limitations of the current work and suggest potential future directions including monitoring the impact of Leigh mutations in different brain regions using various brain region-specific organoids and related assembloids (**lines 594-598**).

Reviewer #2 (Remarks to the Author):

In this manuscript, Menacho and collaborators report the development of a deep learning algorithm coupled with organoids derived from Leigh Syndrome patients for the identification of potential drugs that can promote neuronal commitment. The authors used iPSC models derived from LS patients and validated two azole compounds which were capable of lowering lactate release and potentially promote neurogenesis. There are several issues with scientific rigor that limit the potential of this study.

We thank the reviewer for highlighting aspects of our work that needed improvements. We realized that our article was formatted as a short article and therefore not sufficiently detailed. We therefore extensively restructured the text and figures to provide more details and clarifications also with respect to controls employed and scientific rigor.

1. Leigh causes neuronal dysfunction in vivo, and while the organoid model suggests defects in neurogenesis, promoting differentiation of mutant neurons won't be a viable option. It is unclear what is the main purpose of this system. It is unlikely that Leigh mutations in vivo are blocking the ability of neural progenitors to commit to a neuronal fate.

We understand the reviewer's opinion. In fact, the prevalent view of the pathomechanisms of Leigh is that the disease develops as early-onset neurodegeneration, through which neurons die due to excessive production of free radicals because of ineffective OXPHOS². However, antioxidant therapies failed to show clinical improvements in Leigh patients^{3,4}, possibly because they may blunt physiological compensatory responses within the affected cells⁵.

At the same time, it is known that children affected by Leigh typically die in the first years of life and exhibit neurodevelopmental delay, mental retardation, and failure to reach developmental milestones even prior to the onset of regressive phenotypes.⁶⁻⁸ Antenatal manifestations are also observed in patients with Leigh or other mitochondrial diseases.^{9,10} Increasing evidence indeed suggest that mitochondria can play a role in neurodevelopment by promoting neurogenesis,^{11,12} indicating that mitochondrial defects might interfere with this physiological process.^{13,14} The neurodevelopmental component of the disease is supported by data obtained in preclinical models. Defective embryonic development occurs in Leigh mice with *Ndufs4* KO,¹⁵ with specific defects in neural stem and progenitor cell proliferation causing significant delays in neurogenesis and gliogenesis.¹⁶ Organoid models of Leigh also showed impaired cellular organization and defective corticogenesis.^{17,18} Therefore, we kindly disagree with the reviewer that Leigh mutations cannot lead to impairment of neural progenitors *in vivo*. We simply do not have sufficient human brain data to come to a definite conclusion on this matter.

In fact, we believe that it is possible to reason that innovative treatment strategies could come by counteracting the neurodevelopmental failure induced by Leigh genetic defects. Using phenotypes related to neurodevelopmental, we were able to identify PDE5 inhibitors as potential drugs for Leigh patients carrying *MT-ATP6* variants.¹⁹ Following these results, we carried out a large screening in patient-derived NPCs with *MT-ATP6* variants which confirmed the potential of the PDE5 inhibitor sildenafil.²⁰ Sildenafil not only improved neuronal morphogenesis in iPSC-derived neurons but also extended the life span of Leigh mice.²⁰ Moreover, Sildenafil led to beneficial effects in compassionate off-labels studies in Leigh individuals carrying *MT-ATP6* variants.²⁰ For Sildenafil use in Leigh, we have already obtained an Orphan Drug Designation (ODD) (EU/3/23/2831) and are currently planning a placebo-controlled interventional trial in Leigh patients carrying *MT-ATP6* variants. These studies demonstrate that focusing on neural progenitor function could lead to identifying innovative interventions that may have a clear impact on patient's lives.

At the same time, we realized that our original version was not sufficiently clear in explaining this rationale. We extensively rewrote the text and have included a section in the introduction and discussion where we discuss this aspect and provide more details with respect to limitations and potentials of our approach in the context of drug discovery of Leigh and other mitochondrial diseases in the introduction (**lines 125-141**) and in the discussion (**lines 601-614**).

2. Overall, there is a lack of description of the results. For example, the authors mention "significant defects occurring five days after NGN2 induction in Leigh iNS" but do not mention what these defects are and do not place these results in the context of the disease. Same is true for their deep-learning drug discovery pipeline. Of the drugs selected it is not clear which ones belong to the DETF1 or DETF2 category, and in a few lines narrowed down to 2 drugs increasing neuronal number with unclear mechanistic insight into the effects of these drugs on the transcription profile of these cells.

We apologize for this lack of clarity. We extensively rewrote the text and restructured the figures to provide more details and explanations. We included explanations regarding the defects in Leigh iNs (**Figure 2c-f, lines 230-255**), the deep learning pipelines (**Figure S1b-f, lines 160-218**), and which selected drugs belong to which category (**lines 216-218**).

3. The investigators used previously generated scRNAseq datasets of isogenic cortical organoids with LS mutations to determine the key affected populations and to develop the deep learning strategy, not data of midbrain organoids. While the populations affected may be equivalent, validation of this assumption is imperative.

To address this important limitation, we performed scRNAseq of midbrain organoids (**Figure 3a-f, Figure S4a-d**). We then carried out pseudotime analysis in a similar manner as it was done for cerebral organoids. The results suggested similar defects in the progression between radial glia to progenitors and neurons (**Figure 3c-d, Figure S4d**). Therefore, we could confirm: i) neuromorphogenesis defects can be seen in different types of brain organoids carrying the same Leigh variants, ii) midbrain organoids may be used as a Leigh model system for downstream analyses and validations of putative treatments.

4. There several inconsistencies of the data presented throughout the paper. This generated concerns about the variability between different batches of organoids. For example, in Figure 1i the sample derived from LS completely lacks TH staining, but in Figure 3d the results are less dramatic.

We thank the reviewer for pointing this out. We have repeated several experiments to increase robustness and have indicated the number of replicates in all figure legends. It is true that individual staining images may be misleading. For this reason, we have included quantification analyses not only of TH but also of SMI312 and MAP2 staining both for Leigh brain organoids vs control organoids (**Figure 3h**) and for Leigh organoids treated with the two azoles (**Figure 4d and Figure S7a-b**). To better illustrate the quantifications, we replaced the original TH staining images with different ones (**Figure 3g, Figure 4d**).

5. It is not clear how in Fig. S3d treatment with FCCP would have recapitulated increased motility observed in Leigh model, even from a theoretical standpoint. This would induce QC pathways that would lead to motor adaptors being targeted, thus decreased motility. Additional rationale and explanation for this conclusion is needed.

We apologize for the lack of clarity. FCCP was indeed used not for the purpose of replicating increased motility seen in Leigh neurons but rather to illustrate that conditions in which mitochondria are disrupted cause an expected reduction in motility. The FCCP control was thus added to prove that our method was capable of detecting a decrease in mitochondrial motility. We think that adding such control is important, given that we found an unexpected increase in motility in Leigh neurons. By comparing the Leigh data to the FCCP data (**Figure S6g**), we could conclude that the increase in motility seen in Leigh neurons was not due to simple mitochondrial impairment (which would have caused decreased motility) but rather to other disease-related mechanisms. We have modified the text to clarify this point (**lines 367-373**).

6. In the data shown in Figures 3g, 3h and S3e, there appears to be differential effects between the DA neurons and organoids in how these motor/anchor genes are impacted. Would be good to see if treating the DA neurons alone results in any rescue. Switching from single cell RNAseq data from cerebral organoids to batch data like a western blot of midbrain organoids doesn't

give the appropriate resolution when seeing if the drugs rescue the phenotype in specific cell types. Figure 3e shows how an increase MAP2 with Sertaconazole and increase in SMI312 with Talarozole. The co-treatment could be informative. Figure 3d is clearly from the same staining as 3e, but not in the same panel.

We thank the reviewer for these suggestions. We now show anchoring proteins immunoblots for midbrain organoids (**Figure S6k-l**) and for mDANs (**Figure S6i-j**). In both contexts, the two azoles did not seem to have consistent effects. Therefore, we do not believe that their mechanism of action may be related to modulation of mitochondrial motility. We have modified the text to comment on these findings (**lines 362-373**).

Regarding the staining presentation, we now show only TH in the main figure, as it was the only one with significant differences based on quantification (**Figure 4d**). The other stainings and related quantifications are in **Figure S7**.

With respect to co-treatments, we have addressed whether the combination of talarozole and sertaconazole could increase the engagement on CYP26 or on PPAR γ . We found that the two drugs do not seem to have synergistic activity on these receptors' expression modulation (**Figure 6h**). We included text in the discussion to mention that future studies should focus on additional interaction and synergies to understand whether the compounds could be used in combination to enhance their effectiveness (**lines 583-588**).

7. Full characterization of the midbrain organoids is needed. Additional midbrain dopaminergic neuronal markers.

To address this aspect, we performed scRNAseq of midbrain organoids from Leigh organoids and compared them to control organoids. We analyzed the population distribution of midbrain organoids and identified 7 clusters (**Figure S4a**). For these clusters, we provide their marker identity (**Figure S4a, Table S2**) and identified cluster 7 corresponding to intermediate progenitors (IPs) and clusters 3 corresponding to neurons (**Figure 3e**). We also included additional identification of known dopaminergic markers and found that they mainly marked cluster 3 (neurons), cluster 7 (IPs) and cluster 1 (which corresponded to aberrant neurons that were only seen in Leigh midbrain organoids) (**Figure S4c**) (**lines 282-293**).

8. Issues with rigor: only 2 independent experiments were used in some cases. Some organoids used in different panels and variability in stainings are worrisome. There is no clear information of how many organoids and how many independent differentiations were used for analysis.

We have repeated several experiments to increase robustness and have indicated the number of replicates and number of used organoids in all figure legends. To address the variability in stainings we included quantification analyses not only of TH but also of SMI312 and MAP2, both for Leigh brain organoids vs control organoids (**Figure 3h**) and for Leigh organoids treated with the two azoles (**Figure 4d, Figure S7a-b**).

9. Limited discussion of the implications of the data. What are the mechanisms for increased neurogenesis? Is the azole compound main mechanism to reduce cell death? Lactate is also released by dying cells, but cell death was not measured in this study.

We agree with the reviewer that our original submission lacked detailed analyses and discussion. We have extensively reconfigured the figures and text and added several details and interpretations.

As mentioned above, our interpretation of the disease is not based on the assumption that neurons are perfectly fine and then degenerating and dying, but rather that there may be an impairment in the process of neuronal generation and morphogenesis, given the importance of neuronal branching and connectivity for physiological brain development.^{21,22} Indeed mitochondrial bioenergetics is crucial for axonal arborization and synaptic function^{23,24}. A failure to establish the orchestrated developmental events responsible for proper neuronal wiring and guidance might affect synaptogenesis and neuronal circuit formation²⁵. These defects could underpin the cognitive and developmental impairment of Leigh patients and their susceptibility to metabolic disturbances that ultimately lead to neuronal cell dysfunction and death.⁶ For this reason, we did not measure cell death, as also this mechanism was not highlighted by the current scRNAseq analysis or by our previous omics datasets in cerebral organoids and DA neurons.¹⁷

With respect to the reduction in lactate release in Leigh organoids treated with the azole compounds (**Figure 4c**), we interpret this as a part of a reconfiguration of energy metabolism (**Figure S8a-d**). We included additional text and explanation to address these important aspects and highlight limitations of the current work of future potential lines of research in the results (**lines 326-331**) and in the discussion (**lines 589-614**).

10. To have a clear idea of the effects of treatments scRNAseq after treatments would be recommended.

As suggested by the reviewer, we performed scRNAseq of Leigh midbrain organoids and compared them to control midbrain organoids (**Figure 3a-f, Figure S4a-d, Figure S5a-d**) and scRNAseq of Leigh midbrain organoids treated with talarozole or sertaconazole (**Figure 4h-k, Figure S7c-d**). We analyzed the effects of the treatments on the different populations with midbrain organoids and identified important differences in the neuronal population and particularly related to mitochondrial function, neuronal development, and lipid metabolism (**Figure 4h-k, Figure S7c-d**).

11. Assessing lipid metabolism via interrogating cholesterol is a limited, narrow analysis. Lipidomics coupled with changes in expression/activity of key enzymes in identified affected pathways is key.

As suggested by the reviewer, we performed lipidomics in control neural cells and in Leigh neural cells treated with DMSO or with sertaconazole or talarozole (**Figure 5d-f, Figure S7e-g, Table S5**). These data nicely confirmed the effects seen by scRNAseq in the neuronal population of Leigh organoids (**Figure 4h-k**). The lipidomics results provided important validations to the potential effects of the azole compounds on membrane-associated lipids, which we saw in the assays related to membrane-bound cholesterol (**Figure 5g-j**).

12. Talarozole targets CYP26A1/B1. But then the authors state, after the modeling, that “both azoles could bind” the target protein based on mRNA expression across models, including DA neurons and iNs. However, no genetic manipulations of the system were included.

To address this aspect, we carried out extensive docking analysis on all the different azoles using two complementary approaches in order to assess their engagement on activity of CYP26 (**Figure 6a-e, Figure S9, Figure S10a**) and PPAR γ (**Figure 6f-g, Figure S10b**). We validated this engagement using luciferase-based expression of CYP26 and PPAR γ in mutant neural cells (**Figure 6h**). These results indicate that among all azoles, talarozole appears to be the one most capable of eliciting an activity of both pathways, which fits with our experimental data. Sertaconazole instead had less effect on these pathways but had a bigger impact on modulation of lipid metabolism (**Figure 5d-j**). We included additional interpretations in the discussion (**lines 534-588**). We also restructured the whole manuscript and all the figures to present the findings in a clearer and more linear manner and to include more details and interpretations.

13. The use of scRNAseq to determine OCR, ATP, and glucose uptake needs to be validated. The publications cited as proof-of-principle did not use scRNAseq data to infer metabolic activity. Differential gene expression analysis provides the basis for testing potential hypotheses but not direct evidence of changes in metabolic activity.

We agree with the reviewer that these data were too preliminary and lacked validation and therefore decided to remove them from the revised manuscript.

Minor

- Recommendation: change brain organoid to neural organoid

We understand the reviewer's opinion. However, we do not believe that changing "brain organoid" into "neural organoid" would enhance the clarity of the message. Organoids are constructed to mimic a certain organ and named in a define manner based on the organs that they attempt to model (e.g. kidney organoids, lung organoids, liver organoids, gut organoids and so on). The organoids that we used here represent a model of the brain (and midbrain region in particular) and not of the overall nervous system, as the term "neural" might imply. In fact, by looking at the literature, "brain organoids" is the term commonly used in recent high-profile publications.²⁶⁻²⁸

- Referring to "our" MO should be changed to MO from LS or MO generated in this study

We thank the reviewer for pointing this out. We corrected this (**Figure S3b-c**).

- Scale bars are sometimes different between

We thank the reviewer for noticing this. We corrected all scale bars.

Reviewer #3 (Remarks to the Author):

The authors describe the development of a deep learning algorithm to predict drug response

from non-cancer perturbation and gene expression in order to validate these in their experimental models of Leigh syndrome. To my knowledge, this is a rare application of this predictive modeling in neurodegeneration and therefore is of significant importance to the community. In particular, because more traditional approaches to identify actionable targets in neurodegeneration have resulted - at best - in mixed clinical results.

As highlighted in the title, a key novelty of this work is applying the objective, data-led DL prediction of drugs in Leigh Syndrome.

The authors conclude that the algorithm predicted 27 (or 28 - conflicting numbers are reported) chemical compounds, 9 of which were experimentally tested.

Such a method, if it genuinely enriches the hit rate to identify active drugs, would be of tremendous value to the community. Which is why it is surprising that the authors do not provide any of the key metrics required to evaluate the benefit or added value of the DL method over standard bioinformatics approaches.

Following this reviewer's comment, we have included performance benchmarking results. Please see below for details.

1. Performance metrics

A manuscript pivoting on using DL to make predictions must provide performance or metrics of the algorithm (F1, AUC, precision/recall etc). None are provided. What evidence is there that the DL method was useful in recommending drugs that exceeds what can be achieved using standard approaches of assessing pathway enrichment in differentially expressed genes without the application of DL?

Following this comment, performance metrics for both the protein-target and drug predictions are now provided in the revised Methods of the revised manuscript (**lines 1273-1481**) together with a corresponding figure panel (**Figure S1b-f**). Specifically, our ensemble model achieved a mean F1 score, false positive rate (FPR) and false negative rate (FNR) of 0.700, 0.129 and 0.305, respectively, on the validation set. On an independent test set, which included unseen transcriptomic profiles related to cellular conversion, the model achieved a mean F1 score, FPR and FNR of 0.528, 0.167, and 0.472, respectively. To evaluate the utility of the DL-based framework for drug prediction, we compared early enrichment of true positives against GSEA-based methods and random ranking. Within the top 1st percentile of ranked drugs, our framework recalled 9.2% of known hits, outperforming GSEA (best case: 2.1%) and random ranking (1%). These results demonstrate that the DL framework adds value beyond conventional pathway enrichment by enabling more accurate early prioritization of candidate drugs for experimental validation. We thank the reviewer for this important and constructive comment.

2. Neural network architecture

The authors use a simple feed forward neural network with three hidden layers. This simplicity of the NN is sensible given the size of the data set. The general approach described is a standard one, but appropriate for the question being asked.

The reviewer is indeed right about questioning the original model architecture. Based on further model development, we have updated the neural network architecture and the drug prioritization strategy in the revised manuscript. In the updated framework, we introduce a variational autoencoder (VAE) to denoise and regularize graph-derived features, followed by an ensemble of three independently trained, 5-hidden-layer feedforward neural networks. This ensemble design improved overall model performance while still maintaining architectural simplicity. In addition, the prior version of our tool relied on pathway enrichment for drug prioritization. In the revised approach, we now compute a Bayesian enrichment score (BES) that evaluates statistical overlap between predicted target proteins and known drug targets. This new scoring method integrates multiple components, hypergeometric p-value, normalized ranking, and empirical p-value robustness, to offer a more flexible and statistically grounded prioritization of drugs. Unlike pathway-based methods, BES does not depend on curated pathway boundaries, making it better suited to handle incomplete pathway annotations, particularly in non-cancer contexts where annotation coverage is limited. Details of the updated framework are now included in the revised results (**lines 160-218**) and methods section (**lines 1273-1481**) together with a graphical summary in **Figure S1b**.

3. Training and validation

The authors state that they use iterative cross validation. However, the authors do not provide any information on the segmentation of the data, the cross validation results etc. What are the results of the final cross validation? How predictive is the algorithm? What are the false positive and false negative rates?

We appreciate the reviewer's comment. As partially answered in the first comment by this reviewer above, we used a 1-fold validation strategy with 10% of the protein-level training samples randomly held out for validation. This approach was selected due to the large and regularized feature space, and because we conducted extensive hyperparameter tuning to identify the best-performing models for subsequent ensembling. Specifically, we tested different combinations of batch size, learning rate, dropout rate, number of hidden layers, initial hidden layer size, L1 and L2 regularization coefficients, and weight decay. Performing full 5-fold cross-validation across all hyperparameter configurations would have been computationally infeasible. The final ensemble model achieved an F1 score of 0.700, a FPR of 0.129, and a FNR of 0.305 on the validation set. These performance metrics are now clearly reported in the revised Methods section (**lines 1448-1481**) and illustrated in the new figure panel (**Figure S1b-f**).

3. Independent test set

A major problem faced by the community in these approaches is overfitting: optimizing an algorithm on the training data where it cannot perform well outside these specific data. For this reason, an independent hold out test set, unseen by the algorithm during training and optimization is used to assess generalizability. Did the authors keep a hold out independent test set? If so, how did the algorithm perform? If not, how was algorithm generalizability assessed?

We thank the reviewer for pointing this out. As also partially answered in the first comment by this reviewer above, we evaluated the framework using a strictly independent hold-out 47 test datasets associated with diverse cell conversion processes, including differentiation, metabolic shifts, and reprogramming. All gene expression profiles in this test set were entirely excluded from the training and validation phases, ensuring unbiased assessment. On this hold-out test set, the model achieved a mean F1 score of 0.528, with a mean FPR of 0.167 and a mean FNR of 0.472 for protein target prediction. For drug prioritization, given the lack of well-defined positive and negative drug sets, we focused on early recall of true positives within top percentiles, as this is a clinically relevant metric for downstream experimental validation. Based on the Bayesian enrichment score (BES), the framework achieved a 9.2% average recall of true positives within the top 1st percentile of ranked candidates, outperforming gene set enrichment analysis (GSEA)-based approaches, which achieved a 2.1% mean recall in the best case, and random ranking (1%). In the GSEA-based baseline, enriched pathways were identified from input differentially expressed genes (DEGs) using Gene Ontology Biological Process (GOBP), Reactome, and KEGG via enrichR,²⁹ applying adjusted p-value cutoffs of 0.05, 0.001, and 0.005 to each database. DEGs within enriched pathways were then used as predicted target proteins, which were subsequently used to rank drugs based on a BES. The framework achieved 16.3% in the top 2nd percentile, while the best GSEA-based approach was 6.1% and far exceeding the ~2% obtained from random ranking. These results show our framework effectively enriches for biologically relevant drugs. These results are now described in detail in the “DL-based framework test performance” section of Methods in the revised manuscript (**lines 1448-1481**) together with a figure panel (**Figure S1e**). The 47 test datasets are now listed in **Table S1**.

4. Selection of drugs for testing

The authors state that the algorithm predicted 27/28 compounds, of which 9 were selected for testing. Did the algorithm ONLY predict 27 compounds? What values were used to define a 'prediction'?

We thank the reviewer for highlighting an important aspect. We indeed realized that the provided numbers were not correct. In the revised manuscript, we added a description of the algorithm that outputs a ranked list of candidate drugs, based on a Bayesian enrichment score (BES). A “prediction” in this context refers to any drugs ranked by the framework, with higher BES values indicating stronger support. In this particular application, to prioritize drugs for experimental testing, we selected drugs within the top 2.5th percentile of the ranked list (143 drugs), reflecting a balance between stringency and biological variability. From this set, we further filtered for drugs with known metabolic functions, narrowing the list to 20 candidates. Finally, 4 drugs were selected for testing based on their safety profiles. The updated numbers and prioritization strategy are described in the results section of the revised manuscript (**lines 203-218**).

Importantly, what was the null hypothesis testing to demonstrate the value? How were the drugs predicted by DL these superior to what would have been obtained by simple annotation of differentially expressed pathways?

In the context of the Leigh CO scRNA-seq application, standard annotation-based approaches failed to identify talarozole. Specifically, the differentially expressed genes (DEGs) did not include any known talarozole target proteins, placing it at the bottom of the ranking. Similarly, querying the CMAP database^{30,31} using gene expression fold changes as input did not yield talarozole among the predicted candidates. This demonstrates that the DL framework can identify mechanistically relevant drug candidates that would have been missed by conventional enrichment or similarity-based methods, highlighting its added value in complex disease contexts. We added more text and explanations in the discussion to highlight the DL design and relevance (**lines 497-528**).

As it stands, it is unclear that the DL algorithm provided any value to the outcome of this study.

We appreciate the reviewer's constructive concerns and hope that our point-by-point responses above clarify the role and added value of the DL-based framework in this study. We also greatly thank the reviewer for giving us relevant comments that resulted in the substantial improvement of the manuscript.

Reviewer #3 (Remarks on code availability):

The authors do not provide information on code availability.

I was unable to find the specific Python function developed by the authors in the zip download file.

I also found the manuscript on BioRxiv and searched for the code there but with no avail.

We apologize for this oversight. The latest program code and input files are available at: (https://github.com/temporary-code-2025/manuscript_code/tree/main). The password for opening the zip file containing code files and documentation is: Manuscript2025Access! Due to the size limitation by Github, the FNN model weight files are separately uploaded, and some other larger files could not be uploaded. Nevertheless, all program code files are included in the zip file.

References related to the point-by-point rebuttal:

1. Puighermanal, E., Luna-Sanchez, M., Gella, A., van der Walt, G., Urpi, A., Royo, M., Tena-Morraja, P., Appiah, I., de Donato, M.H., Menardy, F., Bianchi, P., Esteve-Codina, A., Rodriguez-Pascau, L., Vergara, C., Gomez-Pallares, M., Marsicano, G., Bellocchio, L., Martinell, M., Sanz, E., Jurado, S., Soriano, F.X., Pizcueta, P., and Quintana, A. (2024). Cannabidiol ameliorates mitochondrial disease via PPARgamma activation in preclinical models. *Nat Commun* 15, 7730. 10.1038/s41467-024-51884-8.
2. Baertling, F., Rodenburg, R.J., Schaper, J., Smeitink, J.A., Koopman, W.J., Mayatepek, E., Morava, E., and Distelmaier, F. (2014). A guide to diagnosis and treatment of Leigh syndrome. *J Neurol Neurosurg Psychiatry* 85, 257-265. 10.1136/jnnp-2012-304426.
3. Weissig, V. (2020). Drug Development for the Therapy of Mitochondrial Diseases. *Trends Mol Med* 26, 40-57. 10.1016/j.molmed.2019.09.002.

4. Russell, O.M., Gorman, G.S., Lightowlers, R.N., and Turnbull, D.M. (2020). Mitochondrial Diseases: Hope for the Future. *Cell* 181, 168-188. 10.1016/j.cell.2020.02.051.
5. Dogan, S.A., Cerutti, R., Beninca, C., Brea-Calvo, G., Jacobs, H.T., Zeviani, M., Szibor, M., and Viscomi, C. (2018). Perturbed Redox Signaling Exacerbates a Mitochondrial Myopathy. *Cell Metab* 28, 764-775 e765. 10.1016/j.cmet.2018.07.012.
6. Falk, M.J. (2010). Neurodevelopmental manifestations of mitochondrial disease. *J Dev Behav Pediatr* 31, 610-621. 10.1097/DBP.0b013e3181ef42c1.
7. Tinker, R.J., Falk, M.J., Goldstein, A., George-Sankoh, I., Xiao, R., Adang, L., and Ganetzky, R. (2022). Early developmental delay in Leigh syndrome spectrum disorders is associated with poor clinical prognosis. *Mol Genet Metab* 135, 342-349. 10.1016/j.ymgme.2022.02.006.
8. Zilber, S., Woleben, K., Johnson, S.C., de Souza, C.F.M., Boyce, D., Friert, K., Boggs, C., Messahel, S., Burnworth, M.J., Afolabi, T.M., and Kayani, S. (2023). Leigh syndrome global patient registry: uniting patients and researchers worldwide. *Orphanet J Rare Dis* 18, 264. 10.1186/s13023-023-02886-0.
9. von Kleist-Retzow, J.C., Cormier-Daire, V., Viot, G., Goldenberg, A., Mardach, B., Amiel, J., Saada, P., Dumez, Y., Brunelle, F., Saudubray, J.M., Chretien, D., Rotig, A., Rustin, P., Munnich, A., and De Lonlay, P. (2003). Antenatal manifestations of mitochondrial respiratory chain deficiency. *J Pediatr* 143, 208-212. 10.1067/S0022-3476(03)00130-6.
10. Adelizzi, A., Giri, A., Di Donfrancesco, A., Boito, S., Prigione, A., Bottani, E., Bollati, V., Tiranti, V., Persico, N., and Brunetti, D. (2024). Fetal and obstetrics manifestations of mitochondrial diseases. *J Transl Med* 22, 853. 10.1186/s12967-024-05633-6.
11. Iwata, R., Casimir, P., Erkol, E., Boubakar, L., Planque, M., Gallego Lopez, I.M., Ditekowska, M., Gaspariunaite, V., Beckers, S., Remans, D., Vints, K., Vandekerere, A., Poovathingal, S., Bird, M., Vlaeminck, I., Creemers, E., Wierda, K., Corthout, N., Vermeersch, P., Carpentier, S., Davie, K., Mazzone, M., Gounko, N.V., Aerts, S., Ghesquiere, B., Fendt, S.M., and Vanderhaeghen, P. (2023). Mitochondria metabolism sets the species-specific tempo of neuronal development. *Science* 379, eabn4705. 10.1126/science.abn4705.
12. Iwata, R., Casimir, P., and Vanderhaeghen, P. (2020). Mitochondrial dynamics in postmitotic cells regulate neurogenesis. *Science* 369, 858-862. 10.1126/science.aba9760.
13. Khacho, M., Harris, R., and Slack, R.S. (2019). Mitochondria as central regulators of neural stem cell fate and cognitive function. *Nat Rev Neurosci* 20, 34-48. 10.1038/s41583-018-0091-3.
14. Brunetti, D., Dykstra, W., Le, S., Zink, A., and Prigione, A. (2021). Mitochondria in neurogenesis: Implications for mitochondrial diseases. *Stem Cells*. 10.1002/stem.3425.
15. Wang, M., Huang, Y.P., Wu, H., Song, K., Wan, C., Chi, A.N., Xiao, Y.M., and Zhao, X.Y. (2017). Mitochondrial complex I deficiency leads to the retardation of early embryonic development in Ndufs4 knockout mice. *PeerJ* 5, e3339. 10.7717/peerj.3339.
16. Biswas, S.R., Tomsick, P.L., Kelly, C., Lester, B.A., Milner, J.P., Henry, S.N., Soto, Y., Brindley, S., DeFoor, N., Morton, P.D., and Pickrell, A.M. (2025). Impaired Complex I dysregulates neural/glial precursors and corpus callosum development revealing postnatal defects in Leigh Syndrome mice. *bioRxiv*, 2025.2005.2016.654318. 10.1101/2025.05.16.654318.
17. Inak, G., Rybak-Wolf, A., Lisowski, P., Pentimalli, T.M., Juttner, R., Glazar, P., Uppal, K., Bottani, E., Brunetti, D., Secker, C., Zink, A., Meierhofer, D., Henke, M.T., Dey, M., Ciptasari, U., Mlody, B., Hahn, T., Berruezo-Llacuna, M., Karaikos, N., Di Virgilio, M., Mayr, J.A., Wortmann, S.B., Priller, J., Gotthardt, M., Jones, D.P., Mayatepek, E., Stenzel, W., Diecke, S., Kuhn, R., Wanker, E.E., Rajewsky, N., Schuelke, M., and Prigione, A. (2021). Defective metabolic programming impairs early neuronal

- morphogenesis in neural cultures and an organoid model of Leigh syndrome. *Nat Commun* 12, 1929. [10.1038/s41467-021-22117-z](https://doi.org/10.1038/s41467-021-22117-z).
18. Romero-Morales, A.I., Robertson, G.L., Rastogi, A., Rasmussen, M.L., Temuri, H., McElroy, G.S., Chakrabarty, R.P., Hsu, L., Almonacid, P.M., Millis, B.A., Chandel, N.S., Cartailier, J.P., and Gama, V. (2022). Human iPSC-derived cerebral organoids model features of Leigh syndrome and reveal abnormal corticogenesis. *Development* 149, 10.1242/dev.199914.
 19. Lorenz, C., Lesimple, P., Bukowiecki, R., Zink, A., Inak, G., Mlody, B., Singh, M., Semtner, M., Mah, N., Aure, K., Leong, M., Zabiegajlov, O., Lyras, E.M., Pfiffer, V., Fauler, B., Eichhorst, J., Wiesner, B., Huebner, N., Priller, J., Mielke, T., Meierhofer, D., Izsvak, Z., Meier, J.C., Bouillaud, F., Adjaye, J., Schuelke, M., Wanker, E.E., Lombes, A., and Prigione, A. (2017). Human iPSC-Derived Neural Progenitors Are an Effective Drug Discovery Model for Neurological mtDNA Disorders. *Cell Stem Cell* 20, 659-674 e659. [10.1016/j.stem.2016.12.013](https://doi.org/10.1016/j.stem.2016.12.013).
 20. Zink, A., Dai, D.-F., Wittich, A., Henke, M.-T., Pedrotti, G., Heiduschka, S., Aguilar, G.S., Pentimalli, T.M., Brueser, C., Notopoulou, S., Zhaivoron, A., Umar, A.R., Petersilie, L., Jerred, C., Bergmans, J., Schumacher, F., Keller-Findeisen, J., Rybak-Wolf, A., Stach, D., Reinshagen, J., Zaliani, A., Haferkamp, U., Euro, L., Di Donfrancesco, A., Santanatoglia, C., Cappelozza, E., Cubero, M.S., Pavez-Giani, M., Bakumenko, O., Meierhofer, D., Foley, A., Morales-Gonzalez, S., Tolle, I., Herebian, D., Bonesso, D., Szabo, I., Cecchetto, G., Wong, S., Moresco, M., Maresca, A., Decimo, I., Adjobo-Hermans, M.J.W., Kleuser, B., Cyganek, L., Mühlhausen, C., Schlotawa, L., Tiranti, V., Rossi, A., Mayatepek, E., La Morgia, C., Carelli, V., Klopstock, T., Distelmaier, F., Ullah, G., Jakobs, S., Rajewsky, N., Rose, C.R., Petrakis, S., Edenhofer, F., Koopmann, W., Brunetti, D., Lisowski, P., Suomalainen, A., del Sol, A., Bottani, E., Pless, O., Schuelke, M., and Prigione, A. (2025). Pluripotent stem cell-based drug discovery uncovers sildenafil as a treatment for mitochondrial disease. *medRxiv*, 2025.2005.2015.25325571. [10.1101/2025.05.15.25325571](https://doi.org/10.1101/2025.05.15.25325571).
 21. Chedotal, A., and Richards, L.J. (2010). Wiring the brain: the biology of neuronal guidance. *Cold Spring Harb Perspect Biol* 2, a001917. [10.1101/cshperspect.a001917](https://doi.org/10.1101/cshperspect.a001917).
 22. Baumann, N., Wagener, R.J., Javed, A., Conti, E., Abe, P., Lopes, A., Sansevrino, R., Lavalley, A., Magrinelli, E., Szalai, T., Fuciec, D., Ferreira, C., Fievre, S., Fouassier, A., D'Amico, D., Harschnitz, O., and Jabaudon, D. (2025). Regional differences in progenitor metabolism shape brain growth during development. *Cell* 188, 3567-3582 e3520. [10.1016/j.cell.2025.04.003](https://doi.org/10.1016/j.cell.2025.04.003).
 23. Rangaraju, V., Calloway, N., and Ryan, T.A. (2014). Activity-driven local ATP synthesis is required for synaptic function. *Cell* 156, 825-835. [10.1016/j.cell.2013.12.042](https://doi.org/10.1016/j.cell.2013.12.042).
 24. Vaarmann, A., Mandel, M., Zeb, A., Wareski, P., Liiv, J., Kuum, M., Antsov, E., Liiv, M., Cagalinec, M., Choubey, V., and Kaasik, A. (2016). Mitochondrial biogenesis is required for axonal growth. *Development* 143, 1981-1992. [10.1242/dev.128926](https://doi.org/10.1242/dev.128926).
 25. Kolodkin, A.L., and Tessier-Lavigne, M. (2011). Mechanisms and molecules of neuronal wiring: a primer. *Cold Spring Harb Perspect Biol* 3, [10.1101/cshperspect.a001727](https://doi.org/10.1101/cshperspect.a001727).
 26. Li, C., Fleck, J.S., Martins-Costa, C., Burkard, T.R., Themann, J., Stuempflen, M., Peer, A.M., Vertesy, A., Littleboy, J.B., Esk, C., Elling, U., Kasprian, G., Corsini, N.S., Treutlein, B., and Knoblich, J.A. (2023). Single-cell brain organoid screening identifies developmental defects in autism. *Nature* 621, 373-380. [10.1038/s41586-023-06473-y](https://doi.org/10.1038/s41586-023-06473-y).
 27. Jain, A., Gut, G., Sanchis-Calleja, F., Tschannen, R., He, Z., Luginbuhl, N., Zenk, F., Chrisnandy, A., Streib, S., Harmel, C., Okamoto, R., Santel, M., Seimiya, M., Holtackers, R., Rohland, J.K., Jansen, S.M.J., Lutolf, M.P., Camp, J.G., and Treutlein, B. (2025). Morphodynamics of human early brain organoid development. *Nature*. [10.1038/s41586-025-09151-3](https://doi.org/10.1038/s41586-025-09151-3).
 28. Anton-Bolanos, N., Faravelli, I., Faits, T., Andreadis, S., Kastli, R., Trattaro, S., Adiconis, X., Wei, A., Sampath Kumar, A., Di Bella, D.J., Tegtmeier, M., Nehme, R., Levin, J.Z., Regev, A., and Arlotta, P. (2024). Brain Chimeroids reveal individual

- susceptibility to neurotoxic triggers. *Nature* 631, 142-149. 10.1038/s41586-024-07578-8.
29. Kuleshov, M.V., Jones, M.R., Rouillard, A.D., Fernandez, N.F., Duan, Q., Wang, Z., Koplev, S., Jenkins, S.L., Jagodnik, K.M., Lachmann, A., McDermott, M.G., Monteiro, C.D., Gundersen, G.W., and Ma'ayan, A. (2016). Enrichr: a comprehensive gene set enrichment analysis web server 2016 update. *Nucleic Acids Res* 44, W90-97. 10.1093/nar/gkw377.
 30. Lamb, J., Crawford, E.D., Peck, D., Modell, J.W., Blat, I.C., Wrobel, M.J., Lerner, J., Brunet, J.P., Subramanian, A., Ross, K.N., Reich, M., Hieronymus, H., Wei, G., Armstrong, S.A., Haggarty, S.J., Clemons, P.A., Wei, R., Carr, S.A., Lander, E.S., and Golub, T.R. (2006). The Connectivity Map: using gene-expression signatures to connect small molecules, genes, and disease. *Science* 313, 1929-1935. 10.1126/science.1132939.
 31. Subramanian, A., Narayan, R., Corsello, S.M., Peck, D.D., Natoli, T.E., Lu, X., Gould, J., Davis, J.F., Tubelli, A.A., Asiedu, J.K., Lahr, D.L., Hirschman, J.E., Liu, Z., Donahue, M., Julian, B., Khan, M., Wadden, D., Smith, I.C., Lam, D., Liberzon, A., Toder, C., Bagul, M., Orzechowski, M., Enache, O.M., Piccioni, F., Johnson, S.A., Lyons, N.J., Berger, A.H., Shamji, A.F., Brooks, A.N., Vrcic, A., Flynn, C., Rosains, J., Takeda, D.Y., Hu, R., Davison, D., Lamb, J., Ardlie, K., Hogstrom, L., Greenside, P., Gray, N.S., Clemons, P.A., Silver, S., Wu, X., Zhao, W.N., Read-Button, W., Wu, X., Haggarty, S.J., Ronco, L.V., Boehm, J.S., Schreiber, S.L., Doench, J.G., Bittker, J.A., Root, D.E., Wong, B., and Golub, T.R. (2017). A Next Generation Connectivity Map: L1000 Platform and the First 1,000,000 Profiles. *Cell* 171, 1437-1452 e1417. 10.1016/j.cell.2017.10.049.

We thank the reviewers for their important suggestions and insightful comments that have led to significantly improving the quality of our work. We highlighted in red font the parts of the manuscript that we have modified. Please find below our point-by-point reply in blue font.

REVIEWER COMMENTS

Reviewer #1 (Remarks to the Author):

Authors have sufficiently and elegantly addressed my concerns, thus I have no further comments on the manuscript.

We thank the reviewer for the important feedback and suggestions.

Reviewer #2 (Remarks to the Author):

This reviewer recognizes the authors' efforts in addressing the reviewers' concerns and revising the manuscript, which presents a promising platform for drug discovery with the potential to advance understanding of rare diseases such as Leigh Syndrome. The improvements addressed some of the concerns that were raised about the clarity and rigor of the work. However, several critical issues remain that need to be addressed to meet the standards for publication in Nature Communications. The main concern is the consistent lack of control samples across multiple figures, which weakens the ability to evaluate the significance and specificity of the reported drug treatment effects.

Control samples are essential in all studies, but particularly in drug discovery research, as they provide a baseline for comparison to determine the specific effects of experimental treatments. In the current manuscript, many figures lack appropriate controls relevant to the drug discovery platform. This omission makes it difficult to interpret whether the observed effects are specifically due to the drugs acting on Leigh syndrome cells or influenced by other factors. For example: Figure 2b, Figure 2g/h (WT strains were not treated with azole compounds), Figure 4 (Control MOs were not treated with sertaconazole or talarozole), and Figure 6h (control NPCs were not treated with the drugs). Without controls in at least these critical experiments, it is unclear whether the platform consistently distinguishes actual drug effects in Leigh Syndrome samples from general effects in controls. Including control samples would allow researchers and labs interested in using this platform to quantify the magnitude of drug effects relative to baseline conditions, evaluate the specificity of drug responses, and demonstrate the robustness of the platform by showing its ability to reliably detect expected outcomes in Leigh syndrome samples and not in controls. If the drugs also enhance neuronal differentiation and

activity in control samples, this does not negate their potential for treating rare diseases and may even support the broader use of this platform in various drug discovery applications.

We thank the reviewer for taking the time to look at our data and for providing feedback. In our first revision, we followed several of the excellent suggestions from Reviewer #2, including:

- scRNAseq of midbrain organoids to dissect the impact of the mutations (**Figure 3a-f, Figure S4a-e; Figure S5a-d**).
- scRNAseq of midbrain organoids to dissect the effects of the two compounds (**Figure 4h-k, Figure S7c-d**).
- lipidomics analysis (**Figure 5a-f, Figure S7e-g**).
- experiments in dopaminergic neurons and midbrain organoids to assess the expression of anchoring proteins (**Figure S6i-l**).
- replication experiments in brain organoids to enhance the robustness of the data.
- combination treatment experiments for the two compounds (**Figure 6h**).
- additional modeling and functional validations of CYP26A1/B1 engagement (**Figure 6d-e; Figure 6h**).
- functional calcium experiments in treated midbrain organoids to replace the use of scRNAseq data to infer activity (**Figure 4e-g**).

We also agree with the reviewer that having proper controls is essential. In drug discovery applications, healthy controls are important for two reasons: 1) for providing a base line with respect to diseased samples to allow dissecting the potential effectiveness in terms of correction of disease-related defects, 2) to ensure that the compounds are not toxic.

Accordingly, in our work, we have used healthy controls for reaching these goals:

1) we identified disease phenotypes that needed to be corrected in yeasts (**Figure 2g-h**), in human brain organoids (**Figure 4b-d**), and in human neural precursor cells (**Figure 5a-j; Figure 6h**).

2) we demonstrated that the azole compounds that we have discovered (talarozole and sertaconazole) were not toxic in healthy control human brain organoids (**Figure S6a-b**).

3) all our *in silico* binding experiments were based on healthy proteins (**Figure 6a-g, Figure S9, Figure S10**). Therefore, the mechanisms of action of the two azoles were modeled in healthy scenarios and then validated in patient neural cells to dissect the specific impact on Leigh neural cells (**Figure 6h**), as our goal was to demonstrate the potential use of these compounds in this specific disease context.

The use of new approach methodologies (NAMs), such as human brain organoids, is currently considered by regulators to replace toxicity assessment in animals (Wadman, *Science* 2023). Therefore, our experiments based on healthy brain organoids for toxic assessment of compounds are in line with gold-standard approaches in drug discovery.

The reviewer asks now to assess the effect of the azole compounds also in healthy control cells. This request was not raised in the first round of revision. The reviewer mentioned that adding information in control samples could support a “broader use”. While this is possible, control cells are poor models for assessing potential treatments and no substitute for carrying out tests in other disease models, which is beyond the scope of our study.

In our work, we compared the effects of the two drugs in Leigh syndrome cells to the effects of the vehicle alone (DMSO). Based on these results, we described the potential utility of these drugs in this particular disease. Our therapeutic strategy was evaluated based on the restoration of disease-specific phenotypes that are absent in healthy cells.

Correcting a defect in patient cells and improving a feature in control cells are often two very different questions. For example, the growth deficiency in a non-fermentable growth medium seen in mutant yeast is not present in WT yeast, and so the compounds cannot “improve” the growth of WT yeast. Likewise, enhancing differentiation in the context of cells with defective differentiation is different from enhancing differentiation in control cells with normal differentiation capacity and will likely act via different mechanisms, and so would add no value to the study. We were careful not to claim that azole drugs enhance differentiation *per se*, rather that they may be helpful in the context of patient cells where there is a clear defect to be rescued.

To confirm that our approach was in line with the published literature, we looked at recent publications that used iPSCs for assessing treatments. We recently published two review articles focusing on iPSC models of mitochondrial and neurological disorders (Lickfett et al, *Stem Cells* 2025; Heiduschka et al, *Neurobiology of Disease* 2025). Among the works that we discussed in these reviews, the vast majority applied treatments only on patient cells and not on healthy control cells.

Some examples: Zheng et al, *ELIFE* 2016 tested rapamycin only in patient neurons; Gunnewiek et al, *Stem Cell Reports* 2021 tested sonlicromanol only in patient neurons with high mutation level; Winanto et al, *Cell Death Dis* 2020 tested DAPT only in patient brain organoids; Korecka et al, *Stem Cell Reports* 2019 tested LRRK2 kinase inhibitor only in patient neurons; Conforti et al, *PNAS* 2018 tested GI254023X only in patient brain organoids; Fang et al, *Nature Aging* 2021 tested sildenafil only in patient neurons, Danese et al, *Cell Reports* 2022 tested idebenone only in patient neurons; Soragni et al, *Annals of Neurology* 2014 tested

HDAC inhibitor only in patient neurons; Codazzi et al, *Human Molecular Genetics*, 2016 tested HDAC inhibitor only in patient neurons; Crombi et al, *Aging* 2017 tested nifedipine only in patient cardiomyocytes; Igoillo-Esteve et al, *JCI Insight* 2020 tested exenatide only in patient cells (pancreatic beta cells, neurons, and sensory neurons); Jing et al, *Cell Reports* 2018 tested NAD treatment only in patient hepatocyte-like cells.

We also looked at recent articles published in *Nature Communications* and *Cell Stem Cell* that used compound treatments in iPSC models. We found that most of the articles assessed the effectiveness of compound treatments only in patient cells and not in control healthy cells.

Some examples: Safari et al, *Nat Comm* 2024 tested compounds only in patient neurons; Pan et al, *Cell Stem Cell* 2025 tested compounds only in patient endothelial cells; Liu et al, *Cell Stem Cell* 2024 tested indisulam only in patient cardiomyocytes.

Lastly, we looked in the section on repurposing of marketed drugs of the Assay Guidance Manual (<https://www.ncbi.nlm.nih.gov/books/NBK53196/>). This manual provides methodological support and highlights the need for focusing on disease-relevant models in drug repurposing studies, rather than requiring testing in healthy control cells.

Altogether, we believe that we have employed appropriate controls in all our experiments:

- 1) we compared patient cells treated with a drug to patient cells treated with the corresponding vehicle.
- 2) we reported the values in healthy cells to show the extent of correction that the treatment may provide with respect to the healthy situation.
- 3) we assessed the potential toxic effects of the compounds in healthy control human brain organoids as NAMs.
- 4) we modeled the mechanisms of action and binding activity of the compounds in healthy scenarios and validated them in disease-specific human neural cells.

Our approach is in line with current literature and with articles published in the journal *Nature Communications*.

To clarify the ways in which we used healthy control samples in our work, we included a section in the discussion section (**lines 587-594**):

“Regulators are currently promoting drug discovery approaches based on non-animal models such as new approach methodologies (NAMs) to possibly reduce or eliminate the need for validations in animals for drug approval.^{84,85} In this context, complex human models such as 3D organoids are emerging as promising alternatives.⁸⁶ Indeed, in our work we used healthy human brain organoids as NAMs to ensure that the compounds did not elicit toxic effects at

the concentrations used to rescue the disease phenotypes. We concluded that the compounds were able to ameliorate Leigh phenotypes at concentrations that appeared to be non-toxic in healthy cells.”

Reviewer#3 / Internal Reviewer:

The main concern remains, as Reviewer #3 noted in the previous round, that "it is unclear whether the deep learning algorithm provided any value to the outcome of this study." I hope the authors can make these points clearer in their revision.

Thank you for your feedback and for highlighting this important aspect. As explained in the previous rebuttal letter, to evaluate the utility of the DL-based framework for drug prediction, we compared early enrichment of true positives against GSEA-based methods and random ranking. Within the top 1st percentile of ranked drugs, our framework recalled 9.2 % of known hits, outperforming GSEA (best case: 2.1 %) and random ranking (1 %). These results demonstrated that the DL framework adds value beyond conventional pathway enrichment by enabling more accurate early prioritization of candidate drugs for experimental validation. For example, talarozole was not predicted by the GSEA-based methods.

To articulate this approach, we included new text in the result section to explain that we have benchmarked our DL approach and have proved that it outperforms conventional gene expression-based approaches, thereby providing a clear added value **(lines 190-195)**:

“We compared our DL approach against gene set enrichment analysis (GSEA)-based methods and random ranking. Within the top 1st percentile of ranked drugs, our framework recalled 9.2 % of known hits, outperforming GSEA (best case: 2.1 %) and random ranking (1 %). These results demonstrated that our DL framework added value beyond conventional pathway enrichment by enabling more accurate early prioritization of candidate drugs for experimental validation.”

To elucidate the importance of results of talarozole from the DL perspective, we included the following text in the Results session **(lines 257-258)**:

“Interestingly, talarozole showed one of the highest BES values in the DL screen (BES = 9.034277; 38th out of 5,692 drugs, ~99.3rd percentile) (Figure S1h-i).”

We also included a clarification in the discussion section **(lines 515-519)**:

“Here, we developed a DL-based framework based on ChemPert, our recently developed database of perturbations in non-cancer cells. We demonstrated that this approach delivered

added value beyond conventional pathway enrichment by enabling more accurate early prioritization of candidate drugs for experimental validation.”

Lastly, we make clear in the discussion section that the use of DL enabled us to accelerate the drug discovery process for a neuronal disorder by reducing the number of potential drugs to be assessed in complex human brain organoid systems **(lines 607-612)**:

“This pipeline could represent a blueprint for rare neurodevelopmental disorders. Talarozole, our most effective candidate, had one of the highest BES values (Figure S1h-i), yet it was not recovered by GSEA-style enrichment, CMap-based signature reversal, or ChemPert-based similarity, suggesting that talarozole’s prioritization by our DL framework was method-specific and not trivially recapitulated by existing approaches.”

We hope that the added value of DL is clearer in our revised version.

1. Regarding the training and validation, please ensure that the size and diversity of the test set has been clearly introduced.

Thank you for this comment. We used 24,134 transcriptional signatures from 264 unique, diverse non-cancer cell types exposed to 2,905 unique perturbagens (chemical and biological disruptive agents) for model training and validation. This resource, originally compiled in (PMID: 36200827) represents, to our knowledge, the most comprehensive collection of non-cancer perturbational transcriptomics datasets currently available.

We have clarified this description in the Methods section **(lines 1134-1139)**:

“A multi-stage deep learning framework for predicting direct protein targets of drugs was developed using perturbation transcriptomics data (Figure S1b). Training data were retrieved from ChemPert, which provides transcriptomic profiles before and after drug treatment based on non-cancer cell types. We used 24,134 transcriptional signatures of 264 unique, diverse non-cancer cell types exposed to 2905 unique perturbagens (chemical and biological disruptive agents) for the model training (Table S7).”

We also summarized the composition of each training and validation dataset in a new table: **Table S7**.

2. For selection of drugs for testing, please briefly explain why 2.5th percentile was chosen - was this empirically determined or based on prior benchmarks? It might help to mention how

BES scores are distributed to justify percentile-based cutoff. Clarifying whether any drugs outside the top 2.5% were tested or considered would help reinforce the rigor of the selection.

We selected the top 2.5th percentile because the early-recall curve plateaus at approximately 2.5 % of the ranked list, indicating diminishing returns in recovering known actives beyond this cutoff. This is shown in **Figure S1f**:

Figure S1f: Mean drug recall at top-ranked predictions, comparing the DL-based framework against conventional gene set enrichment analysis (GSEA)-based methods using Gene Ontology Biological Processes (GOBP), Reactome, and KEGG pathway annotations at varying significance thresholds. Black dotted line indicates randomized ranking baseline.

This was the primary empirical rationale for our threshold choice. Consistent with this, the empirical cumulative distribution function (ECDF) of BES across all test datasets shows a clear inflection (“knee”) in the high-score tail at an average ECDF value of 0.975436, irrespective of the absolute BES values in each dataset. An example is presented in **Figure S1g**:

Figure S1g: Empirical cumulative distribution function (ECDF) of BES illustrating the distribution of DL framework-based drug prioritization values with the high-score tail showing a consistent inflection (“knee”) at approximately the 97.5th percentile. The dashed lines indicate the 97.5th percentile threshold, which was used to define the top 2.5 % high-confidence candidates.

Thus, it is not a single BES value that is shared across datasets, but rather the percentile location (~97.5th percentile) at which the BES distribution changes slope and at which known positives are enriched. Using a percentile-based threshold therefore provides a dataset-invariant definition of score threshold, while allowing the actual score corresponding to the 97.5th percentile to adapt to the calibration and dynamic range of each screen. This cutoff also yields a tractable number of compounds for experimental follow-up. As with any computational method, our DL framework is intended as a prioritization tool to guide experimental validation,

not as a strict classifier. We do not expect that every drug in the top 2.5 % will be experimentally confirmed, nor that all drugs below this threshold are inactive. Rather, the top 2.5 % represents a statistically and empirically supported high-confidence region in which the concentration of true positives is likely enriched, making it a reasonable and practical operating point for selecting candidates for further testing.

We included this description in the Results section (lines 217-219):

“To prioritize drug candidates from the prediction set, we selected drugs ranked within the top 2.5th percentile of the BES. This cutoff was chosen based on the test recall curve, which plateaued at 2.5 % (Figure S1f-g), indicating diminishing returns for lower-ranked drugs.”

In addition, to address the reviewer concerns regarding the rigor of our selection, we included validation experiments performed for three non-predicted drugs (NPDs) that were outside the top 2.5 % cutoff. The results are shown in Figure S2j-k:

Figure S2j-k: Number of Leigh iNs treated with DMSO or non-predicted drugs (NPDs) from the DL screen. Boxplots: mean +/- SD; n=5 biological replicates (dots). * $p < 0.001$, *** $p < 0.005$, **** $p < 0.001$, one-way ANOVA, NPD-treated Leigh iNs vs DMSO-treated Leigh iNs.

We included the following text in the Results session (lines 258-260):

“We also assessed three non-predicted drugs (NPDs) that were outside the top 2.5 % BES cutoff and found that they did not increase the number of neurons in Leigh iNs (Figure S2j-k).”

3. Inconsistency in results: In your Line 211-217, you have mentioned that "2.5 % threshold (top 142 drugs); This strategy highlighted 28 DL-predicted drugs (DLDs). Among these, we selected 5 based on their safety profile in humans." However, in your response letter, you have mentioned "143 drugs, reflecting a balance between stringency and biological variability; Narrowing the list to 20 candidates. Finally, 4 drugs were selected for testing based on their safety profiles." We are concerned about the accuracy of the reported number of selected drugs. We request that you provide the complete dataset of model predictions, including the corresponding metric scores, and clearly indicate which ones were selected.

Thank you for noticing the errors. Due to the ties (same ranking), the numbers are not always exactly the same. The correct numbers at 2.5 % threshold are 140 for DETF1 (radial glia vs neurons) and 143 for DETF2 (IPCs vs neurons).

To clarify these numbers and related prioritization strategy, we included new text in the Results section (**lines 219-224**):

“This cutoff resulted in 140 drugs for DETF1 and 143 drugs for DETF2, which we further prioritized based on annotation of metabolic function, given the central role of mitochondrial metabolism in Leigh pathophysiology and neural differentiation processes. This strategy highlighted a total of 34 DL-predicted drugs (DLDs). Among these, we selected 5 based on their safety profile in humans (DLD1, DLD3, DLD4, DLD5 from DETF1, and DLD2 from DETF2).”

We also included new text in the Methods section (**lines 1349-1357**):

“To prioritize drug candidates from the prediction set, we selected drugs ranked within the top 2.5th percentile of the BES based on the test recall curve, which plateaued at 2.5 % (Figure S1f-g). After removing duplicates, this cutoff resulted in 140 drugs for DETF1 and 143 drugs for DETF2. We then prioritized the hits based on their postulated primary target or mode of action and selected those with known modulation of metabolism according to ChEMBL (<https://www.ebi.ac.uk>) and Drug Bank (<https://go.drugbank.com/>). This strategy resulted in a total of 34 DL-predicted drugs (DLDs) (20 for DETF1 and 14 for DETF2). Among these, we selected 5 based on their safety profile in humans (DLD1, DLD3, DLD4, DLD5 from DETF1, and DLD2 from DETF2).”

In the interest of transparency, we included as **confidential data** the complete list of predicted drugs (“**predicted_drug_combined.xlsx**”). This combined list contains the 140 drugs from DETF1 and the 143 drugs from DETF2 (after removal of duplicates). In the excel file, the metabolism-related drugs (based on MedChemExpress) are highlighted in yellow. In red font, are the five drugs that we selected (4 from the DETF1 and 1 from DETF2).

However, we would prefer not to make this dataset public. The list includes additional high-scoring drugs that we intend to evaluate in future work, and positive results from these candidates may be subject to intellectual property protection. We therefore provide the complete prediction outputs to the journal editors and reviewers for their confidential evaluation. At the same time, we would like to preserve our ability to pursue patent protection for future experimentally validated candidates. We hope that the editors will understand our position and will agree with our choice.

4. Regarding the null hypothesis testing, we suggest that you clarify whether this example of talarozole is representative or anecdotal. Ideally, more than one case should be used to generalise the claim. Additional quantitative comparisons (e.g., ranking position of in each method) would strengthen the argument.

Thank you for pointing this out. In our framework, the null hypothesis is tested at the level of the component statistics that feed into the BES score, rather than on BES in isolation. Specifically, BES is constructed from two independent sources of evidence: (i) an empirical p-value that quantifies how often a given drug is predicted from randomized input signaling protein sets, and (ii) a hypergeometric p-value that tests enrichment of known drug targets within the predicted signaling modules. In this sense, null hypothesis testing is already embedded in the BES calculation, and a high BES implies that the drug is supported by statistically significant evidence from both components. For talarozole, BES is 9.034277 and it is the 38th out of 5,692 considered drugs in this disease context, placing it in the top ~0.7 % of all candidates. To quantify how extreme this is without assuming any particular (e.g. Gaussian) form for the BES distribution, we used a simple rank-based empirical tail probability. Under the null that any drug is equally likely to occupy any rank, the probability that a randomly chosen drug attains a rank as good as or better than talarozole is $38/(5,692 + 1) \approx 0.0067$. Equivalently, talarozole lies at approximately the 99.3rd percentile of the empirical BES distribution, as can be seen in **Figure S1h-i**:

Figure S1h: Histogram of BES for all 5,692 screened compounds in the DETF1 comparison. (i) Histogram of BES in the DETF1 comparison restricted to the non-zero subset. The red dashed vertical line marks the BES of talarozole (BES = 9.034277).

Even among drugs with BES > 0 (**Figure S1h**), only a very small fraction achieve scores in the range of talarozole (≥ 9), indicating that talarozole lies in the extreme right tail of the distribution.

In addition, we evaluated if talarozole could be confidently predicted by other methods using GSEA-style enrichment, CMap signature reversal (PMIDs: 17008526, 29195078), and ChemPert (PMID: 36200827). None of these methods prioritized talarozole. In fact, all of them were unable to place talarozole among their predicted drug lists. This divergence reflects their dependence on pathway over-representation or direct signature correlation, which could not readily capture talarozole drug targets. Combined with the yeast class-level evidence, this cross-method comparison indicates that talarozole's prioritization is method-specific and is methodologically robust, rather than anecdotal. Nevertheless, we agree with the reviewer that talarozole currently represents a single positive example, and therefore it should be interpreted as a proof-of-concept case rather than as a basis for broad generalization about all high-BES drugs. In the revised text, we have softened the wording accordingly to emphasize that talarozole illustrates how the DL-derived BES can prioritize a biologically plausible and experimentally confirmed candidate, while explicitly acknowledging that additional prospective validations of other high-ranked drugs will be required to generalize this observation beyond this initial example.

We included the following text in the Discussion section (**lines 612-615**).

“At the same time, talarozole currently represents a single prospectively validated hit and should therefore be viewed as proof-of-concept example, with further testing of additional top-ranked candidates needed to generalize the framework's predictive performance.”

5. If the paper is accepted for publication, all code will need to be made publicly available to ensure transparency and reproducibility. Please confirm that the code will be shared accordingly.

In our work, we are fully committed in making all codes publicly available. Currently, the latest program code and input files are available at: (https://github.com/temporary-code-2025/manuscript_code/tree/main). The password for opening the zip file containing code files and documentation is: Manuscript2025Access! The restrictions will be removed as soon as the paper is accepted in order to make the code available to the scientific community.